# BENCHMARKING BIAS MITIGATION TOWARD FAIR-NESS WITHOUT HARM FROM VISION TO LVLMS

**Xuwei Tan[1], Ziyu Hu[2], Xueru Zhang[1]**
[1]The Ohio State University, [2]Stevens Institute of Technology
`tan.1206@osu.edu, zhu31@stevens.edu, zhang.12807@osu.edu`

## ABSTRACT

Machine learning models trained on real-world data often inherit and amplify biases against certain social groups, raising urgent concerns about their deployment at scale. While numerous bias mitigation methods have been proposed, comparing the effectiveness of bias mitigation methods remains difficult due to heterogeneous datasets, inconsistent fairness metrics, isolated evaluation of vision versus multi-modal models, and insufficient hyperparameter tuning that undermines fair comparisons. We introduce NH-Fair, a unified benchmark for fairness without harm that spans both vision models and large vision–language models (LVLMs) under standardized data, metrics, and training protocols, covering supervised and zero-shot regimes. Our key contributions are: (1) a systematic ERM tuning study that identifies training choices with large influence on both utility and disparities, yielding empirically grounded guidelines to help practitioners reduce expensive hyperparameter tuning space in achieving strong fairness and accuracy; (2) evidence that many debiasing methods do not reliably outperform a well-tuned ERM baseline, whereas a composite data-augmentation method consistently delivers parity gains without sacrificing utility, emerging as a promising practical strategy. (3) an analysis showing that while LVLMs achieve higher average accuracy, they still exhibit subgroup disparities, and gains from scaling are typically smaller than those from architectural or training-protocol choices. NH-Fair provides a reproducible, tuning-aware pipeline for rigorous, harm-aware fairness evaluation. Code: `https://github.com/osu-srml/NH-Fair`.

## 1 INTRODUCTION

Machine learning (ML) models increasingly shape high-stakes decisions, raising concerns that they replicate or amplify societal biases, leading to unfair outcomes across social groups. Various metrics have been proposed to quantify the model unfairness/bias such as *risk disparity* (Hashimoto et al., 2018), *demographic parity* (Dwork et al., 2012), *equal opportunity*, *equalized odds* (Hardt et al., 2016), *overall accuracy parity* (Berk et al., 2021), and *max-min fairness* (Lahoti et al., 2020), yet each captures different notions and priorities can depend on practical considerations.

Fairness interventions span pre-processing (Qraitem et al., 2023; Sagawa* et al., 2020; Pang et al., 2024; Jang et al., 2021), in-processing (Madras et al., 2018; Xu et al., 2021; Chuang and Mroueh, 2021; Park et al., 2022; Zafar et al., 2019; Zuo et al., 2024), and post-processing (Hardt et al., 2016; Yin et al., 2024; Dehdashtian et al., 2024; Jung et al., 2024; Tan et al., 2026) methods. But they often add complexity, require extensive hyperparameter tuning, or degrade overall accuracy (for example, a model may degrade performance for some groups to meet fairness criteria like demographic or accuracy parity). In safety-critical domains such as healthcare, sacrificing performance for fairness violates ethical principles of beneficence and non-maleficence (Beauchamp and Childress, 1994). Motivated by this, a growing line of work seeks *fairness without harm*, i.e., improving group parity without materially reducing performance for any group (Ustun et al., 2019; Martinez and Bertran, 2019; Yin et al., 2024). Instead of solely enforcing fairness constraints across different groups, these approaches ensure that model performance for every group does not deteriorate.

Despite significant progress, existing approaches were often designed for different problem settings and evaluated in inconsistent, limited environments, making it difficult to assess their general applicability. In particular, several key questions remain unanswered:

- **Comparability under a standardized protocol.** In fairness research, there are usually inconsistent experimental settings when comparing different methods, e.g., arbitrary choice of optimizer, different hyperparameters, and insufficiently trained baselines. These inconsistencies prevent us from accurately assessing the real utility and fairness performance of state-of-the-art debiasing methods. How do these methods perform in utility, fairness, and overhead, under the same setting with comprehensive hyperparameter sweeping? And can they outperform a carefully trained ERM?
- **What makes ERM strong—and how far can it go?** *There is a practice–research gap: industrial workflows rarely explore the full suite of fairness algorithms and instead prioritize hyperparameter optimization (HPO), like using Ray Tune (Liaw et al., 2018) or Optuna (Akiba et al., 2019), on incumbent models.* Yet, exhaustive HPO is computationally expensive. Can we bridge this gap by revealing which training decisions, such as optimizer, model depth, or augmentation, most impact fairness? In addition, if we start from a *powerful* ERM via principled tuning and selection, can conventional group-disparity mitigations still deliver fairness *without* materially compromising accuracy? Does simply scaling model size lead to better fairness?
- **Where do foundation and multimodal models stand?** In the era of foundation models and large pretrained models, are these multi-modal models already fair enough compared with specifically trained models due to pretraining on larger training data with larger model size?

Although some benchmarks have been proposed to make the comparison of previous methods, like MEDFAIR (Zong et al., 2023), FFB (Han et al., 2024), and ABC (Defrance et al., 2024). They often focus on specific domains, omit recent methodological advances, or suffer from limited hyperparameter tuning and dataset diversity. For example, MEDFAIR is restricted to medical datasets and does not evaluate fairness in general vision or multimodal contexts. FFB includes primarily older methods, omitting recent advances in representation learning and data-centric approaches, and lacks sufficient hyperparameter tuning, which may result in suboptimal models. ABCFair focuses on tabular datasets only and uses fixed hyperparameter settings, which limits scalability and may misrepresent method performance. The questions mentioned above are not yet fully addressed. Moreover, existing benchmarks (Xia et al., 2024; Jin et al., 2024a) generally treat classical vision and emerging multimodal models separately, limiting our understanding of how multimodal pretraining affects fairness relative to other models. We introduce these benchmarks in Appendix A.4.

To bridge this gap, we propose NH-Fair, a comprehensive benchmark for *fairness without harm* in complex image and multimodal settings. NH-Fair unifies evaluation across classical vision models and vision–language models, providing broader insights into fairness comparisons of architectures, pretraining strategies, and model scales. Beyond benchmarking both classical and recent fairness algorithms, we systematically evaluate the role of training choices, the overhead of existing methods, and the performance of state-of-the-art LVLMs. Our goal is to provide not just a benchmark, but actionable insights for developing and deploying fair ML systems. We summarize **contributions and key observations with practical implications for practitioners**:

1. **ERM and Training Choices Matter.** We investigate how training choices affect fairness.
   - *Observation.* Prior work often overlooks hyperparameter tuning, like optimizer and learning rate choice. We show that fixing hyperparameters across methods and datasets can yield unfair comparisons. Our findings challenge this practice, where some papers claim "state-of-the-art" fairness without sufficient evaluations, and underscore the need for more equitable and transparent evaluation protocols in future research.
   - *Takeaway.* Optimizer choice (e.g., SGD, Adam, AdamW, Adagrad) and its learning rate affect both fairness and utility. We recommend focusing on tuning resources here, as these settings have a clear impact. This also applies to selecting the correct pretrained weights, while model depth, batch size, or weight decay have less impact on fairness.

2. **Revisiting Mitigation Methods.** We offer an extensive comparison of recent and classic bias mitigation algorithms, assessing both fairness and accuracy to determine if current methods can achieve equitable performance without compromise.
   - *Observation.* Most fairness-specific algorithms do not significantly outperform a carefully tuned ERM when utilities are accounted for. **Data augmentation** is a simple yet effective strategy that often improves *both* fairness and utility.

- *Takeaway.* In practice, prioritize augmentation strategies before exploring a wide range of specialized algorithms; this pathway most often achieves fairness without utility loss or overhead.

3. **LVLMs Are Not Inherently More Fair** We extend evaluation to multi-modal models and LVLMs to probe whether large-scale, diverse pretraining reduces disparities or propagates stereotypes.
   - *Observation.* Despite greater data diversity and larger capacity, LVLMs still exhibit fairness issues comparable to task-specific vision models. For example, the Llama series, while achieving strong headline accuracy, remains highly susceptible to dataset-specific biases. In addition, the model scale shows a weak correlation with fairness improvements.
   - *Takeaway.* Practitioners concerned with fairness should prioritize evaluating different architectures and pretraining choices rather than relying on scaling up models alone.

## 2    PROBLEM FORMULATION: FAIRNESS WITHOUT HARM

In this paper, we evaluate each ML model based on two criteria: 1) *performance disparity* across different groups, and 2) *negative impact* on model performance caused by fairness interventions. Therefore, we consider the **fairness without harm** problem formulation, which aligns with our goal.

**Group fairness notion**    Consider a dataset of $n$ samples, where sample $i$ is represented as a triple $(x_i, y_i, a_i)$ from the joint distribution $P(X, Y, A)$. Here, $x_i \in \mathbb{R}^d$ is the feature vector, $y_i \in \mathcal{Y}$ is the label, and $a_i \in \mathcal{A}$ is the sensitive attribute (e.g., gender or race). We learn a model $h : \mathbb{R}^d \to \mathcal{Y}$ that achieves strong predictive performance while satisfying a fairness constraint. We evaluate performance using a risk function $R : \mathcal{Y} \times \mathcal{Y} \to \mathbb{R}_+$ that quantifies the discrepancy between predictions and labels. To assess disparity across groups, we consider the following fairness criteria.

- **Overall Accuracy Parity** requires that the classifier's accuracy be equal across groups:

$$\mathbb{P}\big[h(X) = Y \mid A = a\big] = \mathbb{P}\big[h(X) = Y \mid A = a'\big], \quad \forall a, a' \in \mathcal{A}. \tag{1}$$

We use the accuracy gap between the two groups to denote overall accuracy parity in this paper.

- **Demographic Parity**, also known as statistical parity, requires that the classifier's decisions be independent of the sensitive attribute:

$$\mathbb{P}[h(X) = y \mid A = a] = \mathbb{P}[h(X) = y \mid A = a'], \quad \forall a, a' \in \mathcal{A}, \ y \in \mathcal{Y}. \tag{2}$$

- **Equalized Odds** requires conditional independence of $h(X)$ and $a$ given the true label, e.g., the true positive rate and the false positive rate are equal across groups:

$$\mathbb{P}[h(X) = y \mid Y = y, A = a] = \mathbb{P}[h(X) = y \mid Y = y, A = a'], \quad \forall a, a' \in \mathcal{A}, \ y \in \mathcal{Y}. \tag{3}$$

- **Max-Min Fairness** uplifts disadvantaged groups by minimizing the worst group risk:

$$\max_{a \in \mathcal{A}} \ \mathbb{E}_{X, Y \mid A=a}\Big[R\big(h(X), Y\big)\Big]. \tag{4}$$

We do not consider individual fairness (Dwork et al., 2012) or counterfactual fairness (Kusner et al., 2017). The former requires a well-defined similarity function between individuals, which is hard to specify in images, while the latter usually assumes access to causal graphs, which are unavailable in most datasets. Given these constraints, we focus on group fairness.

**Fairness without harm**    Given group fairness notions defined in Sec. 2, enforcing these fairness constraints inevitably harms model performance. Consider Demographic Parity (DP) as an example, if the base rates of outcomes differ across groups ($\mathbb{P}\big[h(X) = y \mid A = a\big] \neq \mathbb{P}\big[h(X) = y \mid A = a'\big], \forall a, a' \in \mathcal{A}$.), enforcing independence between $h(X)$ and $A$ may require distorting predictions to align with group-agnostic rates. Suppose we have two groups and the optimal unconstrained classifier for group 0 satisfies $\mathbb{P}\big[h(X) = y \mid A = 0\big] = p_0$. To achieve DP, we must enforce $\mathbb{P}\big[h(X) = y \mid A = 0\big] = \mathbb{P}\big[h(X) = y \mid A = 1\big] = p$ for some value of $p$, which could force $p$ to deviate from the group-specific optima $p_0$, thereby increasing overall risk. In extreme cases, this might result in a trivial classifier (e.g. $h(X) = 1$ for all data) that satisfies DP but incurs maximal risk. Thus, fairness interventions risk creating a "race to the bottom," where fairness is not achieved by elevating disadvantaged groups by reducing accuracy for all groups. To avoid this, we adopt the

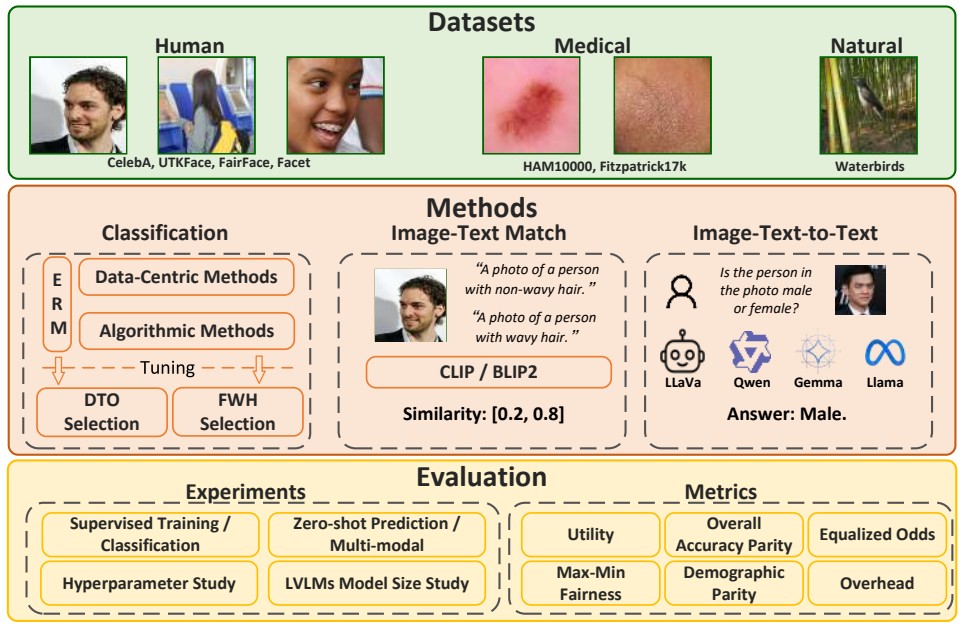

Figure 1: Overview of NH-Fair, evaluating fairness across domains, tasks, models, and methods.

principle of fairness without harm, which augments group fairness with a no-harm condition. Let $h_{\text{erm}}$ denote the baseline classifier trained via unconstrained empirical risk minimization (ERM):

$$h_{\text{erm}} = \arg\min_{h \in \mathcal{H}} \sum_{i=1}^{n} R\big(h(x_i), y_i\big). \tag{5}$$

The *no-harm* criterion requires that for every group $a \in \mathcal{A}$, the risk incurred by our fairness-enhanced classifier does not exceed that of the baseline:

$$\mathbb{E}_{X,Y|A=a}\Big[R\big(h(X), Y\big)\Big] \leq \mathbb{E}_{X,Y|A=a}\Big[R\big(h_{\text{erm}}(X), Y\big)\Big], \quad \forall a \in \mathcal{A}. \tag{6}$$

## 3 NH-FAIR BENCHMARK

### 3.1 DATASETS

We evaluate fairness algorithms on seven publicly available datasets spanning facial attributes, medical imaging, and spurious correlation tests: CelebA, UTKFace, FairFace, Facet, HAM10000, Fitz17k, and Waterbirds. Table 1 summarizes target tasks and sensitive attributes used in our evaluation. Due to the huge computational resource needs, we did not exhaustively use all available sensitive attributes (e.g., gender in UTKFace). Instead, we focused on attributes that exhibited the clear disparity in model predictions, enabling a more effective comparison of existing fairness methods.

**Criteria for Dataset Selection**. By definition, fairness evaluation requires demographic information (e.g., race, gender, or age) to measure disparities in model performance across subgroups. Therefore, our primary selection criterion is that datasets must include explicit demographic annotations. This ensures that fairness metrics are meaningful and grounded in socially relevant groups. Following this principle, we include the first six datasets. In addition, we considered: application domains, potential sources of bias, such as class imbalance or spurious correlations, and relevance to prior fairness studies. **Waterbirds** represents a pragmatic exception: it lacks demographic labels but has been widely used in fairness research due to its spurious correlations (e.g., background vs. object) resembling biases. Including it enables consistency with prior work (Reddy et al., 2021; Dehdashtian et al., 2024; Qiang et al., 2024) and provides a test for algorithms in non-demographic bias settings. However, we suggest carefully using domain generalization datasets (e.g., Waterbirds, Colored MNIST) in fairness evaluations since these datasets might be over-simplistic and cannot reflect the real challenge of the ML fairness issue. We discuss this dataset in Section 4.2.

**Potential Sources of Bias.** We use datasets that may introduce bias from multiple sources, including image quality, class imbalance, and spurious correlations. Importantly, each dataset often reflects more than one of these factors simultaneously, making it difficult to attribute disparities to a single cause. For this reason, we do not perform a dataset-level bias source analysis.

Table 1: Overview of the seven image datasets, detailing the number of samples, classification target, sensitive attribute(s), and approximate imbalance ratios expressed as percentages. In this table, M denotes Male, F denotes Female, W denotes White, B denotes Black, LH denotes Latino Hispanic, EA denotes East Asian, SA denotes Southeast Asian, IN denotes Indian, ME denotes Middle Eastern, N denotes the negative class, and P denotes the positive class.

| Dataset | # Samples | Target | Sensitive | Target Ratio | Sensitive Ratio |
|---|---|---|---|---|---|
| CelebA (Liu et al., 2015) | 200k | Wavy Hair | Gender | 68% (F) : 32% (M) | 41.7% (N) : 58.3% (P) |
| UTKFace (Zhang et al., 2017) | 23k | Gender | Race | 52% (M) : 48% (F) | 57.6% (W) : 42.4% (O) |
| FairFace (Karkkainen and Joo, 2021) | 100k | Ethnicity | Gender | 19.1% (W) : 14.1% (B) : 15.3% (LH) : 14.2% (EA) : 12.5% (SA) : 14.2% (IN) : 10.7% (ME) | 53% (M) : 47% (F) |
| Facet (Gustafson et al., 2023) | 30k | Visible Face | Gender | 67% (N) : 33% (P) | 76% (M) : 24% (F) |
| HAM10000 (Maron et al., 2019) | 10k | Malignant | Age | 86% (N) : 14% (P) | 71% (Young) : 29% (Old) |
| Fitz17k (Groh et al., 2021) | 17k | Malignant | Skin Type | 86.5% (N) : 13.5% (P) | 48% (Light) : 52% (Dark) |
| Waterbirds (Sagawa* et al., 2020) | 11k | Bird Species | Background | Train: 76.8% (N) : 23.2% (P) 
 Test: 77.8% (Water) : 22.2% (Land) | Train: 74.1% (N) : 25.9% (P) 
 Test: 50% (Water): 50% (Land) |

## 3.2 METHODS

We evaluate a diverse set of 12 baseline algorithms, spanning both fairness-specific and general-purpose approaches. To organize them, we group methods into two broad categories: **Data-Centric** and **Algorithmic** methods. **Data-Centric** methods focus on modifying the input distribution through data augmentation and sampling strategies, including RandAugment (Cubuk et al., 2020), Mixup (Zhang et al., 2018), Resampling (Buda et al., 2018; Sagawa* et al., 2020), Bias Mimicking (BM) (Qraitem et al., 2023), and FIS (Pang et al., 2024). **Algorithmic** methods, by contrast, alter the training process through adversarial training or fairness-aware objectives. These include Decoupled Classifier (Ustun et al., 2019; Wang et al., 2020), LAFTR (Madras et al., 2018), FSCL (Park et al., 2022), GapReg (Chuang and Mroueh, 2021), MCDP (Jin et al., 2024b), GroupDRO (Sagawa* et al., 2020), DFR (Kirichenko et al., 2023), and OxonFair (Delaney et al., 2024). Together, these methods span both established baselines and recent state-of-the-art techniques and include pre-processing, in-processing, and post-processing methods. Note that we do not consider causal fairness approaches since most of them assume access to causal graphs or intervention variables. However, these assumptions are rarely feasible in our datasets. A more detailed description of each method is provided in Appendix B.3.

Beyond supervised models, we also evaluate zero-shot predictions from multi-modal models such as CLIP (Radford et al., 2021) and BLIP2 (Li et al., 2023), as well as LVLMs, including LLaVA v1.6 (7B, 13B, 34B) (Liu et al., 2024), Qwen2.5-VL (7B, 32B, 72B) (Bai et al., 2025), Gemma 3 (4B, 12B, 27B) (Team et al., 2025), and Llama models (3.2–11B, 3.2–90B, and 4-Scout–109B) (Grattafiori et al., 2024; Meta, 2025) . For CLIP and BLIP2, we construct prompts for text-image matching tasks using pairs of positive and negative labels (e.g., *"A photo of a person with non-wavy hair."* vs. *"A photo of a person with wavy hair."*). For LVLMs, we frame image-text-to-text tasks that elicit binary responses, using prompts such as *"Is the person in the photo wavy-haired? Answer 'Yes' for wavy hair, 'No' for non-wavy hair."* Full prompt templates are provided in Appendix Table 8.

## 3.3 MODEL SELECTION

Selecting the best-performing model affects not only overall utility but also disparities across subgroups. Conventional selection methods, which prioritize average accuracy or loss, often reinforce majority-group performance while neglecting minorities. This motivates model selection strategies that explicitly balance utility and fairness. Prior work has explored approaches such as Minimax Pareto Selection (Martinez et al., 2020) and Distance to Optimal (DTO) (Han et al., 2022) to identify fair models. In this work, we adopt a DTO-based strategy to establish a strong ERM baseline, considering both utility and group fairness. We then take the selected ERM model as a baseline and introduce a fairness-without-harm (FWH) procedure to assess whether alternative methods can improve fairness without materially degrading utility compared to the well-established ERM.

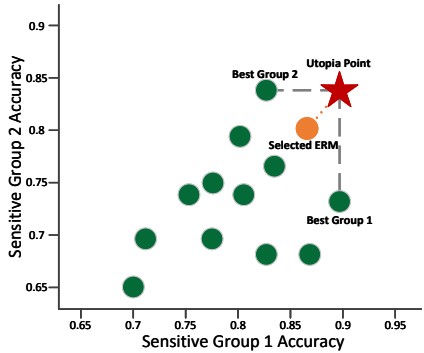 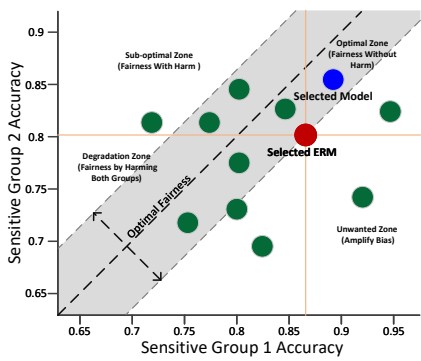

(a) DTO-based ERM model selection  (b) Fairness-Without-Harm Model Selection

Figure 2: Two-stage model selection. (a) First, each green dot represents a candidate ERM model's performance on two sensitive groups. We select the ERM model (orange circle) whose performance is closest to the utopia point (red star) in Euclidean distance. (b) With the ERM baseline established, we classify models trained by bias-mitigation methods into four zones based on how their subgroup performance compares to the ERM. Starting from the Optimal Zone and moving counterclockwise, we check whether any model is located in the shaded region, which demonstrates improved fairness.

**1) DTO-based selection.** The DTO strategy identifies the ERM model that minimizes Euclidean distance to **utopia point**, where each subgroup achieves its best observed performance. This approach aims to find the model that maximizes overall performance while minimizing disparities across subgroups. By establishing this model as the optimal ERM baseline, we provide a reference point for evaluating fairness-aware models. As shown in Figure 2a, the red star denotes the utopia point, where both subgroups reach their respective maximum performance. The model with the smallest DTO value (the shortest distance to the utopia point) is selected as the ERM baseline.

**2) Fairness-without-harm selection.** Given the optimal ERM, we evaluate other methods by comparing subgroup-specific accuracies. Based on the group accuracy of the selected ERM model, we categorize the models from other methods into four distinct zones:

- **Optimal** (Fairness Without Harm): Models here already achieve better utilities for both groups compared with the established baseline ERM. A smaller accuracy gap is desirable to show fairness. Thus, we select the model with the smallest accuracy gap.
- **Sub-optimal** (Fairness by Compromising): Models reduce disparities by decreasing the advantaged group's utility while improving the disadvantaged group. If no candidates exist in **Optimal** zone, we select the model here with the *smallest accuracy gap* to evaluate how it equalizes two groups.
- **Degradation** (Fairness by Harming Both Groups): Models are undesirable since they degrade performance for all groups, achieving fairness at the expense of overall utility. We select a model with the *smallest L2 distance to the optimal ERM* to maintain necessary utility rather than fairness metrics, since the models can significantly decrease utility (random guess) to equalize accuracy.
- **Unwanted**: Models in this zone widen disparities by benefiting the advantaged group and harming the disadvantaged group, which exacerbate unfairness and are not considered.

When evaluating fairness-aware models against the optimal ERM, we follow the selection order: **Optimal → Sub-optimal → Degradation**. This ensures fairness while minimizing performance trade-offs. Using this strategy, we select the model that best meets the fairness-without-harm criteria.

## 4 RESULTS

**Implementation.** For all methods, we independently search for the best hyperparameters on each dataset. Specifically, we search over optimizers (SGD and Adam), learning rates, weight decay values, and method-specific hyperparameters. Implementation details are provided in Appendix B.4 and B.5. In total, we spent over 10,000 A100 GPU hours to obtain these benchmarking results.

**Metrics.** We use **Accuracy (ACC)** to evaluate utility on most datasets, while **AUC** is adopted for disease prediction tasks, since medical datasets often exhibit class imbalances. For fairness, we report

four metrics: **Overall Accuracy Parity (Gap ↓)**, **Max–Min Fairness (Worst ↑)**, **Demographic Parity (DP ↑)**, and **Equalized Odds (EqOdd ↑)**. They are introduced in Appendix B.2.

## 4.1 HOW TO TUNE A FAIR ERM AND WHY TRAINING CHOICES MATTER

This part is motivated by two observations. First, ML practitioners rarely conduct extensive testing of different fairness-specific algorithms and instead focus on tuning existing models. Yet exhaustive hyperparameter search is computationally expensive, raising the question: *which training choices should be prioritized for fairness-sensitive applications?* Second, many prior studies report fairness or utility gains while keeping core training parameters fixed across baselines and datasets. *But can such practices truly yield fair and reliable comparisons?*

To answer these, we conduct a systematic study under different training setups (see Appendix C.1). We find that some choices, most notably initialization (pretraining vs. scratch) and optimizer, consistently impact the fairness–utility trade-off. For example, pretrained models often improve utility without harming disadvantaged groups, and the right optimizer (e.g., SGD on CelebA, Adam on Fitz17k) can shift both utility and subgroup performance. In contrast, factors like batch size, weight decay, or model depth have weaker or inconsistent effects. **These findings suggest that impactful training choices could be prioritized during HPO. In addition, fixing such parameters across baselines and datasets can lead to biased comparisons. It underscores the need for more equitable and transparent evaluation practices in future fairness research.**

## 4.2 REVISITING MITIGATION METHODS

**A well-tuned ERM baseline could match or surpass fairness-specific algorithms.** Table 2 reports utility and fairness outcomes for single-modality models across seven datasets. For each dataset–method pair, results are averaged over five runs, with the best ERM model selected under DTO criteria and the best models for mitigation methods selected under FWH criteria. Table 3 extends the comparison to multimodal models. Despite being treated as a "basic" base-line, a well-tuned ERM, selected via DTO, consistently performs competitively across fairness metrics (Gap, Worst, DP, EqOdd). Figure 3 shows that ERM performs competitively with, or sometimes outperforms, specialized fairness methods under FWH selection, and no single mitigation approach dominates across all datasets. We further conduct a Friedman test across all datasets, followed by a Nemenyi post-hoc analysis. We visualize the results using Critical Difference (CD) plots in Figure 4, where methods connected by a horizontal bar are not significantly different under the Nemenyi test.

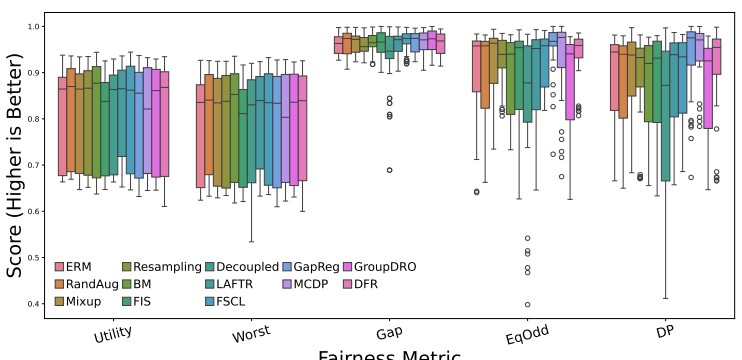

Figure 3: Comparative Analysis of Bias Mitigation Methods. OxonFair is excluded from here due to missing FairFace results.

**Data augmentation is a simple yet effective path to fairness without harm.** Our results show that fairness improvements do not inherently require sacrificing accuracy. Prior studies like Dutta et al. (2020) established the theoretical possibility of achieving both. In the paper, **RandAug**, though not designed for bias mitigation, improves both fairness and accuracy across multiple datasets, empirically demonstrating that fairness improvements do not always require a trade-off in model performance. This suggests that increasing data variability can naturally mitigate biases, offering a lightweight and practical strategy for fairness-aware training without the need for explicit fairness constraints.

**The utility–fairness trade-offs are pronounced in regularization-based methods.** Both GapReg and MCDP account for explicit fairness constraints in the loss functions, directly penalizing subgroup disparities. This design explains why they consistently achieve strong scores on fairness metrics

Table 2: Average utility and fairness results (**standard deviations are reported** in Appendix Table 20). Results better than ERM are highlighted. We also report the number of selected models located in each zone by the FWH selection in "Optimal|Sub-optimal|Degradation|Unwanted" on the validation set, and further verify these findings on the test set.

| Dataset | Metric | ERM | RandAug | Mixup | Resampling | BM | FIS | Decoupled | LAFTR | FSCL | GapReg | MCDP | GroupDRO | DFR | OxonFair |
|---|---|---|---|---|---|---|---|---|---|---|---|---|---|---|---|
| **CelebA** | ACC | 86.57 | **86.72** | 85.61 | 86.35 | 85.93 | 83.05 | 86.35 | 86.55 | 85.61 | 85.62 | 80.26 | 86.12 | **86.58** | 86.49 |
| | Worst | 83.76 | **83.89** | 82.74 | 83.44 | 82.86 | 79.33 | 83.46 | 83.67 | 82.56 | 83.17 | 77.13 | 83.50 | **83.78** | 83.63 |
| | Gap | 6.76 | 6.80 | 6.90 | 6.98 | 7.38 | 8.94 | 6.93 | 6.93 | 7.35 | **5.90** | 7.52 | **6.31** | **6.74** | 6.87 |
| | EqOdd | 81.91 | 81.73 | **87.08** | 81.80 | 78.84 | 75.79 | 80.59 | 81.15 | **85.45** | **93.94** | **89.63** | 78.73 | 81.83 | 78.10 |
| | DP | 67.20 | **67.37** | **70.83** | 67.39 | 66.91 | 66.84 | 66.68 | 66.90 | **69.72** | **75.91** | **93.11** | 66.41 | 67.30 | 65.10 |
| **UTKFace** | ACC | 92.75 | **93.19** | 92.62 | 92.70 | **93.33** | 91.97 | 91.68 | **93.17** | **93.52** | 92.53 | 92.49 | 92.45 | 92.73 | 92.36 |
| | Worst | 91.78 | **92.19** | 91.55 | 91.60 | **92.27** | 90.91 | 90.84 | **92.05** | **92.62** | 91.70 | 91.63 | 91.41 | 91.60 | 91.11 |
| | Gap | 2.26 | 2.34 | 2.50 | 2.56 | 2.47 | 2.48 | **1.97** | 2.61 | **2.10** | **1.91** | **2.00** | 2.44 | 2.63 | 2.91 |
| | EqOdd | 97.62 | **97.62** | 97.61 | 97.49 | 97.39 | 97.51 | 97.43 | 97.44 | 97.44 | **98.10** | **98.04** | 97.06 | 97.39 | 96.41 |
| | DP | 94.55 | **94.83** | 94.51 | **95.34** | 94.34 | **94.69** | 96.27 | 95.44 | 94.18 | 95.30 | 95.80 | 94.78 | 94.83 | 94.68 |
| **FairFace** | ACC | 66.76 | **68.37** | 65.40 | 65.40 | 65.66 | 65.31 | **67.03** | 66.44 | 65.42 | 65.02 | 66.06 | 65.51 | 63.20 | – |
| | Worst | 66.34 | **67.69** | 64.50 | 64.51 | 65.20 | 64.59 | **66.61** | 65.60 | 64.64 | 64.12 | 65.62 | 65.22 | 62.45 | – |
| | Gap | 0.87 | 1.44 | 1.93 | 1.90 | 0.97 | 1.53 | **0.87** | 1.76 | 1.66 | 1.92 | 0.90 | **0.60** | 1.59 | – |
| | EqOdd | 96.22 | 96.14 | **96.55** | **96.83** | 95.70 | 95.88 | 95.14 | 94.73 | **97.05** | 96.15 | **97.85** | **96.31** | 95.81 | – |
| | DP | 97.61 | 97.55 | **97.62** | **97.80** | 97.30 | 97.40 | 96.88 | 96.86 | **98.10** | 97.51 | **98.50** | 97.47 | 97.43 | – |
| **Facet** | ACC | 67.55 | 67.83 | 67.86 | 67.56 | 65.87 | 67.60 | 67.33 | 70.74 | 67.79 | 67.01 | **67.91** | 67.20 | 66.87 | **68.09** |
| | Worst | 64.25 | **64.94** | 64.54 | 64.13 | 62.67 | 63.53 | 62.60 | **68.60** | 65.02 | 62.22 | 64.21 | 64.07 | 63.10 | 64.11 |
| | Gap | 4.31 | **3.78** | 4.33 | 4.48 | **4.18** | 5.33 | 6.17 | **2.82** | **3.61** | 6.26 | 4.84 | **4.08** | 4.92 | 5.21 |
| | EqOdd | 96.47 | 96.68 | 97.50 | 94.37 | 96.32 | **98.10** | 88.04 | 96.38 | 96.50 | **98.92** | **98.23** | 93.76 | 96.82 | 83.40 |
| | DP | 95.40 | 95.91 | 96.71 | 93.96 | 95.09 | **97.84** | 87.30 | 96.00 | 95.66 | **98.73** | **98.46** | 93.16 | 96.09 | 83.95 |
| **HAM10000** | AUC | 88.35 | **89.09** | 86.51 | 87.75 | **89.54** | 85.97 | 87.87 | 86.71 | **89.40** | 84.97 | 82.96 | 87.66 | 87.06 | **88.46** |
| | Worst | 84.67 | **84.67** | 82.31 | **84.77** | **86.49** | 82.98 | 84.04 | 81.68 | **85.89** | 82.57 | 80.29 | 83.98 | 82.49 | 83.83 |
| | Gap | 4.11 | 4.99 | 4.14 | **3.52** | **3.04** | **3.11** | 5.17 | 6.18 | **3.71** | **3.07** | **3.10** | 4.98 | 5.30 | 5.46 |
| | EqOdd | 86.79 | 88.43 | 91.14 | 91.84 | 91.84 | 94.05 | 77.52 | 93.28 | 84.57 | 99.52 | 90.94 | 91.25 | | 99.27 |
| | DP | 81.48 | 84.73 | 88.54 | 85.05 | 78.41 | 88.58 | 75.15 | 91.00 | 77.64 | 96.74 | 99.58 | 83.86 | 84.65 | 99.34 |
| **Fitz17k** | AUC | 89.74 | 91.29 | 90.62 | 90.76 | 91.02 | 88.34 | 89.63 | 90.95 | 90.71 | 89.59 | 91.65 | 90.72 | 89.99 | 89.56 |
| | Worst | 88.39 | 90.15 | 89.38 | 88.99 | 89.93 | 87.02 | 88.45 | 89.67 | 89.77 | 88.52 | 90.49 | 90.06 | 88.57 | 88.40 |
| | Gap | 2.92 | 2.51 | 2.43 | 3.62 | 2.34 | 3.06 | 2.55 | 2.87 | 2.22 | 1.84 | 2.87 | 1.92 | 2.93 | 3.06 |
| | EqOdd | 94.92 | 95.61 | 96.20 | 95.27 | 94.99 | 95.17 | 94.09 | 93.68 | 97.45 | 95.47 | 95.68 | 94.36 | 94.30 | 99.26 |
| | DP | 94.46 | 94.53 | 94.51 | 93.31 | 93.66 | 95.60 | 94.06 | 94.46 | 95.32 | 96.14 | 95.24 | 95.26 | | 99.88 |
| **Waterbirds** | ACC | 85.63 | 86.09 | 87.67 | 87.35 | 88.20 | 83.72 | 74.64 | 85.72 | 86.83 | 86.45 | 85.98 | 85.46 | 89.83 | 90.27 |
| | Worst | 84.20 | 84.52 | 85.99 | 84.85 | 85.96 | 82.67 | 64.45 | 83.94 | 86.28 | 85.72 | 84.83 | 84.45 | 89.09 | 89.52 |
| | Gap | 2.87 | 3.14 | 3.36 | 4.98 | 4.48 | 2.09 | 20.38 | 3.56 | 1.10 | 1.47 | 2.31 | 2.02 | 1.47 | 1.50 |
| | EqOdd | 66.53 | 68.99 | 81.42 | 90.87 | 77.21 | 65.89 | 47.31 | 68.22 | 90.00 | 87.48 | 72.97 | 67.31 | 97.79 | 94.99 |
| | DP | 77.67 | 77.25 | 86.00 | 90.93 | 81.78 | 75.30 | 52.30 | 77.47 | 92.53 | 91.39 | 80.70 | 76.91 | 98.61 | 97.38 |
| **FWH Selection** | Validation | - | 7|0|0|0 | 4|1|2|0 | 3|2|2|0 | 3|2|2|0 | 2|1|4|0 | 1|2|4|0 | 4|3|0|0 | 6|0|1|0 | 4|1|2|0 | 4|1|2|0 | 2|3|2|0 | 7|0|0|0 | 0|0|6|0 |
| | Test | - | 7|0|0|0 | 3|0|3|1 | 2|1|2|2 | 3|1|3|0 | 0|0|6|1 | 1|1|3|2 | 3|0|1|3 | 4|1|2|0 | 0|2|5|0 | 2|0|4|1 | 1|1|5|0 | 2|1|3|1 | 1|1|2|2 |
| | Match | - | 7/7 | 5/7 | 3/7 | 5/7 | 3/7 | 3/7 | 3/7 | 5/7 | 2/7 | 4/7 | 4/7 | 2/7 | 3/7 |

(a) Utility     (b) Worst     (c) Gap

(d) EqOdd     (e) DP

Figure 4: Critical difference plots comparing methods across five metrics. Lower ranks indicate better performance. OxonFair is excluded since it does not support multi-class classification on FairFace.

such as EqOdd and DP across datasets. However, the fairness penalty can pull decision boundaries away from the utility-optimal surface, which often results in lower accuracy and occasionally lower Worst-group accuracy. They highlight the classic fairness–utility trade-off: fairness comes at the expense of overall predictive strength. In contrast, the contrastive learning method FSCL, which encourages representations of the same class to cluster across different groups, delivers strong

fairness improvements while keeping accuracy competitive, implying that learning fair and robust representations is also a promising path to fairness without harm.

**Rethinking spurious correlation datasets for fairness evaluations.** In prior studies (Dehdashtian et al., 2024; Qiang et al., 2024; Reddy et al., 2021), datasets with spurious correlations are often used for fairness evaluation since spurious correlations resemble the types of confounding factors that lead to fairness issues. We noticed that, in Table 2, many methods (e.g., Mixup, Resampling, BM, OxonFair...) can achieve both higher utility and fairness on Waterbirds, but it is more difficult to achieve such a gain on fairness datasets with socially meaningful group disparities. This raises a potential concern: spurious correlation might be easier to resolve than fairness-specific distribution shifts. Fairness involves systematic performance gaps across protected groups, which are typically more subtle and persistent than background–object correlations. Over-reliance on datasets like Waterbirds may therefore underestimate the difficulty of fairness challenges and overstate algorithmic effectiveness. While useful for studying robustness to spurious correlations, such datasets are not ideal for fairness evaluation. We recommend using datasets that provide coherent and well-justified data groupings, preferably with clear, socially meaningful attributes (especially for readers from social science domains), and exercising caution when incorporating other datasets with data shift (e.g., domain generalization benchmarks) for fairness assessment.

## 4.3 FAIRNESS IN MULTI-MODAL MODELS

We showed that even the best-tuned vision models leave non-trivial subgroup gaps. A natural hypothesis is that these gaps reflect limited data diversity and model capacity. Here, we evaluate pre-trained multi-modal models, which are trained on massive, diverse datasets and are expected to generalize better. Specifically, we consider two families: (1) image–text matching models such as CLIP, BLIP-2, and two CLIP-based debiasing variants (FairerCLIP Dehdashtian et al. (2024), SFID Jung et al. (2024)); and (2) LVLMs such as LLaVA-1.6, Qwen2.5-VL, Gemma 3, and Llama. All models are evaluated in zero-shot predictions. Fig. 5 visualizes the joint landscape of utility and parity, where *Qwen2.5-VL 72B* lies closest to the outer envelope across datasets, indicating the best overall balance between high accuracy and low disparity among the evaluated LVLMs.

**Fairness persists as a challenge.** From Table 3, we found that fairness outcomes diverge sharply between image–text matchers and newer large models. BLIP-2 and CLIP still exhibit sizable subgroup gaps on CelebA and Waterbirds. Even though debiasing methods like FairerCLIP and SFID can use the validation set to tune the model, they still don't significantly address the fairness issue. As for LVLMs, despite being trained on massive, heterogeneous corpora, multimodal models do not universally outperform carefully tuned unimodal baselines in either utility or fairness. On relatively balanced datasets like UTKFace, LVLMs such as Qwen2.5-VL and Gemma-3 deliver strong accuracy while keeping subgroup gaps small.

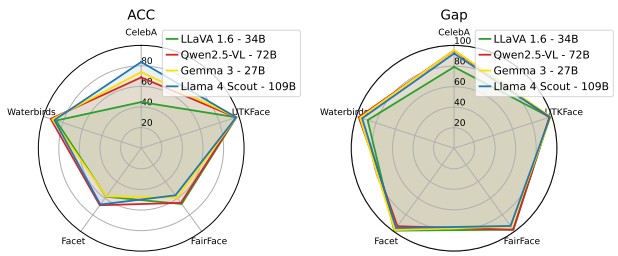

Figure 5: LVLM performance using ACC and $100 -$ Gap. Other metrics are presented in Figure 11.

However, on more challenging datasets (CelebA and Facet), they exhibit subgroup disparities, often with worse worst-group accuracy than ERM and sometimes a larger accuracy gap for certain models. These results emphasize that multi-modal models still inherit and sometimes amplify fairness challenges rather than resolving them.

**Scaling is not enough.** We also studied different LVLM sizes to evaluate whether larger LVLMs could achieve better fairness on our tasks (Appendix Table 16, simplified in Table 4). Increasing LVLM size improves average accuracy but does not consistently resolve fairness gaps. Larger models (e.g., Gemma-3–27B, Llama3.2-90B) achieve higher average accuracy compared to their smaller counterparts, yet subgroup disparities (Gap) remain non-trivial and in some cases even widen. The fairness gains from scaling are much smaller than those obtained by switching to a different model

family, suggesting that the training protocol plays a more decisive role in fairness. Hence, in fairness-sensitive applications, we recommend first exploring model choice before allocating resources to scaling.

Table 3: Comparison of multi-modal models. **Standard deviations** are in Appendix Table 21. We omit HAM10000 and Fitz17k for LVLMs, since they do not yield calibrated probabilities for AUC.

| Dataset | Metric | ERM | RandAug | BLIP2 | CLIP | Fairer-CLIP | CLIP-SFID | LLaVA-1.6 34B | Qwen2.5-VL 72B | Gemma3 27B | Llama4 Scout |
|---|---|---|---|---|---|---|---|---|---|---|---|
| CelebA | ACC | 86.57 | **86.72** | 47.38 | 74.07 | 73.78 | 72.05 | 44.83 | 68.76 | 74.04 | 83.71 |
| | Worst | 83.76 | **83.89** | 36.82 | 67.43 | 67.32 | 66.36 | 32.69 | 66.02 | 72.18 | 80.54 |
| | Gap | 6.76 | 6.80 | 18.09 | 15.97 | 15.56 | 13.69 | 20.75 | **4.70** | **4.47** | 7.62 |
| | EqOdd | 81.91 | 81.73 | **97.24** | 83.72 | 83.79 | 95.70 | **97.41** | 92.69 | 92.76 | 84.81 |
| | DP | 67.20 | **67.37** | 95.90 | 81.32 | 81.06 | 93.23 | 91.92 | 91.71 | 74.53 | 71.52 |
| UTKFace | ACC | 92.75 | **93.19** | 94.23 | 96.72 | 96.79 | 96.70 | **97.12** | 96.78 | 97.25 | 97.02 |
| | Worst | 91.78 | 92.19 | 94.00 | 95.90 | 96.05 | 96.03 | 96.34 | 95.76 | 96.89 | 96.29 |
| | Gap | 2.26 | 2.34 | **0.45** | 1.90 | 1.72 | 1.55 | 1.81 | 2.36 | 0.85 | 1.70 |
| | EqOdd | 97.62 | 97.62 | **99.24** | 97.96 | 98.17 | 98.36 | 98.16 | 97.59 | 99.20 | 98.27 |
| | DP | 94.55 | **94.83** | 95.61 | 93.91 | 94.27 | 94.96 | 95.22 | 94.33 | 95.28 | 94.90 |
| FairFace | ACC | 66.76 | **68.37** | 52.57 | 57.36 | 56.81 | 52.73 | **66.91** | 65.57 | 59.40 | 56.52 |
| | Worst | 66.34 | **67.69** | 50.74 | 57.20 | 56.41 | 51.59 | 66.04 | 64.73 | 56.71 | 53.53 |
| | Gap | 0.87 | 1.44 | 3.46 | **0.34** | **0.86** | 2.40 | 1.84 | 1.79 | 5.71 | 6.35 |
| | EqOdd | 96.22 | 96.14 | 93.86 | **97.16** | 96.78 | 97.14 | 96.22 | **96.26** | 95.13 | **96.54** |
| | DP | 97.61 | 97.55 | 96.89 | **98.36** | 98.32 | 98.46 | **98.07** | 97.57 | 97.16 | **98.00** |
| Facet | ACC | 67.55 | **67.83** | 41.16 | 33.10 | 33.17 | 33.26 | 58.14 | **68.41** | 57.75 | 67.54 |
| | Worst | 64.25 | **64.94** | 40.40 | 31.43 | 31.69 | 31.54 | 57.97 | 63.79 | 57.25 | **64.61** |
| | Gap | 4.31 | **3.78** | **3.22** | 7.13 | 6.32 | 7.37 | **0.72** | 6.04 | **1.42** | **3.83** |
| | EqOdd | 96.47 | **96.68** | 93.52 | **98.72** | **98.82** | **99.75** | 95.90 | **97.04** | 96.22 | **97.55** |
| | DP | 95.40 | 95.91 | 94.10 | **98.92** | **99.00** | **99.80** | 93.91 | **96.15** | 94.56 | **96.28** |
| Waterbirds | ACC | 85.63 | **86.09** | 52.30 | 78.05 | 77.77 | 75.70 | **86.95** | **92.41** | 90.77 | **88.62** |
| | Worst | 84.20 | **84.52** | 39.18 | 74.53 | 73.49 | 72.81 | 81.26 | **91.27** | 88.97 | 85.74 |
| | Gap | 2.87 | 3.14 | 26.23 | 7.04 | 8.56 | 5.78 | 11.39 | **2.28** | 3.59 | 5.76 |
| | EqOdd | 66.53 | **68.99** | 68.24 | 73.96 | 73.19 | 95.57 | 80.88 | 94.03 | 92.99 | 89.38 |
| | DP | 77.67 | 77.25 | 63.48 | **78.12** | 76.73 | 94.09 | 80.46 | 94.62 | 93.17 | 89.44 |
| HAM10000 | AUC | 88.35 | **89.09** | 38.87 | 52.15 | 51.88 | 52.53 | – | – | – | – |
| | Worst | 84.68 | 84.67 | 39.00 | 51.56 | 52.01 | 53.41 | – | – | – | – |
| | Gap | 4.11 | 4.99 | 6.89 | 4.14 | **3.12** | **2.24** | – | – | – | – |
| | EqOdd | 88.17 | **88.43** | 98.19 | 96.24 | 95.75 | 96.19 | – | – | – | – |
| | DP | 82.22 | 84.73 | 97.89 | 99.09 | 98.94 | 98.32 | – | – | – | – |
| Fitz17k | AUC | 89.74 | **91.29** | 67.08 | 69.92 | 69.81 | 69.37 | – | – | – | – |
| | Worst | 88.39 | **90.15** | 66.46 | 69.78 | 69.85 | 69.35 | – | – | – | – |
| | Gap | 2.92 | **2.51** | 3.74 | **2.31** | **2.52** | 4.73 | – | – | – | – |
| | EqOdd | 94.92 | **95.61** | 97.06 | 89.87 | 87.95 | 85.88 | – | – | – | – |
| | DP | 94.46 | **94.53** | 98.40 | 95.03 | 93.02 | 92.19 | – | – | – | – |

Table 4: Effect of LVLM model scale on accuracy and fairness metrics.

| Dataset | Metric | LLaVA-1.6 | | | Qwen2.5-VL | | | Gemma 3 | | | Llama | | |
|---|---|---|---|---|---|---|---|---|---|---|---|---|---|
| | | 7B | 13B | 34B | 7B | 32B | 72B | 4B | 12B | 27B | 3.2-11B | 3.2-90B | 4-Scout-109B |
| Average | ACC | 68.16 | 69.87 | 70.79 | 77.36 | 72.72 | 78.39 | 67.33 | 73.48 | 75.84 | 74.54 | 76.03 | **78.68** |
| | Worst | 66.10 | 66.55 | 66.86 | 75.65 | 68.18 | **76.31** | 61.58 | 72.60 | 74.40 | 69.32 | 70.51 | 76.14 |
| | Gap | 3.89 | 5.70 | 7.30 | 2.89 | 7.66 | 3.43 | 10.53 | **2.09** | 3.21 | 10.08 | 11.09 | 5.05 |
| | EqOdd | 87.91 | 89.81 | 93.71 | 94.98 | **95.70** | 95.52 | 89.09 | 93.68 | 95.26 | 88.01 | 91.32 | 93.31 |
| | DP | 87.05 | 88.60 | 91.92 | 91.94 | **96.16** | 94.88 | 86.31 | 90.47 | 90.94 | 84.66 | 87.91 | 90.03 |

## 5 CONCLUSION

In this study, we introduce NH-Fair, a rigorously curated benchmark for evaluating fairness interventions in image models, and show that AI fairness issues remain challenging in computer vision domains, even as new methods are proposed and model capacity continues to increase. A carefully tuned ERM with the hyperparameter search often rivals specialized debiasing methods. It highlights the crucial role of hyperparameter tuning and model selection in achieving fairness without harm. We further find that utility need not be sacrificed: data augmentation can deliver simultaneous gains in accuracy and subgroup parity. In addition, large vision–language models are not exempt from fairness issues. Their pre-training choices can still imprint spurious correlations that widen gaps. By releasing NH-Fair and all accompanying code, we aim to make fairness results reproducible and to provide the community with a solid baseline on which to build more robust, bias-aware methods.

## ETHICS STATEMENT

This work focuses on evaluating fairness in vision and multimodal models. While our study does not involve direct interaction with human subjects, it makes extensive use of publicly available datasets with demographic information. We acknowledge that these datasets may contain label noise, spurious correlations, or demographic imbalances, which themselves reflect broader social biases. Demographic prediction tasks (e.g., gender or race classification) are used solely for academic analysis of fairness interventions and are not intended for deployment or endorsement in real-world applications.

## REPRODUCIBILITY STATEMENT

We have taken several steps to ensure the reproducibility of our work. All datasets used in our experiments are publicly available, with detailed descriptions provided in Appendix B.1. Model architectures, training procedures, and hyperparameters are documented in Appendix B.4 and Appendix B.5. The definitions and formulas of all fairness metrics are included in Section 2 and Appendix B.2. We also provide an anonymous code repository linked in the manuscript.

## ACKNOWLEDGEMENTS

This work was funded in part by the National Science Foundation under award number IIS-2202699, IIS-2416895, IIS-2301599, and CMMI-2301601.

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

APPENDIX

# A    RELATED WORK

## A.1    FAIRNESS NOTIONS

Various notions have been proposed in the ML literature to measure the unfairness of ML outcomes; they can be roughly classified into the following categories:

- **Unawareness**: it prohibits the use of sensitive attributes in the training and decision-making under the principle that excluding such features avoids direct discrimination (Dwork et al., 2012).

- **Parity-based fairness**: it requires certain statistical measures to be equalized across different groups (Zhang and Liu, 2021; Zhang et al., 2020). Prominent examples include Demographic Parity (predicted positive rates should be similar across groups) (Dwork et al., 2012), Equal Opportunity (true positive rates are aligned) (Hardt et al., 2016), Equalized Odds (both false positive and false negative rates are aligned) (Hardt et al., 2016), Predictive Parity (predictive value measures are balanced) (Chouldechova, 2017), Accuracy Parity (overall accuracy remains comparable across groups) (Khalili et al., 2023; Zhang et al., 2019; Jin et al., 2026), etc.

- **Preference-based fairness**: inspired by the fair-division and envy-freeness literature in economics, it ensures that given the choice between various sets of decision outcomes, every group of users would collectively prefer its perceived outcomes, regardless of the (dis)parity compared to the other groups (Zafar et al., 2017; Ustun et al., 2019).

- **Counterfactual fairness:** this notion leverages tools from causal inference and structural causal models to define fairness at the level of individual causal pathways. Intuitively, a model is counter-factually fair if for any individual, the predicted outcome remains unchanged in a "counterfactual world" where the individual belonged to a different demographic group, while all other non-sensitive attributes and causal mechanisms remain fixed (Kusner et al., 2017; Zuo et al., 2023).

- **Individual fairness**: unlike other notions that ensure fairness at the group level, individual fairness attains fairness at the individual level, by ensuring similar individuals are treated similarly (Dwork et al., 2012).

Our work is mostly related to parity-based fairness. Unlike most existing methods that achieve fairness at the cost of reducing accuracy, we aim to achieve fairness without harm. By assigning group-specific models to different groups, our goal is to reduce the accuracy gap between different groups without sacrificing accuracy for any group (compared to the baseline classifier trained on data collected from all groups).

## A.2    APPROACHES TO MITIGATING UNFAIRNESS

A large body of research has focused on mitigating bias in ML models, which can be broadly categorized into three strategies: pre-processing, in-processing, and post-processing interventions.

- **Pre-processing methods** aim to reduce unfairness at the data level. Common approaches include re-weighting or re-sampling training examples to balance demographic groups (Kamiran and Calders, 2012; Qraitem et al., 2023; Sagawa* et al., 2020; Pang et al., 2024), generating synthetic data to fill representation gaps (Jang et al., 2021), or learning fair representations through data transformations (Celis et al., 2020; Chuang and Mroueh, 2021).

- **In-processing methods** modify the training procedure by directly incorporating fairness constraints. Representative approaches include adversarial training, which encourages representations to be invariant to sensitive attributes (Madras et al., 2018; Xu et al., 2021; Jin et al., 2024b; Pham et al., 2023), and fairness-based regularization terms in the loss function (Chuang and Mroueh, 2021; Park et al., 2022; Zafar et al., 2019). More recent work has leveraged contrastive learning (Shen et al., 2021; Park et al., 2022; Wang et al., 2022) or disentangled representations (Creager et al., 2019; Park et al., 2021; Lee et al., 2021) to de-bias learned features.

- **Post-processing methods** adjust model outputs to better satisfy fairness criteria without retraining (Khalili et al., 2021a;b). Examples include modifying decision thresholds (Hardt et al., 2016), calibrating prediction scores (Pleiss et al., 2017), or applying confidence-based and surrogate adjustments (Menon and Williamson, 2018; Yin et al., 2024).

Recently, fairness in multi-modal learning has gained increasing attention, especially due to the rising use of vision-and-language models. Several approaches (Dehdashtian et al., 2024; Jung et al., 2024) have also been proposed to mitigate bias in multi-modal settings. For example, FairCLIP (Luo et al., 2024) minimizes Sinkhorn distance between the two distributions to debias pre-trained vision-language models. (Seth et al., 2023) ensures fair representation from learning additive residual image representations. (Gerych et al., 2024) equalizes the distances between the debiased embeddings and images to achieve test-time fairness for VLMs.

In this paper, we mainly employ pre-processing and in-processing methods (with one post-processing technique Kirichenko et al. (2023)). For clarity of presentation, we therefore re-group these strategies into two broader categories: data-centric methods, which intervene at the dataset or input level, and algorithmic methods, which modify the training process or outputs. This framing better reflects the scope of our study and emphasizes the practical trade-offs practitioners face when choosing between data-level and model-level interventions.

### A.3 FAIRNESS WITHOUT HARM

Achieving fairness at the cost of lowering the performance of any group may be undesirable and even prohibited in certain applications. For example, in safety-critical domains such as healthcare, sacrificing model accuracy for fairness is undesirable, as it violates both the beneficence (i.e., doing what is best for patients) and non-maleficence (i.e., avoiding harm) principles in healthcare ethics (Beauchamp and Childress, 1994). Given this concern, some works aim to achieve fairness without negatively impacting model accuracy (Dutta et al., 2020; Ustun et al., 2019; Yin et al., 2024; Pang et al., 2024; Cai et al., 2025; Tan et al., 2026). Instead of solely enforcing fairness constraints across different groups, these approaches ensure that model performance for every group does not deteriorate. For example, Martinez and Bertran (2019) seeks a Pareto-optimal fair ML model that minimizes performance gaps among groups while preventing *unnecessary* harm (i.e., minimal accuracy reduction for any group). Ustun et al. (2019) leverages individuals' sensitive attributes to train *decoupled* ML models, ensuring each group receives the best possible performance from its assigned model compared to a pooled model (trained on data from all groups) or the decoupled models of other groups. Yin et al. (2024) proposes a method using *abstention*, where a pre-trained ML model selectively defers certain predictions to human decision-makers, thus achieving group fairness without reducing accuracy.

### A.4 FAIRNESS BENCHMARKS

A number of toolkits have been developed to standardize fairness evaluation and mitigation. AI Fairness 360 (Bellamy et al., 2019) offers an extensible library of fairness metrics and bias mitigation algorithms across datasets and tasks. Similarly, Fairlearn (Bird et al., 2020) provides practical tools for assessing and improving fairness in machine learning pipelines, with an emphasis on industry adoption. Beyond toolkits, early benchmarking efforts such as Reddy et al. (2021) compared bias mitigation algorithms in representation learning, highlighting trade-offs across fairness metrics. While these efforts established important foundations, they were largely limited to classical ML or relatively simple datasets and did not extend to complex vision or multi-modal contexts.

Subsequent benchmarks have attempted to unify evaluation but remain limited in scope. MEDFAIR (Zong et al., 2023) targets fairness in medical datasets, addressing sensitive healthcare applications in vision tasks. FFB (Han et al., 2024) primarily evaluates older fairness algorithms before 2021 and omits recent advances in representation learning and data-centric strategies, while also lacking systematic hyperparameter tuning. ABCFair (Defrance et al., 2024) focuses on tabular datasets and adopts fixed hyperparameter settings, which restrict scalability and may misrepresent performance in more complex domains.

With the rise of large vision–language models (LVLMs), newer benchmarks have begun to address multimodal fairness. Xia et al. (2024); Jin et al. (2024a) investigate fairness in medical multimodal foundation models. GenderBias-VL Xiao et al. (2025) and VLBiasBench Wang et al. (2024) explore bias in LVLMs but typically cover smaller models, leaving out the larger LVLMs increasingly deployed in real-world applications. VLAs (Girrbach et al., 2025) is the most recent work, focusing specifically on gender bias in LVLMs and corresponding mitigation strategies.

In addition to benchmarking comparison on fairness, recent research has also emphasized the importance of tailored analyses to determine appropriate mitigation strategies based on the specific nature of bias. For instance, Yang et al. (2023) focuses on addressing subpopulation shifts. Jones et al. (2025) studied the fair representation learning method and found it not useful for performance-sensitive tasks. Furthermore, Schrouff et al. (2024) suggests considering the causal graph before performing data balancing for fairness. Roschewitz et al. (2025) shows that identifying dataset shifts is crucial for selecting the correct intervention. Matos et al. (2025) studies the landscape of fairness metrics, highlighting the fragmentation in current definitions and the necessity of selecting metrics that align with real-world utility.

In contrast, NH-Fair is designed as a general-purpose fairness-without-harm benchmark evaluating multiple demographic attributes (e.g., gender, race, age) across diverse models, datasets, and mitigation strategies. Its distinct contributions are: (1) unifying evaluation across classical vision and multimodal models; (2) systematically analyzing the role of training settings; and (3) extending coverage to state-of-the-art LVLMs at deployment-relevant scales, ranging from 4B to 109B parameters.

## B  EXPERIMENT SETUP

### B.1  DATASETS DETAILS

Unless otherwise noted, we randomly split each dataset into training, validation, and testing sets with a ratio of 8:1:1.

**CelebA (Liu et al., 2015)**: The CelebFaces Attributes Dataset, known as CelebA, is an extensive collection featuring over 200,000 images of celebrities, with each image annotated for 40 distinct attributes. In our research, we focus on the "wavy hair" attribute as the classification target while treating the "male" attribute as a sensitive feature.

**UTKFace (Zhang et al., 2017)**: The UTKFace dataset comprises more than 20,000 facial images, each labeled with information on age, gender, and ethnicity. For our analysis, we simplify the race attribute into two categories: "white" and "non-white as the sensitive attribute and take the "gender" as the classification target.

**FairFace (Karkkainen and Joo, 2021)**: The FairFace dataset contains over 100,000 facial images annotated with age, gender, and race, emphasizing balanced demographic representation. We employ "race" prediction as the seven-class classification problem, and use "gender" as the sensitive attribute.

**Facet (Gustafson et al., 2023)**: The Facet dataset includes facial images annotated with Fitzpatrick skin types, age, and gender. We utilize the binary attribute "visible face" as the classification target. Since a single data entry in Facet may have multiple skin type annotations, we use "gender" as our attribute for simplicity. Entries with incomplete gender annotations were removed.

**HAM10000 (Maron et al., 2019)**: The HAM10000 dataset is a large collection of dermatoscopic images used for skin lesion classification. We reclassified its 7 diagnostic categories into two broad labels: "benign" and "malignant" following (Maron et al., 2019). Images with missing recorded sensitive attributes were excluded from the dataset. The HAM10000 dataset includes two sensitive attributes: age and sex. We binarized the age attribute into two categories: "young" and "old." Individuals aged 0-60 years were classified as "young," while those aged 60 years and above were classified as "old."

**Fitzpatrick17k (Groh et al., 2021)**: The Fitzpatrick17k dataset (Fitz17k) comprises dermatological images labeled with Fitzpatrick skin types and diagnostic categories. We reclassify diagnoses into "benign" and "malignant" groups, following (Groh et al., 2021), and use skin type as the sensitive attribute, binarized into "lighter" (I-III) and "darker" (IV-VI). Images missing skin type or diagnostic labels were excluded.

**Waterbirds (Sagawa* et al., 2020)**: The Waterbirds dataset contains images of waterbirds and landbirds superimposed on either water or land backgrounds. We classify bird type (waterbird vs. landbird) as the target, with background (water vs. land) serving as the sensitive attribute. Instances with ambiguous background or species labels were removed. We use the train/validation/test split provided with the dataset.

Table 5: Download URLs for the datasets.

| Dataset | Download URL | License |
|---|---|---|
| CelebA (Liu et al., 2015) | `http://mmlab.ie.cuhk.edu.hk/projects/CelebA.html` | Non-commercial research only |
| UTKFace (Zhang et al., 2017) | `https://susanqq.github.io/UTKFace/` | Non-commercial research only |
| FairFace (Karkkainen and Joo, 2021) | `https://github.com/joojs/fairface` | CC BY 4.0 |
| Facet (Gustafson et al., 2023) | `https://ai.meta.com/datasets/facet-downloads/` | Meta Images Research |
| HAM10000 (Maron et al., 2019) | `https://dataverse.harvard.edu/dataset.xhtml?persistentId=doi:10.7910/DVN/DBW86T` | CC BY-NC 4.0 |
| Fitzpatrick17k (Groh et al., 2021) | `https://github.com/mattgroh/fitzpatrick17k` | CC BY-NC-SA 4.0 |
| Waterbirds (Sagawa* et al., 2020) | `https://github.com/kohpangwei/group_DRO` | No license specified |

## B.2 FAIRNESS METRICS

- **Overall Accuracy Parity (Gap):** The accuracy/AUC gap between two sensitive groups.

- **Max-Min Fairness (Worst):** The worst accuracy/AUC across two sensitive groups.

- **Demographic Parity (DP):** For binary classification, we focus only on the positive class. For multi-class classifications such as FairFace, we follow the fairness guarantees outlined in (Denis et al., 2024):

$$1 - \max_{y \in [\mathcal{Y}]} \left| \mathbb{P}\big[h(X) = y | A = a\big] - \mathbb{P}\big[h(X) = y | A = a'\big] \right|.$$

- **Equalized Odds (EqOdd):** It is evaluated by ensuring parity in the probability of correct classification:

$$1 - \left| \mathbb{P}\big[h(X) = y \mid Y = y, A = a\big] - \mathbb{P}\big[h(X) = y \mid Y = y, A = a'\big] \right|.$$

We calculate this for all classes and take the average across classes as the final outcome.

## B.3 METHODS

- **Data-centric methods**

  - **RandAugment (Cubuk et al., 2020)**: RandAugment (denoted as RandAug in the experiment results) is commonly used in semi-supervised and unsupervised learning. It randomly selects and applies a set of data augmentations—such as rotations, translations, or brightness adjustments—to diversify the training data. In this study, we aim to evaluate whether training a model on more diverse data, without using demographic information, can lead to improved fairness.

  - **Mixup (Zhang et al., 2018)**: It is a data augmentation technique that combines two training samples and their corresponding labels via linear interpolation to create new synthetic examples. This encourages the model to learn smoother decision boundaries and reduces overconfidence on specific group-correlated features, forcing the model to generalize beyond rigid group distinctions. We apply Mixup by blending data from different groups to evaluate its impact on fairness and performance.

  - **Mixup (Zhang et al., 2018)**: This data augmentation technique combines two training samples and their corresponding labels via linear interpolation to create new synthetic examples. We apply Mixup by blending data from different demographic groups; this encourages the model to

learn smoother decision boundaries and reduces overconfidence on specific group-correlated features, effectively forcing the model to generalize beyond rigid group distinctions.

– **Resampling (Buda et al., 2018; Sagawa* et al., 2020)**: It balances the dataset distribution by either over-sampling underrepresented groups or under-sampling overrepresented groups. This is done by assigning weights to each sample, helping to address class or group imbalances.

– **Bias Mimicking (Qraitem et al., 2023)**: It introduces a class-conditioned sampling technique that breaks the correlation between labels and known attributes. It constructs a subsampled distribution to mimic a biased distribution for all classes during training, thereby statistically minimizing the correlation between sensitive attributes and targets.

– **FIS (Pang et al., 2024)**: Fair Influential Sampling (FIS) mitigates group disparities without using sensitive attributes during training. It assumes group labels are available only for the validation set and scores each candidate by its estimated influence on both accuracy and a fairness objective (e.g., risk disparity) evaluated on that validation set. The algorithm then actively adds the highest-influence examples to the training data, inducing a targeted distribution shift that reduces disparity without harm.

- **Algorithmic methods**

  – **Decoupled Classifier (Ustun et al., 2019; Wang et al., 2020)**: It trains separate classifiers for each sensitive group, allowing group-tailored decision boundaries and then aggregates their predictions.

  – **LAFTR (Madras et al., 2018)**: Learning Adversarially Fair and Transferable Representations (LAFTR) trains a feature extractor (representation function) so that the learned representation is predictive of the target while being uninformative about the sensitive attribute. Concretely, a predictor is optimized to minimize task loss while an adversary is trained to predict sensitive attribute to enforce conditional independence. Since the constraint is imposed at the representation level, the learned feature extractor supports downstream fair prediction.

  – **FSCL (Park et al., 2022)**: Fair Supervised Contrastive Loss (FSCL) integrates fairness considerations into a contrastive loss function. It leverages contrastive learning to separate representations from different classes and align representations of the same class from different groups to achieve fair classification.

  – **GapReg (Chuang and Mroueh, 2021):** Gap Regularization embeds a chosen group-fairness metric, such as Demographic Parity, Equal Opportunity, and Equalized Odds, directly into the training loss function to balance both accuracy and fairness.

  – **MCDP (Jin et al., 2024b):** It introduces a fairness metric that captures the maximal local disparity by evaluating cumulative ratio disparities across varying neighborhoods. MCDP uses a regularization approach similar to GapReg by adding the regularization term to the task loss function during training to optimize the model.

  – **GroupDRO (Sagawa* et al., 2020)**: GroupDRO focuses on protecting the worst-case groups. It maintains per-group weights that are increased for groups with higher loss and forces the model to improve the worst-off subgroup rather than overfitting to the majority, thereby improving fairness for minorities.

  – **DFR (Kirichenko et al., 2023)**: Deep Feature Reweighting (DFR) is a post-processing method. It freezes the feature extractor of a standard ERM model and retrains only the final linear layer (the classifier) on a small reweighted set where the spurious correlation is broken. Following the original setup, we use a group-balanced reweighting set (i.e., a balanced distribution across sensitive groups) so last-layer retraining emphasizes core features over spurious cues. DFR provides a lightweight and efficient approach for achieving better predictions.

  – **Oxonfair (Delaney et al., 2024)**: Oxonfair is an open source toolkit for enforcing fairness in binary classification through post-processing methods. It supports multiple fairness notions and is able to minimize degradation while enforcing fairness. Since it only supports binary classification, we omit the FairFace dataset (the multi-class classification task) for this method. We follow their official implementation that enforces fairness on validation data and test on the test data. We consider five different combinations of optimization target and fairness constraints based on our benchmark tasks and metrics. These combinations can be found in Table 6. For datasets with AUC metrics, we change the accuracy optimization objective to balanced accuracy.

- **Multi-modal models**

- **CLIP (Radford et al., 2021)**: CLIP (Contrastive Language–Image Pretraining) learns visual concepts from natural language supervision by jointly training on image–text pairs. In this paper, we use the ViT (Dosovitskiy et al., 2021) as the backbone. In addition, we consider two CLIP-based post-training debias methods: FairerCLIP (Dehdashtian et al., 2024) and SFID (Jung et al., 2024). FairerCLIP mitigates bias by projecting image and text representations into a Reproducing Kernel Hilbert Space (RKHS) and minimizing the Hilbert-Schmidt Independence Criterion (HSIC), thereby statistically enforcing independence between the embeddings and sensitive attributes. SFID (Unified Selective Feature Imputation for Debiasing) identifies features reliant on spurious correlations and applies a selective imputation strategy to reconstruct representations that are invariant to demographic shifts.

- **BLIP2 (Li et al., 2023)**: BLIP-2 bridges frozen vision encoders and frozen LLMs using a lightweight *Q-Former* trained in two stages to extract language-aligned visual tokens. We use BLIP2 to generate the image embeddings and text embeddings and calculate the embedding similarities to make classifications.

- **LLaVA-1.6 (Liu et al., 2024)**: LLaVA (Large Language and Vision Assistant) is an auto-regressive language model that aligns vision features with a language model using instruction tuning.

- **Qwen2.5-VL (Bai et al., 2025)**: Qwen2.5-VL is the latest model of Qwen vision-language series from Alibaba. It integrates a vision encoder and a language model decoder to process multimodal inputs and achieve comparable performance with GPT-4o and Claude 3.5 Sonnet.

- **Gemma 3 (Team et al., 2025)**: Gemma is a family of lightweight open models released by Google, ranging from 1B to 27B parameters. As the 1B model supports only text inputs, we evaluate the multimodal capabilities of the 4B, 12B, and 27B variants in this paper.

- **Llama 3.2 (Grattafiori et al., 2024) and Llama 4 (Meta, 2025)**: Llama 4 is the latest multimodal model released by Meta. Due to the computation resource limitation, we only evaluate Llama 4 Scout, a smaller variant compared with Llama 4 Maverick, alongside the Llama-3.2 multimodal series.

We present all used models and their sources in Table 6. All data-centric and algorithmic methods except RandAugment use sensitive attributes during training, while the decoupled classifier requires access to the sensitive attribute at deployment.

### B.4 IMPLEMENTATION DETAILS

The experiments were conducted on a supercomputer cluster, where each node is equipped with two AMD EPYC 7643 processors, four NVIDIA A100 GPUs (80GB memory each), and 921GB of RAM. The code is implemented with Python 3.10.12 and PyTorch 2.5.0. All images are resized to $224 \times 224$ pixels, and during training, we apply random horizontal flipping for data augmentation. We use a pre-trained ResNet-18 as the backbone initialization to start the training. In addition, we conduct experiments on pretrained weights and model size in Appendix C.

For RandAug, color-based augmentations were excluded during training, as the sensitive attributes in our study include skin type, and such augmentations could inadvertently alter needed features.

The Fairness Influence Selection (FIS) method was originally designed for an active learning setting, where a portion of the dataset remains unlabeled. In its original formulation, FIS selects necessary unlabeled data using a combination of pseudo-labels and ground-truth labels. To adapt it to our setting, we employ ground-truth labels for influence-guided selection. Specifically, the training set is randomly partitioned into two subsets according to a predefined ratio: one subset is used for supervised training, while during validation, the model assesses the influence of data in the second subset and selectively incorporates a subset of these samples into the training set to achieve fairness without harm. The split ratio is treated as a hyperparameter, as it varies across different datasets.

For Fairness-Sensitive Contrastive Learning (FSCL), training was divided into two phases: The first 60 epochs perform contrastive learning. The subsequent 40 epochs were used for classifier training.

For GapReg and MCDP, the differential privacy (DP) loss function was extended to accommodate multi-class classification, particularly for the FairFace dataset. This extension is based on the multi-class demographic parity formulation proposed by (Denis et al., 2024).

Table 6: Summary of multi-modal models

|  | Model | #Param | URL |
|---|---|---|---|
| CLIP | CLIP-ViT-B/16 | 150M | `https://github.com/openai/CLIP` |
| BLIP-2 | BLIP-2 Base | 1B | `https://github.com/salesforce/LAVIS` |
| LLaVA | LLaVA-v1.6-vicuna-7b-hf | 7B | `https://huggingface.co/LLaVA-hf/LLaVA-v1.6-vicuna-7b-hf` |
|  | LLaVA-v1.6-vicuna-13b-hf | 13B | `https://huggingface.co/LLaVA-hf/LLaVA-v1.6-vicuna-13b-hf` |
|  | LLaVA-v1.6-34b-hf | 34B | `https://huggingface.co/LLaVA-hf/LLaVA-v1.6-34b-hf` |
| Qwen | Qwen2.5-VL-7B-Instruct | 7B | `https://huggingface.co/Qwen/Qwen2.5-VL-7B-Instruct` |
|  | Qwen2.5-VL-32B-Instruct | 32B | `https://huggingface.co/Qwen/Qwen2.5-VL-32B-Instruct` |
|  | Qwen2.5-VL-72B-Instruct | 72B | `https://huggingface.co/Qwen/Qwen2.5-VL-72B-Instruct` |
| Gemma | gemma-3-4b-it | 4B | `https://huggingface.co/google/gemma-3-4b-it` |
|  | gemma-3-12b-it | 12B | `https://huggingface.co/google/gemma-3-12b-it` |
|  | gemma-3-27b-it | 27B | `https://huggingface.co/google/gemma-3-27b-it` |
| Llama | Llama-3.2-11B-Vision-Instruct | 11B | `https://huggingface.co/meta-Llama/Llama-3.2-11B-Vision-Instruct` |
|  | Llama-3.2-90B-Vision-Instruct | 90B | `https://huggingface.co/meta-Llama/Llama-3.2-90B-Vision-Instruct` |
|  | Llama 4 Scout (17Bx16E) | 109B | `https://huggingface.co/meta-Llama/Llama%204-Scout-17B-16E-Instruct` |

For CLIP and BLIP2, we use their public implementations. For LLVMs, we adopt the open-source weights from Huggingface for all models with BF16 or FP16 precision, depending on their suggested model loading.

## B.5 HYPERPARAMETERS

We perform extensive hyperparameter optimization using Bayesian hyperparameter search via *Weights & Biases* on the validation results. The default batch size is set to 256 for all experiments. For the SGD optimizer, we deploy a StepLR scheduler and set momentum to 0.9, where the learning rate is reduced by a factor of 0.1 every 30 epochs. The number of hyperparameter search iterations varies with the complexity of each method–dataset combination, subject to a tuning budget of 100–200 searches per method–dataset pair. The search space includes both discrete choices and continuous ranges, as detailed in Table 7. $\mathcal{U}$ denotes a uniform distribution, and $\log \mathcal{U}$ denotes a log-uniform distribution. The benchmarking process required a total of approximately 1.1 GPU years. All methods were trained for 100 epochs with early stopping to prevent overfitting. Early stopping is used if the validation loss or validation accuracy/AUC does not improve after 10 epochs.

## B.6 PROMPTS

Table 8 lists the zero-shot prompts used throughout our study. For image–text matching models (CLIP, BLIP-2, FairerCLIP, SFID), we use one template sentence per class and compute the image–caption similarity, then pick the label of the highest-scoring caption as the prediction. For large

Table 7: Hyperparameter search space for all methods.

| Category | Hyperparameter | Search Space |
|---|---|---|
| Common Hyperparameters | Learning Rate
Weight Decay
Optimizer | $\log \mathcal{U}(10^{-5}, 10^{-2})$
$\{0, 10^{-5}, 10^{-4}, 10^{-3}, 10^{-2}\}$
$\{$Adam, SGD$\}$ |
| Mixup | Mixup loss coefficient | $\mathcal{U}(0.1, 10)$ |
| Resampling | Resampling method | $\{$Group, Group $\times$ Class$\}$ |
| BM | Sampling mode | $\{$none, us, uw, os$\}$ |
| FIS | Label ratio
Fairness metric | $\{0.1, 0.3, 0.5\}$
$\{$dp, eop, eod$\}$ |
| LAFTR | Class coefficient
Fairness coefficient | $\mathcal{U}(0.1, 1)$
$\mathcal{U}(0.1, 1)$ |
| FSCL | Group normalization | $\{0, 1\}$ |
| GapReg | Fairness regularization objective
Regularization coefficient | $\{$dp, eop, eod$\}$
$\mathcal{U}(0.01, 5)$ |
| MCDP | Regularization coefficient
Temperature | $\mathcal{U}(0.01, 5)$
$\{5, 10, 20, 50, 100\}$ |
| GroupDRO | Alpha
Gamma | $\mathcal{U}(0.01, 5)$
$\mathcal{U}(0.01, 5)$ |
| DFR | Tune class weights
Add train data
Regularization
C options
Class weight options | $\{0, 1\}$
$\{0, 1\}$
$\{$L1, L2$\}$
$\{3, 1, 0.7, 0.3, 0.1, 0.07, 0.03, 0.01, 0.003\}$
$\{1, 2, 3, 10, 100, 300, 1000\}$ |
| Oxonfair | Objective and Constraint | $\{$accuracy, accuracy.diff$\}$ # optimize accuracy without reducing gap
$\{$accuracy.min, accuracy$\}$ # optimize worst without reducing overall
$\{$accuracy.diff, accuracy.min$\}$ # optimize gap without reducing worst
$\{$equalized_odds, accuracy$\}$ # optimize EqOdd without reducing overall
$\{$demographic_parity, accuracy$\}$ # optimize DP without reducing overall |
| FairerCLIP | $\tau_i$
$\tau_{z,i}$
$\tau_t$
$\tau_{z,t}$
RFF Dim | $\mathcal{U}(0, 1)$
$\mathcal{U}(0, 1)$
$\mathcal{U}(0, 1)$
$\mathcal{U}(0, 1)$
$\{1000, 2000, 3000, 4000, 8000\}$ |
| SFID | Confidence threshold
Number of Pruned Feature | $\mathcal{U}(0.5, 1)$
$\{50, 75, 100\}$ |

vision–language models, we feed the prompt with image input to the model and then parse the model's short text answer.

Table 8: Prompts used across datasets for text-image matching and visual-language model (image-text-to-text) inference.

| Dataset | Prompts |
|---|---|
| CelebA | **Image-Text Matching:**
*[ "A photo of a person with non-wavy hair.",*
*"A photo of a person with wavy hair."]*
**Image-Text-to-Text:**
Is the person in the photo wavy-haired?
Only Answer 'Yes' for wavy hair, 'No' for non-wavy hair. |
| UTKFace | **Image-Text Matching:**
*[ "A photo of a male.",*
*"A photo of a female."]*
**Image-Text-to-Text:**
Is the person in the photo male or female?
Only Answer 'Male' or 'Female'. |
| FairFace | **Image-Text Matching:**
*[ "A photo of a White person.",*
*"A photo of a Black person.",*
*"A photo of a Latino or Hispanic person.",*
*"A photo of an East Asian person.",*
*"A photo of a Southeast Asian person.",*
*"A photo of an Indian person.",*
*"A photo of a Middle Eastern person."]*
**Image-Text-to-Text:**
Based on the photo, classify the race of the person into:
0: White, 1: Black, 2: Latino or Hispanic, 3: East Asian,
4: Southeast Asian, 5: Indian, 6: Middle Eastern.
Please answer with the corresponding category number only. |
| Facet | **Image-Text Matching:**
*"A photo of a person with non-visible face."*
*"A photo of a person with visible face."*
**Image-Text-to-Text:**
Does the photo show a visible face?
Only answer 'Yes' if the face is visible, otherwise answer 'No'. |
| Waterbirds | **Image-Text Matching:**
*"A photo of a landbird."*
*"A photo of a waterbird."*
**Image-Text-to-Text:**
Is the bird in the photo a landbird or a waterbird?
Only answer 'Landbird' or 'Waterbird'. |
| HAM10000 / Fitz17k | **Image-Text Matching:**
*"A photo of a benign skin condition."*
*"A photo of a malignant skin condition."* |

# C  ADDITIONAL EXPERIMENTS

## C.1  TRAINING CHOICE EXPLORATIONS

In this paper, we argue that a carefully tuned ERM often rivals specialized debiasing methods. We then organize the further studies to mirror the life cycle of a standard vision pipeline. We start with initialization choices from pre-trained weights versus training from scratch, because the features a model begins with strongly constrain downstream bias. Next, we study optimizer choice to challenge prior work that often fixes optimizers across different settings and evaluate their influence on both utility and fairness. We then turn to other common training choices like batch size, weight decay, and model/checkpoint selection. Together, these investigations shed light on fairness outcomes and also help reduce the search space for future hyperparameter optimization, providing principled guidance on which training choices matter most.

### C.1.1  IMPACT OF PRETRAINED WEIGHTS

**Loading pretrained weights improves overall utility without harming fairness**. In this part, we investigate the impact of loading pretrained weights on ImageNet (Deng et al., 2009) before training on fairness and utility on ERM. For clarity, Figure 6 extracts the most representative panels from the full grid in Figure 7. Each panel plots models trained with pretraining (red circles, "Yes") and from scratch (blue circles, "No") across our seven datasets while sweeping the same hyperparameter grid, making the influence of weight initialization easy to see at a glance. The horizontal axis of each plot measures subgroup performance disparity (e.g., "Gap AUC" or "Gap ACC"), while the vertical axis measures overall utility (e.g., "Overall AUC" or "Overall ACC"). In all cases, points closer to the top-left corner indicate both higher overall performance and smaller subgroup gaps, which represent more favorable fairness-utility trade-offs. A general trend emerges showing that models initialized with pretrained weights tend to achieve higher overall performance than those trained from scratch. This benefit is especially apparent in datasets with relatively small training sets, such as Fitz17k and Waterbirds. However, pretrained models do not consistently outperform models trained from scratch in closing the subgroup performance gap. This observation is further supported by other fairness metrics (such as DP, EqOdd, and Worst) reported in the Figure 7, where the full comparison is provided. Despite this mixed impact on fairness, pretrained models improve overall utility without harming any particular subgroup, particularly benefiting disadvantaged groups (as seen in the Worst metric in the appendix). This aligns with our "fairness without harm" principle. As a result, we adopt pretrained models in our benchmark for their superior overall performance while maintaining acceptable fairness levels.

### C.1.2  IMPACT OF OPTIMIZER CHOICE ON FAIRNESS AND UTILITY

**Using a fixed optimizer across methods and datasets can lead to unfair comparisons.** Many prior studies use a single optimizer across datasets and methods, sometimes even without fine-tuning learning rates. Based on this, we conducted this study on two baseline methods and two datasets to provide empirical evidence that using a fixed optimizer across methods and datasets can lead to unfair comparisons. Our findings in Figure 8 indicate that different datasets respond differently to various optimizers, and selecting the right one can improve both utility and fairness. For instance, models trained on CelebA with SGD tend to achieve better utility and smaller accuracy gaps, resulting in a more balanced model compared to those trained with Adam. This is evident in the results, where SGD-based models cluster around higher accuracy values with lower fairness gaps. On the other hand, in the Fitz17k dataset, models trained with Adam perform better, achieving higher utility while maintaining competitive fairness scores. This variation in optimizer performance highlights that a one-size-fits-all approach can lead to suboptimal results, especially in fairness-sensitive settings. **Our findings underscore the need for more equitable and transparent evaluation practices in future fairness research.**

**Further Analysis (ERM Focus)**. In the main text, due to computational constraints, we primarily experimented with SGD (Bottou, 2012) and Adam (Kingma and Ba, 2015). Here, we extend the study to include two additional optimizers, AdamW (Loshchilov and Hutter, 2019) and Adagrad (Ward et al., 2020), in order to better understand their impact on ERM performance. Across datasets, optimizer choice exhibits a more pronounced and consistent influence on both utility and fairness than model size or batch size. For example, on Waterbirds, AdamW and Adagrad achieve both higher

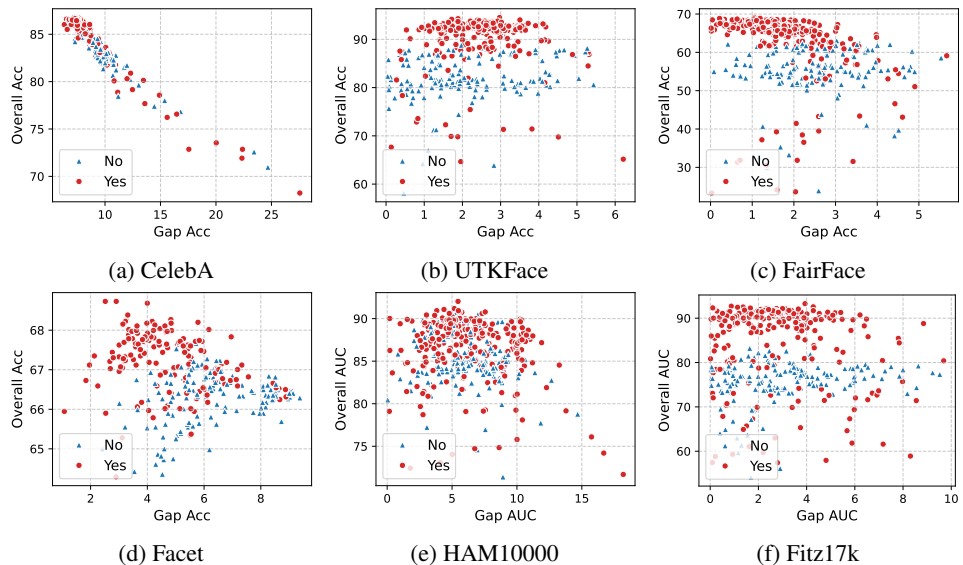

Figure 6: Comparison of utility gap for the pretrained weights. More left indicates greater fairness (smaller accuracy gap), while higher values indicate better performance.

Table 9: Performance comparison across different optimizers.

| Dataset | Optimizer | Utility (Acc or AUC) | Worst | Gap | EqOdd | DP |
|---|---|---|---|---|---|---|
| **CelebA** | SGD | $86.57 \pm 0.18$ | $\mathbf{83.76 \pm 0.23}$ | $\mathbf{6.76 \pm 0.34}$ | $\mathbf{81.91 \pm 0.58}$ | $67.20 \pm 0.69$ |
| | Adam | $86.11 \pm 0.43$ | $82.85 \pm 0.61$ | $7.89 \pm 0.29$ | $80.46 \pm 3.23$ | $67.22 \pm 2.81$ |
| | AdamW | $86.48 \pm 0.14$ | $83.30 \pm 0.19$ | $7.67 \pm 0.12$ | $79.27 \pm 2.36$ | $66.03 \pm 1.88$ |
| | Adagrad | $\mathbf{86.65 \pm 0.14}$ | $83.62 \pm 0.25$ | $7.33 \pm 0.27$ | $81.39 \pm 0.88$ | $\mathbf{67.90 \pm 0.32}$ |
| | **Std** | 0.21 | 0.35 | 0.43 | 1.00 | 0.67 |
| **UTKFace** | SGD | $91.95 \pm 0.03$ | $90.51 \pm 0.22$ | $3.30 \pm 0.55$ | $96.76 \pm 0.58$ | $95.56 \pm 0.34$ |
| | Adam | $92.75 \pm 0.54$ | $\mathbf{91.78 \pm 0.61}$ | $\mathbf{2.26 \pm 0.64}$ | $\mathbf{97.62 \pm 0.53}$ | $94.55 \pm 1.20$ |
| | AdamW | $\mathbf{92.85 \pm 0.28}$ | $91.67 \pm 0.37$ | $2.72 \pm 0.71$ | $97.30 \pm 0.70$ | $\mathbf{95.70 \pm 0.21}$ |
| | Adagrad | $92.84 \pm 0.22$ | $91.27 \pm 0.28$ | $3.59 \pm 0.16$ | $96.45 \pm 0.15$ | $95.17 \pm 0.41$ |
| | **Std** | 0.38 | 0.50 | 0.51 | 0.46 | 0.45 |
| **FairFace** | SGD | $65.68 \pm 0.39$ | $64.64 \pm 0.54$ | $1.86 \pm 0.31$ | $95.98 \pm 0.42$ | $97.36 \pm 0.21$ |
| | Adam | $66.76 \pm 0.21$ | $66.34 \pm 0.37$ | $0.87 \pm 0.51$ | $96.22 \pm 0.57$ | $97.61 \pm 0.41$ |
| | AdamW | $67.09 \pm 0.39$ | $66.79 \pm 0.42$ | $\mathbf{0.64 \pm 0.06}$ | $\mathbf{96.50 \pm 0.38}$ | $\mathbf{97.69 \pm 0.17}$ |
| | Adagrad | $\mathbf{67.54 \pm 0.43}$ | $\mathbf{66.83 \pm 0.73}$ | $1.50 \pm 0.64$ | $96.07 \pm 0.41$ | $97.65 \pm 0.17$ |
| | **Std** | 0.69 | 0.89 | 0.49 | 0.20 | 0.13 |
| **Facet** | SGD | $67.55 \pm 0.37$ | $\mathbf{64.25 \pm 0.97}$ | $4.31 \pm 1.13$ | $96.47 \pm 1.23$ | $95.40 \pm 1.12$ |
| | Adam | $66.34 \pm 0.89$ | $63.09 \pm 1.33$ | $\mathbf{4.23 \pm 0.87}$ | $94.60 \pm 1.40$ | $94.17 \pm 1.23$ |
| | AdamW | $67.46 \pm 0.17$ | $62.23 \pm 0.74$ | $6.80 \pm 1.14$ | $95.67 \pm 1.22$ | $94.62 \pm 0.88$ |
| | Adagrad | $\mathbf{67.62 \pm 0.13}$ | $62.99 \pm 0.23$ | $6.02 \pm 0.31$ | $\mathbf{97.15 \pm 0.91}$ | $\mathbf{95.95 \pm 0.79}$ |
| | **Std** | 0.52 | 0.72 | 1.11 | 0.95 | 0.69 |
| **HAM10000** | SGD | $\mathbf{88.35 \pm 1.83}$ | $\mathbf{84.68 \pm 2.02}$ | $4.11 \pm 2.08$ | $88.17 \pm 3.10$ | $82.22 \pm 4.78$ |
| | Adam | $86.47 \pm 1.13$ | $81.74 \pm 1.72$ | $4.53 \pm 1.66$ | $\mathbf{92.55 \pm 2.43}$ | $\mathbf{87.69 \pm 2.69}$ |
| | AdamW | $88.18 \pm 0.66$ | $84.38 \pm 0.74$ | $\mathbf{3.40 \pm 1.13}$ | $89.02 \pm 6.00$ | $80.23 \pm 7.57$ |
| | Adagrad | $87.00 \pm 0.90$ | $81.98 \pm 2.86$ | $5.27 \pm 3.17$ | $91.33 \pm 3.61$ | $85.52 \pm 1.04$ |
| | **Std** | 0.79 | 1.34 | 0.68 | 1.75 | 2.88 |
| **Fitz17k** | SGD | $89.74 \pm 1.00$ | $88.39 \pm 1.05$ | $2.92 \pm 1.14$ | $94.92 \pm 2.48$ | $\mathbf{94.46 \pm 1.40}$ |
| | Adam | $89.62 \pm 0.53$ | $87.91 \pm 0.47$ | $2.34 \pm 0.41$ | $94.90 \pm 0.92$ | $91.74 \pm 0.82$ |
| | AdamW | $91.37 \pm 0.26$ | $\mathbf{90.45 \pm 0.32}$ | $\mathbf{2.11 \pm 0.91}$ | $93.91 \pm 1.59$ | $92.32 \pm 0.62$ |
| | Adagrad | $\mathbf{91.43 \pm 0.27}$ | $90.40 \pm 0.56$ | $3.51 \pm 1.66$ | $\mathbf{96.09 \pm 0.95}$ | $93.40 \pm 0.52$ |
| | **Std** | 0.86 | 1.15 | 0.54 | 0.77 | 1.04 |
| **Waterbirds** | SGD | $85.45 \pm 0.93$ | $83.72 \pm 1.68$ | $3.57 \pm 1.82$ | $68.23 \pm 1.97$ | $76.09 \pm 2.03$ |
| | Adam | $85.63 \pm 1.36$ | $84.20 \pm 0.94$ | $2.87 \pm 0.85$ | $66.53 \pm 3.31$ | $77.67 \pm 3.52$ |
| | AdamW | $87.09 \pm 0.54$ | $86.25 \pm 0.26$ | $1.68 \pm 0.90$ | $71.90 \pm 0.44$ | $\mathbf{81.54 \pm 0.80}$ |
| | Adagrad | $\mathbf{87.77 \pm 0.64}$ | $\mathbf{87.10 \pm 0.86}$ | $\mathbf{1.35 \pm 1.54}$ | $\mathbf{74.00 \pm 3.04}$ | $81.35 \pm 2.02$ |
| | **Std** | 0.98 | 1.40 | 0.90 | 2.94 | 2.35 |

overall accuracy and smaller subgroup gaps compared to SGD and Adam, indicating a more favorable fairness–utility trade-off. These results suggest that the relative advantages of each optimizer are

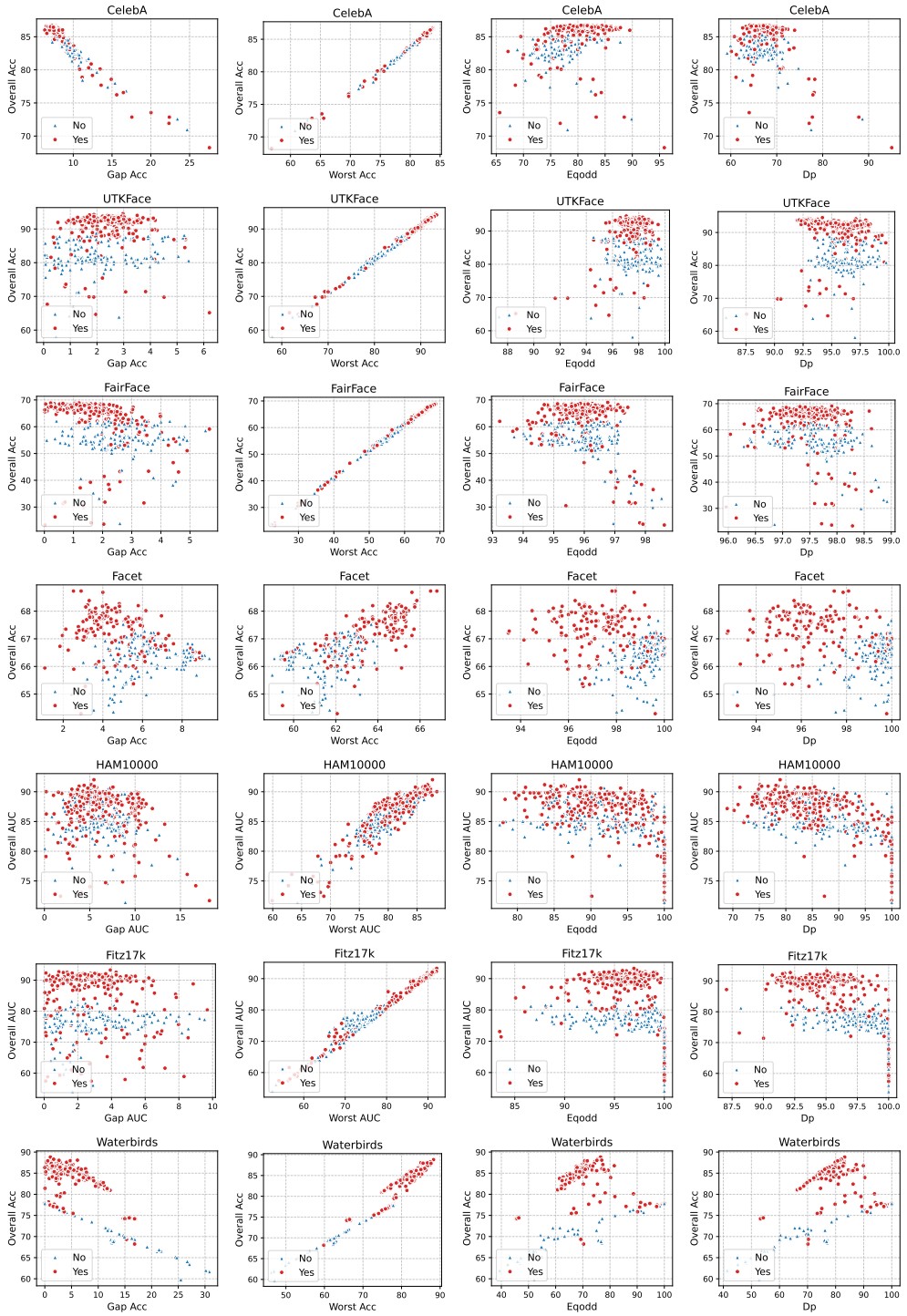

Figure 7: Full comparison of fairness metrics (Gap, Worst, EqOdd, DP) across all datasets for the pretrained weights.

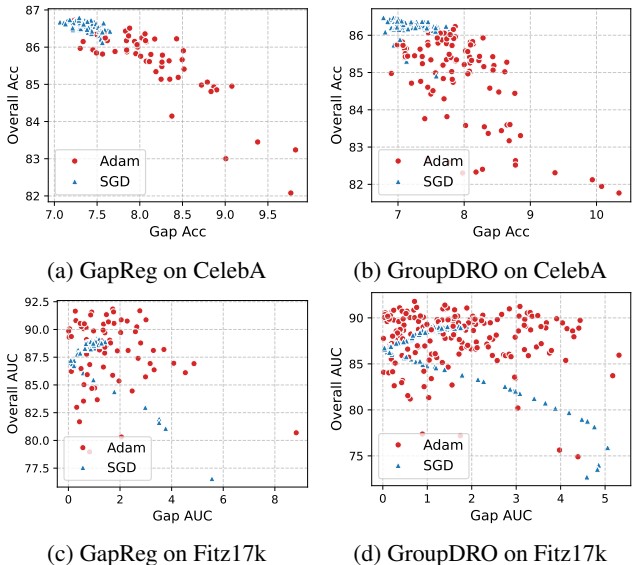

(a) GapReg on CelebA  (b) GroupDRO on CelebA

(c) GapReg on Fitz17k  (d) GroupDRO on Fitz17k

Figure 8: The effect of optimizer choice on fairness and utility trade-offs across CelebA and Fitz17k. The points from the same color are models trained with different learning rates and weight decays.

highly dataset-dependent, yet their effects on utility and fairness are clearer than those of other hyperparameters. **Thus, when conducting hyperparameter optimizations, the choice of optimizer could be prioritized for fairness-sensitive evaluations.**

### C.1.3 IMPACT OF BATCH SIZE

**Batch size has a limited effect on overall utility but can affect fairness slightly.** In this benchmark, we use a batch size of 256 for all methods. We also evaluate if batch size have an influence on model fairness. We conduct experiments on seven datasets with different batch sizes ranging from 32 to 1024. The full results are reported in Table 10, and visualized in Figure 9. Across all datasets, utility remains relatively stable as batch size increases. However, worst-case performance and the accuracy gap fluctuate more significantly. In some cases, such as CelebA and HAM10000, worst-case performance tends to decrease slightly with larger batch sizes. This indicates that larger batch sizes might lead to increased disparities between advantaged and disadvantaged groups. Equalized Odds and Demographic Parity vary highly depending on specific datasets. For datasets like CelebA and Waterbirds, EqOdd tends to decrease as batch size increases, while EqOdd and DP increase with larger batch sizes in HAM10000 and Fitz17k. The results show that batch size can influence fairness, but not in a consistent way. **However, since its influence on fairness is relatively minor and inconsistent, batch size can be given lower priority in future hyperparameter searches.**

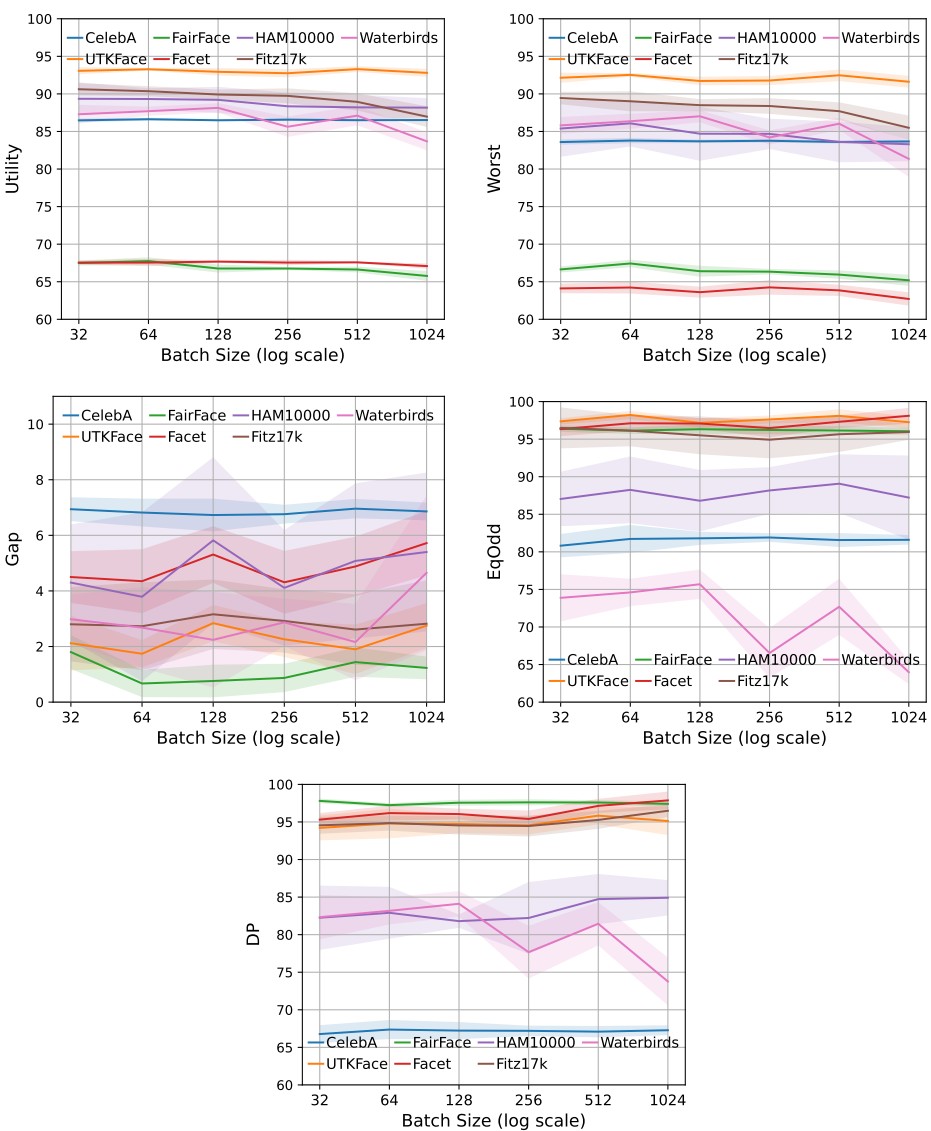

Figure 9: Effect of batch size on all metrics across seven datasets derived from Table 10. Each subplot corresponds to one evaluation metric, and curves represent different datasets. The x-axis is plotted on a $\log_2$ scale for clarity while keeping tick labels in the original batch-size units.

Table 10: Performance comparison across different batch sizes.

| Dataset | Batch Size | Utility (Acc or AUC) | Worst | Gap | EqOdd | DP |
|---|---|---|---|---|---|---|
| **CelebA** | 32 | $86.48 \pm 0.25$ | $83.59 \pm 0.35$ | $6.94 \pm 0.43$ | $80.82 \pm 1.56$ | $66.77 \pm 1.19$ |
| | 64 | $\mathbf{86.63 \pm 0.20}$ | $\mathbf{83.79 \pm 0.35}$ | $6.82 \pm 0.50$ | $81.72 \pm 1.86$ | $\mathbf{67.37 \pm 1.27}$ |
| | 128 | $86.49 \pm 0.12$ | $83.69 \pm 0.23$ | $\mathbf{6.73 \pm 0.59}$ | $81.80 \pm 0.87$ | $67.23 \pm 1.14$ |
| | 256 | $86.57 \pm 0.18$ | $83.76 \pm 0.23$ | $6.76 \pm 0.34$ | $\mathbf{81.91 \pm 0.58}$ | $67.20 \pm 0.69$ |
| | 512 | $86.50 \pm 0.12$ | $83.60 \pm 0.21$ | $6.96 \pm 0.35$ | $81.57 \pm 0.93$ | $67.09 \pm 0.76$ |
| | 1024 | $86.52 \pm 0.09$ | $83.67 \pm 0.14$ | $6.86 \pm 0.32$ | $81.60 \pm 0.64$ | $67.28 \pm 0.66$ |
| | **Std** | 0.05 | 0.07 | 0.09 | 0.35 | 0.19 |
| **UTKFace** | 32 | $93.07 \pm 0.47$ | $92.15 \pm 0.68$ | $2.12 \pm 0.98$ | $97.37 \pm 0.42$ | $94.21 \pm 1.66$ |
| | 64 | $93.28 \pm 0.18$ | $\mathbf{92.53 \pm 0.25}$ | $\mathbf{1.74 \pm 0.49}$ | $\mathbf{98.22 \pm 0.55}$ | $94.80 \pm 1.98$ |
| | 128 | $92.94 \pm 0.50$ | $91.72 \pm 0.56$ | $2.84 \pm 0.65$ | $97.16 \pm 0.63$ | $94.73 \pm 1.23$ |
| | 256 | $92.75 \pm 0.54$ | $91.78 \pm 0.61$ | $2.26 \pm 0.64$ | $97.62 \pm 0.53$ | $94.55 \pm 1.20$ |
| | 512 | $\mathbf{93.30 \pm 0.39}$ | $92.48 \pm 0.78$ | $1.90 \pm 0.89$ | $98.10 \pm 0.88$ | $\mathbf{95.84 \pm 1.17}$ |
| | 1024 | $92.80 \pm 0.54$ | $91.62 \pm 0.79$ | $2.75 \pm 0.81$ | $97.25 \pm 0.82$ | $95.13 \pm 1.88$ |
| | **Std** | 0.21 | 0.36 | 0.41 | 0.41 | 0.51 |
| **FairFace** | 32 | $67.49 \pm 0.33$ | $66.64 \pm 0.41$ | $1.80 \pm 0.61$ | $\mathbf{96.39 \pm 0.43}$ | $\mathbf{97.80 \pm 0.31}$ |
| | 64 | $\mathbf{67.75 \pm 0.50}$ | $\mathbf{67.43 \pm 0.50}$ | $\mathbf{0.67 \pm 0.49}$ | $96.13 \pm 0.44$ | $97.24 \pm 0.27$ |
| | 128 | $66.76 \pm 0.54$ | $66.40 \pm 0.73$ | $0.76 \pm 0.59$ | $96.32 \pm 0.66$ | $97.56 \pm 0.40$ |
| | 256 | $66.76 \pm 0.21$ | $66.34 \pm 0.37$ | $0.87 \pm 0.51$ | $96.22 \pm 0.57$ | $97.61 \pm 0.41$ |
| | 512 | $66.63 \pm 0.39$ | $65.95 \pm 0.56$ | $1.44 \pm 0.53$ | $96.15 \pm 0.58$ | $97.59 \pm 0.31$ |
| | 1024 | $65.77 \pm 0.65$ | $65.19 \pm 0.75$ | $1.23 \pm 0.41$ | $96.05 \pm 0.46$ | $97.41 \pm 0.34$ |
| | **Std** | 0.64 | 0.68 | 0.40 | 0.12 | 0.17 |
| **Facet** | 32 | $67.55 \pm 0.27$ | $64.11 \pm 0.64$ | $4.50 \pm 0.93$ | $96.33 \pm 0.93$ | $95.31 \pm 0.87$ |
| | 64 | $67.55 \pm 0.48$ | $64.23 \pm 0.80$ | $4.35 \pm 1.15$ | $97.11 \pm 1.05$ | $96.18 \pm 0.99$ |
| | 128 | $\mathbf{67.68 \pm 0.14}$ | $63.61 \pm 0.73$ | $5.31 \pm 1.02$ | $97.08 \pm 0.74$ | $96.05 \pm 0.72$ |
| | 256 | $67.55 \pm 0.37$ | $\mathbf{64.25 \pm 0.97}$ | $\mathbf{4.31 \pm 1.13}$ | $96.47 \pm 1.23$ | $95.40 \pm 1.12$ |
| | 512 | $67.58 \pm 0.11$ | $63.85 \pm 0.75$ | $4.88 \pm 1.07$ | $97.31 \pm 0.83$ | $97.14 \pm 0.94$ |
| | 1024 | $67.08 \pm 0.35$ | $62.71 \pm 0.89$ | $5.72 \pm 1.16$ | $\mathbf{98.11 \pm 1.06}$ | $\mathbf{97.87 \pm 1.18}$ |
| | **Std** | 0.19 | 0.53 | 0.52 | 0.58 | 0.92 |
| **HAM10000** | 32 | $\mathbf{89.35 \pm 2.18}$ | $85.39 \pm 3.76$ | $4.30 \pm 2.09$ | $87.04 \pm 3.63$ | $82.25 \pm 4.28$ |
| | 64 | $89.32 \pm 1.44$ | $\mathbf{86.08 \pm 3.08}$ | $\mathbf{3.79 \pm 3.04}$ | $88.25 \pm 4.44$ | $82.91 \pm 3.45$ |
| | 128 | $89.21 \pm 1.73$ | $84.70 \pm 3.60$ | $5.82 \pm 2.99$ | $86.80 \pm 4.09$ | $81.80 \pm 0.89$ |
| | 256 | $88.35 \pm 1.83$ | $84.68 \pm 2.02$ | $4.11 \pm 2.08$ | $88.17 \pm 3.10$ | $82.22 \pm 4.78$ |
| | 512 | $88.21 \pm 1.69$ | $83.61 \pm 2.71$ | $5.08 \pm 2.79$ | $\mathbf{89.07 \pm 3.90}$ | $84.73 \pm 3.34$ |
| | 1024 | $88.17 \pm 1.26$ | $83.30 \pm 2.27$ | $5.40 \pm 2.86$ | $87.22 \pm 5.60$ | $\mathbf{84.91 \pm 2.35}$ |
| | **Std** | 0.53 | 0.96 | 0.73 | 0.80 | 1.23 |
| **Fitz17k** | 32 | $\mathbf{90.62 \pm 0.82}$ | $\mathbf{89.45 \pm 0.83}$ | $2.80 \pm 1.34$ | $\mathbf{96.50 \pm 2.73}$ | $94.55 \pm 1.12$ |
| | 64 | $90.36 \pm 0.67$ | $89.02 \pm 1.34$ | $2.73 \pm 1.58$ | $96.13 \pm 2.08$ | $94.85 \pm 1.00$ |
| | 128 | $89.92 \pm 0.66$ | $88.50 \pm 0.80$ | $3.16 \pm 1.25$ | $95.51 \pm 2.53$ | $94.54 \pm 1.22$ |
| | 256 | $89.74 \pm 1.00$ | $88.39 \pm 1.05$ | $2.92 \pm 1.14$ | $94.92 \pm 2.48$ | $94.46 \pm 1.40$ |
| | 512 | $88.94 \pm 1.15$ | $87.70 \pm 1.18$ | $\mathbf{2.61 \pm 1.27}$ | $95.65 \pm 2.36$ | $95.27 \pm 1.20$ |
| | 1024 | $86.97 \pm 1.38$ | $85.49 \pm 1.63$ | $2.82 \pm 1.56$ | $95.95 \pm 1.05$ | $\mathbf{96.49 \pm 0.84}$ |
| | **Std** | 1.22 | 1.28 | 0.17 | 0.50 | 0.71 |
| **Waterbirds** | 32 | $87.29 \pm 1.25$ | $85.81 \pm 1.11$ | $2.98 \pm 0.89$ | $73.88 \pm 3.14$ | $82.33 \pm 2.95$ |
| | 64 | $87.70 \pm 0.61$ | $86.36 \pm 1.00$ | $2.68 \pm 1.45$ | $74.60 \pm 1.83$ | $83.17 \pm 1.81$ |
| | 128 | $\mathbf{88.14 \pm 0.62}$ | $\mathbf{87.02 \pm 0.80}$ | $2.24 \pm 1.72$ | $\mathbf{75.69 \pm 1.96}$ | $\mathbf{84.11 \pm 1.70}$ |
| | 256 | $85.63 \pm 1.36$ | $84.20 \pm 0.94$ | $2.87 \pm 0.85$ | $66.53 \pm 3.31$ | $77.67 \pm 3.52$ |
| | 512 | $87.11 \pm 1.30$ | $86.03 \pm 0.81$ | $\mathbf{2.16 \pm 1.38}$ | $72.69 \pm 3.75$ | $81.45 \pm 2.89$ |
| | 1024 | $83.68 \pm 1.19$ | $81.35 \pm 2.41$ | $4.65 \pm 2.78$ | $63.98 \pm 1.60$ | $73.77 \pm 3.19$ |
| | **Std** | 1.52 | 1.89 | 0.83 | 4.38 | 3.60 |

### C.1.4 IMPACT OF WEIGHT DECAY

Weight decay is a critical hyperparameter that affects both model generalization and utility. The results in Table 11 suggest that while weight decay influences fairness, its effects vary significantly across different datasets. L2 regularization tends to maintain or slightly improve fairness metrics, whereas L1 regularization can, in some cases, amplify disparities, particularly in datasets with imbalanced subgroup distributions. Since L2 regularization is typically the default and widely adopted choice in hyperparameter optimization pipelines, while L1 regularization is used less frequently, we additionally include a comparison between L2 regularization and no regularization ($\Delta_{L2-No}$) in the table. This expanded setup provides a clearer view of the marginal effects of L2 regularization on fairness. We observe that tuned L2 weight decay rates have only minor effects (less than 0.5% in most cases) on fairness metrics for most datasets, suggesting that a smaller tuning budget could be allocated to this hyperparameter.

Table 11: Performance comparison across different types of weight decay.

| Dataset | Method | Utility (Acc or AUC) | Worst | Gap | EqOdd | DP |
|---|---|---|---|---|---|---|
| **CelebA** | L1 | $86.52 \pm 0.16$ | $83.68 \pm 0.26$ | $6.82 \pm 0.42$ | $81.48 \pm 1.06$ | $67.10 \pm 1.03$ |
| | L2 | $\mathbf{86.57 \pm 0.18}$ | $\mathbf{83.76 \pm 0.23}$ | $\mathbf{6.76 \pm 0.34}$ | $81.91 \pm 0.58$ | $67.20 \pm 0.69$ |
| | No | $86.52 \pm 0.20$ | $83.65 \pm 0.34$ | $6.91 \pm 0.43$ | $\mathbf{82.15 \pm 1.37}$ | $67.69 \pm 1.06$ |
| | $\Delta_{L2-No}$ | 0.05 | 0.11 | -0.15 | -0.24 | -0.49 |
| **UTKFace** | L1 | $93.02 \pm 0.34$ | $91.85 \pm 0.54$ | $2.71 \pm 1.21$ | $97.21 \pm 1.12$ | $\mathbf{95.13 \pm 1.23}$ |
| | L2 | $92.75 \pm 0.54$ | $91.78 \pm 0.61$ | $\mathbf{2.26 \pm 0.64}$ | $\mathbf{97.62 \pm 0.53}$ | $94.55 \pm 1.20$ |
| | No | $\mathbf{93.11 \pm 0.49}$ | $\mathbf{91.86 \pm 0.85}$ | $2.92 \pm 1.01$ | $97.22 \pm 0.97$ | $94.69 \pm 1.26$ |
| | $\Delta_{L2-No}$ | -0.36 | -0.08 | -0.66 | 0.40 | -0.14 |
| **FairFace** | L1 | $66.60 \pm 0.16$ | $66.07 \pm 0.26$ | $1.13 \pm 0.65$ | $96.20 \pm 0.37$ | $\mathbf{97.61 \pm 0.25}$ |
| | L2 | $66.76 \pm 0.21$ | $66.34 \pm 0.37$ | $0.87 \pm 0.51$ | $\mathbf{96.22 \pm 0.57}$ | $97.61 \pm 0.41$ |
| | No | $\mathbf{67.01 \pm 0.17}$ | $\mathbf{66.71 \pm 0.31}$ | $\mathbf{0.63 \pm 0.54}$ | $95.99 \pm 0.52$ | $97.47 \pm 0.42$ |
| | $\Delta_{L2-No}$ | -0.25 | -0.37 | 0.24 | 0.23 | 0.14 |
| **Facet** | L1 | $\mathbf{67.85 \pm 0.19}$ | $63.71 \pm 0.69$ | $5.41 \pm 0.90$ | $\mathbf{96.84 \pm 0.85}$ | $\mathbf{96.12 \pm 0.68}$ |
| | L2 | $67.55 \pm 0.37$ | $\mathbf{64.25 \pm 0.97}$ | $\mathbf{4.31 \pm 1.13}$ | $96.47 \pm 1.23$ | $95.40 \pm 1.12$ |
| | No | $67.60 \pm 0.40$ | $64.24 \pm 0.64$ | $4.39 \pm 0.87$ | $96.26 \pm 1.09$ | $95.39 \pm 1.02$ |
| | $\Delta_{L2-No}$ | -0.05 | 0.01 | -0.08 | 0.21 | 0.01 |
| **HAM10000** | L1 | $87.66 \pm 1.11$ | $83.70 \pm 3.05$ | $3.86 \pm 3.46$ | $87.75 \pm 4.03$ | $\mathbf{83.95 \pm 5.10}$ |
| | L2 | $\mathbf{88.35 \pm 1.83}$ | $84.68 \pm 2.02$ | $4.11 \pm 2.08$ | $\mathbf{88.17 \pm 3.10}$ | $82.22 \pm 4.78$ |
| | No | $88.12 \pm 1.89$ | $\mathbf{84.85 \pm 2.03}$ | $\mathbf{3.42 \pm 1.54}$ | $88.14 \pm 4.13$ | $83.78 \pm 3.87$ |
| | $\Delta_{L2-No}$ | 0.23 | -0.17 | 0.69 | 0.03 | -1.56 |
| **Fitz17k** | L1 | $82.26 \pm 1.68$ | $79.79 \pm 2.52$ | $4.70 \pm 2.39$ | $\mathbf{97.10 \pm 1.45}$ | $\mathbf{97.91 \pm 0.45}$ |
| | L2 | $\mathbf{89.74 \pm 1.00}$ | $\mathbf{88.39 \pm 1.05}$ | $\mathbf{2.92 \pm 1.14}$ | $94.92 \pm 2.48$ | $94.46 \pm 1.40$ |
| | No | $89.65 \pm 0.96$ | $88.30 \pm 1.03$ | $2.94 \pm 1.10$ | $95.08 \pm 2.80$ | $94.36 \pm 1.31$ |
| | $\Delta_{L2-No}$ | 0.09 | 0.09 | -0.02 | -0.16 | 0.10 |
| **Waterbirds** | L1 | $77.46 \pm 0.64$ | $76.79 \pm 1.09$ | $\mathbf{1.33 \pm 0.93}$ | $\mathbf{88.83 \pm 2.35}$ | $\mathbf{91.68 \pm 1.99}$ |
| | L2 | $\mathbf{85.63 \pm 1.36}$ | $\mathbf{84.20 \pm 0.94}$ | $2.87 \pm 0.85$ | $66.53 \pm 3.31$ | $77.67 \pm 3.52$ |
| | No | $85.38 \pm 0.68$ | $83.38 \pm 1.58$ | $4.00 \pm 2.01$ | $69.10 \pm 1.04$ | $76.53 \pm 1.56$ |
| | $\Delta_{L2-No}$ | 0.25 | 0.82 | -1.13 | -2.57 | 1.14 |

### C.1.5 IMPACT OF MODEL SIZE

**Increasing model size does not consistently improve fairness.** We analyze the effect of model size on fairness by comparing different ResNet architectures (ResNet-18, ResNet-34, ResNet-50, and ResNet-101) across all seven datasets in Table 12, with corresponding visualizations provided in Figure 10. We observe that larger models tend to have slightly higher utility (overall accuracy or AUC), however, this improvement is not consistent in the disadvantaged group in terms of the worst accuracy/AUC. For fairness metrics, larger models exhibit varying trends across datasets without an obvious pattern. The results suggest that increasing model size does not offer a reliable path toward improved fairness for single-modality classic supervised learning.

### C.1.6 SENSITIVITY ANALYSIS

While the previous sections analyzed training choices in isolation, Table 13 provides a unified quantitative comparison of their relative impact. We aggregated the standard deviations from Table 9, 10, and 12 to measure the model's sensitivity to that specific choice in Table 13. Our results reveal that optimizer choice yields the highest standard deviations across the majority of datasets and metrics (23 in 35). This aggregate view reinforces our earlier qualitative findings. First, optimizer

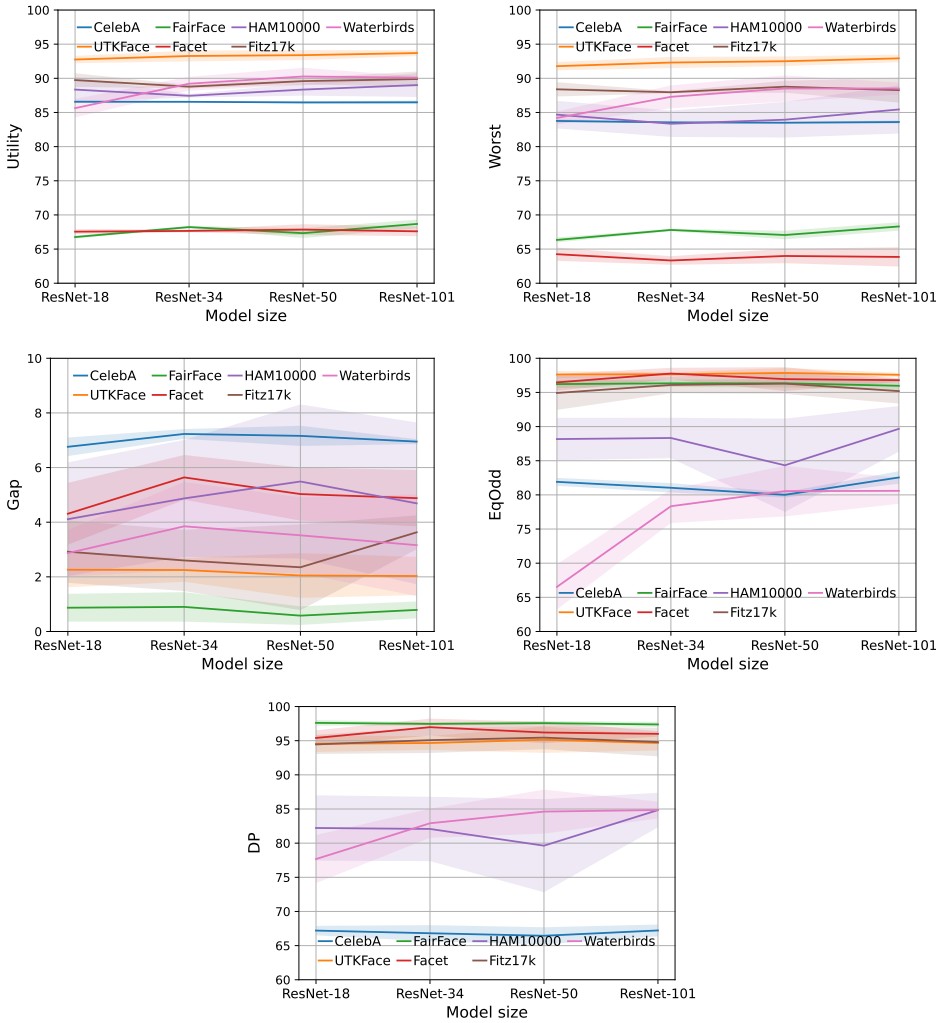

Figure 10: Effect of model size on utility and fairness metrics across seven datasets, derived from Table 12. Each subplot corresponds to one evaluation metric, and curves represent different datasets. The x-axis uses ResNet depth (18/34/50/101) as evenly spaced categorical positions for visual clarity.

choice is the most leverage-efficient knob for fairness-aware HPO: the search space is relatively small (a few commonly used optimizers), yet it leads most of the variance in both accuracy and disparity. In contrast, batch size does influence utility and fairness on specific datasets like Fitz17k, but the potential search space is much larger (a wide integer range), making exhaustive tuning significantly more expensive for a lower expected payout in fairness gains. Finally, model size generally exhibits the lowest sensitivity, while increasing model depth significantly increases computational complexity. For fairness-sensitive applications under resource constraints, it is therefore more effective to fix a moderate model size and allocate tuning budget to optimization hyperparameters rather than to model sizes.

Table 12: Performance comparison across different model sizes.

| Dataset | Batch Size | Utility (Acc or AUC) | Worst | Gap | EqOdd | DP |
|---|---|---|---|---|---|---|
| CelebA | ResNet-18 | **86.57 ± 0.18** | **83.76 ± 0.23** | **6.76 ± 0.34** | 81.91 ± 0.58 | 67.20 ± 0.69 |
| | ResNet-34 | 86.56 ± 0.03 | 83.55 ± 0.11 | 7.23 ± 0.18 | 81.04 ± 0.69 | 66.81 ± 1.19 |
| | ResNet-50 | 86.47 ± 0.18 | 83.50 ± 0.10 | 7.16 ± 0.37 | 80.01 ± 0.40 | 66.44 ± 1.25 |
| | ResNet-101 | 86.49 ± 0.26 | 83.60 ± 0.23 | 6.95 ± 0.09 | **82.55 ± 0.92** | **67.22 ± 0.87** |
| | **Std** | 0.04 | 0.10 | 0.18 | 0.95 | 0.32 |
| UTKFace | ResNet-18 | 92.75 ± 0.54 | 91.78 ± 0.61 | 2.26 ± 0.64 | 97.62 ± 0.53 | 94.55 ± 1.20 |
| | ResNet-34 | 93.27 ± 0.79 | 92.31 ± 0.82 | 2.25 ± 0.43 | 97.66 ± 0.28 | 94.66 ± 1.01 |
| | ResNet-50 | 93.39 ± 0.73 | 92.50 ± 0.71 | 2.05 ± 0.82 | **97.85 ± 0.74** | **95.14 ± 1.93** |
| | ResNet-101 | **93.70 ± 0.44** | **92.92 ± 0.56** | **2.03 ± 0.70** | 97.58 ± 0.30 | 94.66 ± 1.10 |
| | **Std** | 0.34 | 0.41 | 0.11 | 0.10 | 0.23 |
| FairFace | ResNet-18 | 66.76 ± 0.21 | 66.34 ± 0.37 | 0.87 ± 0.51 | 96.22 ± 0.57 | **97.61 ± 0.41** |
| | ResNet-34 | 68.22 ± 0.17 | 67.80 ± 0.18 | 0.90 ± 0.54 | 96.33 ± 0.55 | 97.46 ± 0.25 |
| | ResNet-50 | 67.33 ± 0.72 | 67.06 ± 0.63 | **0.58 ± 0.34** | **96.34 ± 0.61** | 97.57 ± 0.32 |
| | ResNet-101 | **68.68 ± 0.64** | **68.31 ± 0.62** | 0.79 ± 0.31 | 95.97 ± 0.68 | 97.38 ± 0.38 |
| | **Std** | 0.75 | 0.75 | 0.12 | 0.15 | 0.09 |
| Facet | ResNet-18 | 67.55 ± 0.37 | **64.25 ± 0.97** | **4.31 ± 1.13** | 96.47 ± 1.23 | 95.40 ± 1.12 |
| | ResNet-34 | 67.66 ± 0.20 | 63.33 ± 0.63 | 5.64 ± 0.82 | **97.76 ± 0.85** | **96.98 ± 1.25** |
| | ResNet-50 | **67.85 ± 0.79** | 63.99 ± 1.06 | 5.03 ± 0.97 | 96.95 ± 1.75 | 96.20 ± 1.56 |
| | ResNet-101 | 67.60 ± 0.71 | 63.85 ± 1.41 | 4.88 ± 1.03 | 96.79 ± 0.26 | 96.00 ± 0.41 |
| | **Std** | 0.11 | 0.34 | 0.47 | 0.48 | 0.56 |
| HAM10000 | ResNet-18 | 88.35 ± 1.83 | 84.68 ± 2.02 | **4.11 ± 2.08** | 88.17 ± 3.10 | 82.22 ± 4.78 |
| | ResNet-34 | 87.44 ± 0.26 | 83.35 ± 1.93 | 4.87 ± 2.14 | 88.33 ± 2.96 | 82.08 ± 4.71 |
| | ResNet-50 | 88.35 ± 0.89 | 83.94 ± 2.61 | 5.49 ± 2.82 | 84.33 ± 6.83 | 79.63 ± 6.81 |
| | ResNet-101 | **89.01 ± 1.74** | **85.44 ± 3.52** | 4.69 ± 2.96 | **89.67 ± 3.34** | **84.84 ± 2.53** |
| | **Std** | 0.56 | 0.79 | 0.49 | 1.99 | 1.84 |
| Fitz17k | ResNet-18 | 89.74 ± 1.00 | 88.39 ± 1.05 | 2.92 ± 1.14 | 94.92 ± 2.48 | 94.46 ± 1.40 |
| | ResNet-34 | 88.78 ± 0.26 | 87.96 ± 0.16 | 2.60 ± 1.12 | 96.09 ± 1.21 | 95.08 ± 1.88 |
| | ResNet-50 | 89.59 ± 0.50 | **88.77 ± 0.88** | **2.35 ± 1.57** | **96.29 ± 1.50** | **95.45 ± 1.70** |
| | ResNet-101 | **89.89 ± 1.10** | 88.27 ± 1.82 | 3.63 ± 0.62 | 95.19 ± 1.82 | 94.80 ± 2.10 |
| | **Std** | 0.43 | 0.29 | 0.48 | 0.58 | 0.36 |
| Waterbirds | ResNet-18 | 85.63 ± 1.36 | 84.20 ± 0.94 | **2.87 ± 0.85** | 66.53 ± 3.31 | 77.67 ± 3.52 |
| | ResNet-34 | 89.21 ± 1.02 | 87.29 ± 1.68 | 3.85 ± 1.62 | 78.34 ± 2.47 | 82.91 ± 2.13 |
| | ResNet-50 | **90.27 ± 1.28** | **88.51 ± 1.92** | 3.52 ± 1.39 | 80.55 ± 3.71 | 84.62 ± 3.22 |
| | ResNet-101 | 90.09 ± 0.22 | **88.51 ± 0.88** | 3.16 ± 1.86 | **80.59 ± 1.90** | **84.85 ± 1.23** |
| | **Std** | 1.87 | 1.76 | 0.37 | 5.83 | 2.89 |

Table 13: Standard deviations comparison across different training choices.

| Dataset | Tuning | Utility | Worst | Gap | EqOdd | DP |
|---|---|---|---|---|---|---|
| CelebA | Optimizer | **0.21** | **0.35** | **0.43** | **1.00** | **0.67** |
| | Batch Size | 0.05 | 0.07 | 0.09 | 0.35 | 0.19 |
| | Model Size | 0.04 | 0.10 | 0.18 | 0.95 | 0.32 |
| UTKFace | Optimizer | **0.38** | **0.50** | **0.51** | **0.46** | 0.45 |
| | Batch Size | 0.21 | 0.36 | 0.41 | 0.41 | **0.51** |
| | Model Size | 0.34 | 0.41 | 0.11 | 0.10 | 0.23 |
| FairFace | Optimizer | 0.69 | **0.89** | **0.49** | **0.20** | 0.13 |
| | Batch Size | 0.64 | 0.68 | 0.40 | 0.12 | **0.17** |
| | Model Size | **0.75** | 0.75 | 0.12 | 0.15 | 0.09 |
| Facet | Optimizer | **0.52** | **0.72** | **1.11** | **0.95** | 0.69 |
| | Batch Size | 0.19 | 0.53 | 0.52 | 0.58 | **0.92** |
| | Model Size | 0.11 | 0.34 | 0.47 | 0.48 | 0.56 |
| HAM10000 | Optimizer | **0.79** | **1.34** | 0.68 | 1.75 | **2.88** |
| | Batch Size | 0.53 | 0.96 | **0.73** | 0.80 | 1.23 |
| | Model Size | 0.56 | 0.79 | 0.49 | **1.99** | 1.84 |
| Fitz17k | Optimizer | 0.86 | 1.15 | **0.54** | **0.77** | **1.04** |
| | Batch Size | **1.22** | **1.28** | 0.17 | 0.50 | 0.71 |
| | Model Size | 0.43 | 0.29 | 0.48 | 0.58 | 0.36 |
| Waterbirds | Optimizer | 0.98 | 1.40 | **0.90** | 2.94 | 2.35 |
| | Batch Size | 1.52 | **1.89** | 0.83 | 4.38 | **3.60** |
| | Model Size | **1.87** | 1.76 | 0.37 | **5.83** | 2.89 |

### C.1.7 Impact of model selections

Table 14: Comparison of model selections. M denotes selecting based on the maximal overall utility.

| Dataset | CelebA | | | | | UTKFACE | | | | | FairFace | | | | | Facet | | | | |
|---|---|---|---|---|---|---|---|---|---|---|---|---|---|---|---|---|---|---|---|---|
| | ACC | Worst | Gap | EqOdd | DP | ACC | Worst | Gap | EqOdd | DP | ACC | Worst | Gap | EqOdd | DP | ACC | Worst | Gap | EqOdd | DP |
| ERM-M | 86.56 | 83.69 | 6.90 | 81.89 | 67.48 | 92.78 | 91.68 | 2.58 | 97.42 | 94.44 | 66.76 | 66.34 | 0.87 | 96.22 | 97.61 | 67.90 | 64.16 | 4.89 | 96.90 | 95.95 |
| ERM-DTO | 86.57 | 83.76 | 6.76 | 81.91 | 67.20 | 92.75 | 91.78 | 2.26 | 97.62 | 94.55 | 66.76 | 66.34 | 0.87 | 96.22 | 97.61 | 67.55 | 64.25 | 4.31 | 96.47 | 95.40 |
| RandAug-M | 86.73 | 83.89 | 6.82 | 81.16 | 66.90 | 93.31 | 92.17 | 2.64 | 97.32 | 95.11 | 68.50 | 67.70 | 1.70 | 96.17 | 97.57 | 67.97 | 64.55 | 4.46 | 96.81 | 96.02 |
| RandAug-FWH | 86.72 | 83.89 | 6.80 | 81.39 | 66.99 | 93.19 | 92.19 | 2.34 | 97.62 | 94.83 | 68.37 | 67.69 | 1.44 | 96.14 | 97.55 | 67.83 | 64.94 | 3.78 | 96.68 | 95.91 |
| GroupDRO-M | 86.14 | 83.51 | 6.32 | 78.76 | 66.32 | 92.37 | 91.43 | 2.18 | 97.46 | 94.46 | 65.81 | 65.03 | 1.64 | 96.53 | 97.66 | 67.32 | 64.19 | 4.09 | 93.57 | 93.16 |
| GroupDRO-FWH | 86.12 | 83.50 | 6.31 | 78.73 | 66.41 | 92.45 | 91.41 | 2.44 | 97.06 | 94.78 | 65.51 | 65.22 | 0.60 | 96.31 | 97.47 | 67.20 | 64.07 | 4.08 | 93.76 | 93.16 |

| Dataset | HAM10000 | | | | | Fitz17k | | | | | Waterbirds | | | | |
|---|---|---|---|---|---|---|---|---|---|---|---|---|---|---|---|
| | ACC | Worst | Gap | EqOdd | DP | ACC | Worst | Gap | EqOdd | DP | ACC | Worst | Gap | EqOdd | DP |
| ERM-M | 88.27 | 83.61 | 5.30 | 86.99 | 82.41 | 89.82 | 88.46 | 3.01 | 94.72 | 94.33 | 85.63 | 84.20 | 2.87 | 66.53 | 77.67 |
| ERM-DTO | 88.35 | 84.67 | 4.11 | 86.79 | 81.48 | 89.74 | 88.39 | 2.92 | 94.92 | 94.46 | 85.63 | 84.20 | 2.87 | 66.53 | 77.67 |
| RandAug-M | 89.53 | 85.33 | 4.63 | 86.34 | 79.94 | 91.24 | 90.13 | 2.52 | 95.36 | 94.48 | 87.48 | 85.60 | 3.76 | 72.12 | 80.35 |
| RandAug-FWH | 89.09 | 84.67 | 4.99 | 88.43 | 84.73 | 91.29 | 90.15 | 2.51 | 95.61 | 94.53 | 86.09 | 84.52 | 3.14 | 68.99 | 77.25 |
| GroupDRO-M | 88.96 | 85.23 | 4.33 | 86.12 | 79.45 | 91.04 | 89.66 | 3.55 | 95.22 | 94.43 | 86.78 | 84.76 | 4.05 | 71.88 | 80.88 |
| GroupDRO-FWH | 87.66 | 83.98 | 4.98 | 90.94 | 83.86 | 90.72 | 90.06 | 1.92 | 94.36 | 95.24 | 85.46 | 84.45 | 2.02 | 67.31 | 76.91 |

We use ERM, RandAug, and GroupDRO as examples to compare the model selection methods in Table 14 since their differences are more obvious to observe compared with others. The DTO-based selection prioritizes models that maximize utility for all groups, ensuring that no subgroup is disproportionately disadvantaged. On the other hand, the FWH selection emphasizes fairness without significantly harming utility based on the DTO-based method selected ERM results, focusing on reducing disparities without major accuracy degradation. The *M* in the Table denotes using maximal overall utility. FWH selection achieves a smaller fairness gap across multiple datasets. These results show that prioritizing fairness constraints does not necessarily mean sacrificing model performance, as long as careful selection strategies are employed. However, since the selection is performed on the validation set, it still faces the potential trade-offs when validation-test discrepancies exist.

### C.2 Computational overhead comparison

In this section, we compare the computational overhead of various methods in terms of additional model parameters and overall computation times. Since methods like **FIS** perform additional computations during validation, we measure the total computational time, including the training, validation, and testing phases. Notably, because **FIS** requires computing the gradient for each sample, we use PyTorch's vmap to accelerate the computation, which aligns with the official **FIS** implementation in JAX.

In Table 15, we report the additional parameters required by specific methods and the average computational times (in seconds) for the evaluated methods. Most methods use the same number of parameters as the standard ERM baseline, so we focus only on those that introduce extra model parameters. The reported times are averaged over 5 runs, including the training, validation, and testing phases. For datasets with a limited number of samples, time differences are negligible and thus omitted from this comparison. Although some methods, like **LAFTR**, exhibit significantly increased computational demands, they do not outperform simpler methods. This highlights the need for easy yet effective approaches that balance performance with computational efficiency.

Table 15: Average computation time in seconds for one round and the extra number of parameters introduced by specific methods. DFR is not included here since it is a post-processing method.

| | ERM | RandAug | Mixup | Resampling | BM | FIS |
|---|---|---|---|---|---|---|
| Parameters | – | – | – | – | 1024 | – |
| CelebA | 113.06 ± 0.25 | 117.06 ± 1.95 | 112.48 ± 1.19 | 112.80 ± 0.38 | 114.08 ± 1.30 | 145.29 ± 0.45 |
| UTKFace | 39.33 ± 1.89 | 39.55 ± 1.09 | 38.44 ± 0.53 | 38.83 ± 1.12 | 39.79 ± 1.25 | 41.65 ± 0.57 |
| FairFace | 54.54 ± 0.31 | 54.53 ± 0.48 | 54.90 ± 0.34 | 54.68 ± 0.39 | 55.40 ± 0.38 | 72.07 ± 0.14 |
| Facet | 26.44 ± 0.15 | 26.51 ± 0.26 | 26.69 ± 0.21 | 26.53 ± 0.20 | 27.21 ± 0.24 | 35.14 ± 0.22 |
| Average | 58.34 | 59.42 | 58.13 | 58.21 | 59.12 | 73.54 |

| | Decoupled | LAFTR | FSCL | GapReg | MCDP | GroupDRO |
|---|---|---|---|---|---|---|
| Parameters | 1024 | 65921 | – | – | – | – |
| CelebA | 114.24 ± 0.62 | 175.77 ± 0.86 | 119.75 ± 2.06 | 112.92 ± 0.71 | 112.94 ± 0.95 | 113.75 ± 1.39 |
| UTKFace | 38.69 ± 0.95 | 40.56 ± 1.10 | 47.62 ± 0.36 | 38.94 ± 0.43 | 38.97 ± 0.75 | 39.00 ± 1.00 |
| FairFace | 55.38 ± 0.61 | 83.76 ± 0.41 | 54.81 ± 0.94 | 55.63 ± 0.54 | 55.63 ± 0.35 | 55.31 ± 0.66 |
| Facet | 26.74 ± 0.18 | 39.64 ± 0.18 | 26.19 ± 0.19 | 26.85 ± 0.25 | 26.82 ± 0.28 | 26.71 ± 0.16 |
| Average | 58.76 | 84.93 | 62.09 | 58.59 | 58.59 | 58.69 |

## C.3 LARGE VISION-LANGUAGE MODEL EXPERIMENTS

### C.3.1 MODEL SCALE IMPACT ON FAIRNESS

Table 16: Effect of LVLM model scale on accuracy and fairness metrics

| Dataset | Metric | LLaVA-1.6 7B | 13B | 34B | Qwen2.5-VL 7B | 32B | 72B | Gemma 3 4B | 12B | 27B | Llama 3.2-11B | 3.2-90B | 4-Scout-109B |
|---|---|---|---|---|---|---|---|---|---|---|---|---|---|
| CelebA | ACC | 50.21 ± 0.20 | 46.50 ± 0.13 | 44.83 ± 0.05 | 65.02 ± 0.25 | 42.58 ± 0.17 | 68.76 ± 0.09 | 45.08 ± 0.16 | 66.23 ± 0.24 | 74.04 ± 0.05 | 82.04 ± 0.21 | 76.23 ± 0.11 | **83.71 ± 0.18** |
| | Worst | 46.79 ± 0.61 | 38.93 ± 0.35 | 32.69 ± 0.27 | 63.41 ± 0.27 | 27.79 ± 0.17 | 66.02 ± 0.35 | 37.15 ± 0.22 | 65.17 ± 0.37 | 72.18 ± 0.04 | 78.00 ± 0.45 | 68.79 ± 0.49 | **80.54 ± 0.21** |
| | Gap | 5.85 ± 0.71 | 12.95 ± 0.45 | 20.75 ± 0.51 | 2.75 ± 0.55 | 25.33 ± 0.55 | 4.70 ± 0.71 | 13.57 ± 0.44 | **1.83 ± 0.39** | 4.47 ± 0.03 | 9.64 ± 0.64 | 17.89 ± 0.90 | 7.62 ± 0.44 |
| | EqOdd | 88.81 ± 0.27 | 92.44 ± 0.17 | **97.41 ± 0.16** | 95.27 ± 0.76 | 95.26 ± 0.54 | 92.69 ± 0.30 | 92.25 ± 0.22 | 89.72 ± 0.49 | 92.76 ± 0.75 | 86.63 ± 0.64 | 94.56 ± 3.57 | 84.81 ± 0.92 |
| | DP | 77.63 ± 0.22 | 83.16 ± 0.24 | 91.92 ± 0.15 | 81.51 ± 0.94 | **99.26 ± 0.12** | 91.71 ± 0.20 | 83.41 ± 0.40 | 73.20 ± 0.41 | 74.53 ± 0.80 | 76.33 ± 0.33 | 88.27 ± 1.46 | 71.52 ± 0.49 |
| UTKFace | ACC | 96.76 ± 0.26 | 97.01 ± 0.04 | 97.12 ± 0.19 | 96.12 ± 0.09 | **97.61 ± 0.22** | 96.78 ± 0.38 | 96.33 ± 0.31 | 97.25 ± 0.23 | 97.25 ± 0.46 | 96.81 ± 0.25 | 97.28 ± 0.13 | 97.02 ± 0.17 |
| | Worst | 95.71 ± 0.29 | 96.28 ± 0.12 | 96.34 ± 0.12 | 95.03 ± 0.20 | 96.86 ± 0.12 | 95.76 ± 0.42 | 95.41 ± 0.22 | 96.70 ± 0.09 | **96.89 ± 0.39** | 96.04 ± 0.21 | 96.31 ± 0.04 | 96.29 ± 0.24 |
| | Gap | 2.44 ± 0.58 | 1.71 ± 0.39 | 1.81 ± 0.20 | 2.51 ± 0.66 | 1.74 ± 0.16 | 2.36 ± 0.81 | 2.15 ± 0.48 | 1.29 ± 0.35 | **0.85 ± 0.21** | 1.81 ± 0.25 | 2.26 ± 0.37 | 1.70 ± 0.27 |
| | EqOdd | 97.51 ± 0.62 | 98.31 ± 0.39 | 98.16 ± 0.24 | 97.09 ± 0.63 | 98.19 ± 0.22 | 97.59 ± 0.82 | 97.68 ± 0.50 | 98.71 ± 0.37 | **99.20 ± 0.21** | 98.03 ± 0.29 | 97.73 ± 0.38 | 98.27 ± 0.29 |
| | DP | 95.15 ± 1.28 | 94.97 ± 1.64 | 95.22 ± 1.33 | 93.05 ± 1.40 | 93.96 ± 1.63 | 94.33 ± 1.63 | 93.54 ± 1.38 | 94.84 ± 1.15 | **95.28 ± 1.11** | 93.53 ± 1.15 | 94.18 ± 1.29 | 94.90 ± 1.31 |
| FairFace | ACC | 53.07 ± 0.00 | 59.59 ± 0.00 | **66.91 ± 0.00** | 66.09 ± 0.02 | 63.98 ± 0.01 | 65.57 ± 0.01 | 52.73 ± 0.07 | 57.27 ± 2.82 | 59.40 ± 0.05 | 51.70 ± 0.15 | 61.26 ± 0.19 | 56.52 ± 0.17 |
| | Worst | 49.98 ± 0.00 | 58.54 ± 0.00 | **66.04 ± 0.00** | 65.07 ± 0.02 | 62.65 ± 0.03 | 64.73 ± 0.01 | 50.55 ± 0.09 | 55.35 ± 3.04 | 56.71 ± 0.01 | 49.73 ± 0.27 | 60.96 ± 0.26 | 53.53 ± 0.05 |
| | Gap | 6.55 ± 0.00 | 1.97 ± 0.00 | 1.84 ± 0.00 | 2.17 ± 0.03 | 2.83 ± 0.04 | 1.79 ± 0.03 | 4.63 ± 0.07 | 4.08 ± 0.30 | 5.71 ± 0.10 | 4.18 ± 0.31 | **0.63 ± 0.19** | 6.35 ± 0.38 |
| | EqOdd | 95.04 ± 0.00 | 94.55 ± 0.00 | 96.22 ± 0.00 | 96.04 ± 0.01 | 95.50 ± 0.02 | 96.26 ± 0.01 | 96.01 ± 0.10 | 93.07 ± 1.93 | 95.13 ± 0.08 | 95.99 ± 0.12 | 95.55 ± 0.18 | **96.54 ± 0.11** |
| | DP | 96.34 ± 0.00 | 95.64 ± 0.00 | **98.07 ± 0.00** | 97.21 ± 0.01 | 97.23 ± 0.01 | 97.57 ± 0.01 | 97.50 ± 0.04 | 95.80 ± 1.12 | 97.16 ± 0.06 | 96.76 ± 0.01 | 96.58 ± 0.04 | 98.00 ± 0.05 |
| Facet | ACC | 63.50 ± 0.21 | 66.80 ± 0.25 | 58.14 ± 0.23 | 67.91 ± 0.46 | 67.80 ± 0.33 | **68.41 ± 0.50** | 66.93 ± 0.37 | 54.47 ± 0.17 | 57.75 ± 0.74 | 66.74 ± 0.42 | 67.75 ± 0.47 | 67.54 ± 0.42 |
| | Worst | 62.41 ± 0.26 | 62.90 ± 1.11 | 57.97 ± 0.35 | 64.06 ± 1.20 | 63.16 ± 1.33 | 63.79 ± 1.30 | 62.39 ± 0.94 | 53.82 ± 0.50 | 57.25 ± 0.52 | 61.23 ± 0.49 | 63.05 ± 1.05 | **64.61 ± 1.04** |
| | Gap | 1.42 ± 0.18 | 5.09 ± 1.29 | **0.72 ± 0.56** | 5.02 ± 1.00 | 6.06 ± 1.56 | 6.04 ± 1.18 | 5.92 ± 0.91 | 2.80 ± 1.81 | 1.42 ± 1.21 | 7.19 ± 0.93 | 6.12 ± 0.78 | 3.83 ± 0.99 |
| | EqOdd | 94.98 ± 1.59 | **98.19 ± 0.89** | 95.90 ± 1.41 | 96.29 ± 0.90 | 97.67 ± 0.75 | 97.04 ± 1.16 | 97.88 ± 0.40 | 97.63 ± 0.42 | 96.22 ± 1.00 | 93.68 ± 1.85 | 97.23 ± 1.06 | 97.55 ± 1.09 |
| | DP | 93.28 ± 1.44 | 96.79 ± 1.14 | 93.91 ± 1.47 | 95.81 ± 0.91 | 97.22 ± 0.57 | 96.15 ± 1.42 | **98.28 ± 0.95** | 96.13 ± 0.70 | 94.56 ± 0.60 | 95.72 ± 1.70 | 96.05 ± 1.08 | 96.28 ± 0.97 |
| Waterbirds | ACC | 77.24 ± 0.00 | 79.46 ± 0.00 | 86.95 ± 0.00 | 91.68 ± 0.00 | 91.62 ± 0.01 | **92.41 ± 0.00** | 75.59 ± 1.79 | 92.20 ± 0.03 | 90.77 ± 0.02 | 75.41 ± 0.32 | 77.62 ± 0.19 | 88.62 ± 0.09 |
| | Worst | 75.63 ± 0.00 | 76.08 ± 0.00 | 81.26 ± 0.00 | 90.68 ± 0.00 | 90.46 ± 0.02 | 91.27 ± 0.00 | 62.40 ± 4.24 | **91.98 ± 0.03** | 88.97 ± 0.19 | 61.61 ± 0.56 | 63.35 ± 0.30 | 85.74 ± 0.17 |
| | Gap | 3.21 ± 0.00 | 6.77 ± 0.00 | 11.39 ± 0.00 | 2.00 ± 0.00 | 2.32 ± 0.02 | 2.28 ± 0.00 | 26.39 ± 4.90 | **0.44 ± 0.07** | 3.59 ± 0.35 | 27.59 ± 0.52 | 28.55 ± 0.32 | 5.76 ± 0.16 |
| | EqOdd | 63.22 ± 0.00 | 65.54 ± 0.00 | 80.88 ± 0.00 | 90.21 ± 0.00 | 91.90 ± 0.01 | **94.03 ± 0.00** | 61.61 ± 0.65 | 89.28 ± 0.19 | 92.99 ± 0.48 | 65.72 ± 0.63 | 71.54 ± 0.58 | 89.38 ± 0.13 |
| | DP | 72.83 ± 0.00 | 72.45 ± 0.00 | 80.46 ± 0.00 | 92.13 ± 0.00 | 93.12 ± 0.02 | **94.62 ± 0.00** | 58.82 ± 3.16 | 92.36 ± 0.10 | 93.17 ± 0.14 | 60.97 ± 0.59 | 64.46 ± 0.39 | 89.44 ± 0.12 |
| Average | ACC | 68.16 ± 19.18 | 69.87 ± 19.30 | 70.79 ± 21.23 | 77.36 ± 15.21 | 72.72 ± 22.28 | 78.39 ± 14.93 | 67.33 ± 20.11 | 73.48 ± 19.95 | 75.84 ± 17.90 | 74.54 ± 16.85 | 76.03 ± 13.61 | **78.68 ± 16.40** |
| | Worst | 66.10 ± 20.09 | 66.55 ± 21.30 | 66.86 ± 24.10 | 75.65 ± 15.79 | 68.18 ± 27.41 | **76.31 ± 15.80** | 61.58 ± 21.59 | 72.60 ± 20.38 | 74.40 ± 18.24 | 69.32 ± 18.01 | 70.49 ± 14.72 | 76.14 ± 17.04 |
| | Gap | 3.89 ± 2.21 | 5.70 ± 4.58 | 7.30 ± 8.67 | 2.89 ± 1.23 | 7.66 ± 10.02 | 3.43 ± 1.84 | 10.53 ± 9.84 | **2.09 ± 1.40** | 3.21 ± 2.05 | 10.08 ± 10.23 | 11.09 ± 11.87 | 5.05 ± 2.32 |
| | EqOdd | 87.91 ± 14.17 | 89.81 ± 13.79 | 93.71 ± 7.23 | 94.98 ± 2.74 | **95.70 ± 2.49** | 95.52 ± 2.08 | 89.09 ± 15.52 | 93.68 ± 4.37 | 95.26 ± 2.64 | 88.01 ± 13.18 | 91.32 ± 11.13 | 93.31 ± 5.93 |
| | DP | 87.05 ± 10.97 | 88.60 ± 10.58 | 91.92 ± 6.78 | 91.94 ± 6.18 | **96.16 ± 2.55** | 94.88 ± 2.19 | 86.31 ± 16.47 | 90.47 ± 9.76 | 90.94 ± 9.29 | 84.66 ± 15.64 | 87.91 ± 13.52 | 90.03 ± 10.83 |

In this paper, we include four families of vision–language foundation models: LLaVA, Qwen-VL, Gemma, and Llama at multiple parameter scales (from roughly 4 billion to 90 billion weights) to probe whether the standard "bigger-is-better" rule extends to fairness (Table 16). The averaged results confirm that scale almost always boosts utility: mean accuracy climbs steadily within each family (e.g., LLaVA 68 → 71 %, Gemma 67 → 76 %), and worst-group accuracy rises in the same trend. Larger models also score better on our parity metrics like demographic parity (DP) and equalized odds (EqOdd), which suggests that they become more confident and consistent across groups. However, larger models do not guarantee a better accuracy gap. For example, the gap more than doubles (3.9 → 7.3 %) for LLaVA. In short, increasing size is a reliable path to higher accuracy and better parity rates, but it does not guarantee a smaller disparity between groups; in some cases, it even amplifies it. These mixed trends highlight that parameter count alone cannot close fairness gaps where architectural choices, alignment strategies, and dataset biases remain decisive factors.

We further select the largest variant from each model family and present the results in Figure 11. Among them, Qwen2.5-VL 72B demonstrates the best overall balance between fairness and utility. Notably, Gemma 3 27B achieves comparable performance to Qwen despite its smaller model size, while LLaMA 4-Scout, although achieving strong headline accuracy, remains susceptible to dataset-specific biases.

### C.3.2 PROMPT SENSITIVITY OF LVLMs

Fairness evaluations of LVLMs often rely on handcrafted prompts, raising concerns about robustness to prompt variations. To examine the sensitivity of LVLMs to prompt wording, we design two paraphrased variants for three dataset tasks (we consider those with relatively larger disparities) in addition to the original instruction (Table 17), and evaluate whether performance holds consistent across different formulations. Results in Table 18 show that, across most cases, models exhibit broadly consistent fairness performance under different prompts, with only a few extreme cases (e.g., Qwen2.5-VL on CelebA) showing large deviations. This indicates that prompt rewording alone does not fundamentally resolve fairness issues, LVLMs remain subject to subgroup disparities regardless of the prompt design.

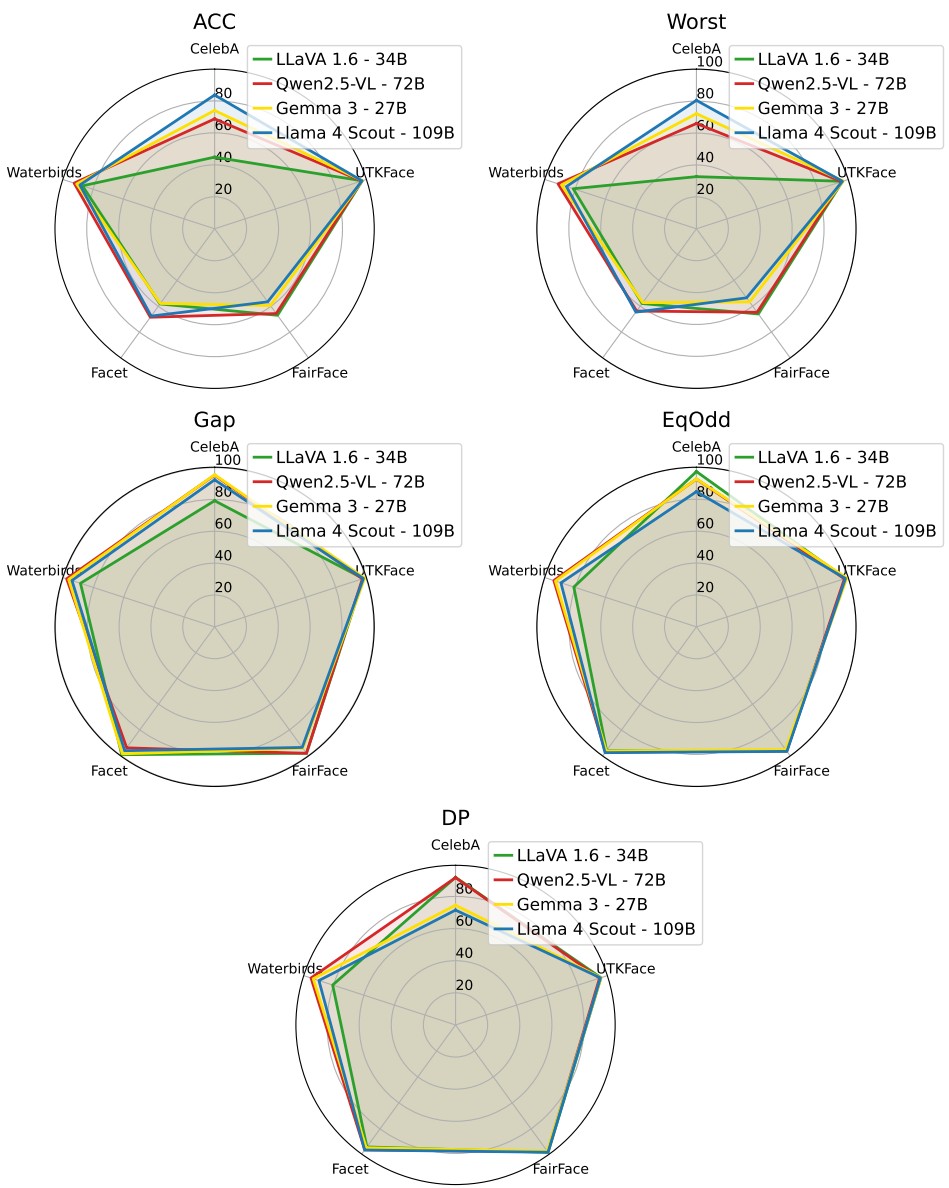

Figure 11: Performance comparison of LVLMs across five datasets and five evaluation metrics. For clarity, the Gap metric is shown as $100 - Gap$, where higher values indicate better performance.

Table 17: Prompts used for evaluating LVLM sensitivity. Each dataset has the original instruction (Origin) and two paraphrased variants.

| Dataset | Prompts |
|---|---|
| CelebA | **Origin:** Is the person in the photo wavy-haired? Answer 'Yes' for wavy hair, 'No' for non-wavy hair. 
 **Variant 1:** Does this person appear to have wavy hair? Reply with Yes/No only. 
 **Variant 2:** Classify hair texture as wavy or non-wavy. Output: Yes or No. |
| FairFace | **Origin:** Based on the photo, classify the race of the person into: 0: White, 1: Black, 2: Latino or Hispanic, 3: East Asian, 4: Southeast Asian, 5: Indian, 6: Middle Eastern. Answer with the category number only. 
 **Variant 1:** Pick a single integer label (0–6) for race: 0-White, 1-Black, 2-Latino/Hispanic, 3-East Asian, 4-Southeast Asian, 5-Indian, 6-Middle Eastern. Output digits only. 
 **Variant 2:** Classify race using numbers: 0=White, 1=Black, 2=Latino/Hispanic, 3=East Asian, 4=Southeast Asian, 5=Indian, 6=Middle Eastern. Answer with number only. |
| Facet | **Origin:** Does the photo show a visible face? Answer 'Yes' if the face is visible, otherwise 'No'. 
 **Variant 1:** Is a face clearly visible? Reply Yes/No only. 
 **Variant 2:** Face visibility check: Yes or No? Output one word. |

Table 18: LVLM performance under different prompts. Each model is evaluated with the original instruction and two paraphrased variants. For CelebA, Llama3.2 often produces non-conforming answers under Variant 2 instead of the required categorical outputs; it is therefore omitted from the reported results.

| Dataset | Metric | LLaVA-1.6-13B | | | Qwen2.5-VL-32B | | | Gemma3-12B | | | Llama3.2-11B | | |
|---|---|---|---|---|---|---|---|---|---|---|---|---|---|
| | | Origin | Variant 1 | Variant 2 | Origin | Variant 1 | Variant 2 | Origin | Variant 1 | Variant 2 | Origin | Variant 1 | Variant 2 |
| CelebA | ACC | $46.50 \pm 0.13$ | $53.84 \pm 0.17$ | $50.30 \pm 0.14$ | $42.58 \pm 0.17$ | $33.47 \pm 0.11$ | $37.08 \pm 0.03$ | $66.23 \pm 0.24$ | $63.76 \pm 0.24$ | $74.77 \pm 0.15$ | $82.04 \pm 0.21$ | $81.34 \pm 0.22$ | - |
| | Worst | $38.93 \pm 0.35$ | $46.81 \pm 0.36$ | $45.10 \pm 0.39$ | $27.79 \pm 0.17$ | $15.16 \pm 0.23$ | $18.22 \pm 0.02$ | $65.17 \pm 0.37$ | $61.51 \pm 0.37$ | $64.98 \pm 0.23$ | $78.00 \pm 0.45$ | $77.10 \pm 0.38$ | - |
| | Gap | $12.95 \pm 0.45$ | $12.03 \pm 0.41$ | $8.89 \pm 0.59$ | $25.33 \pm 0.55$ | $31.32 \pm 0.59$ | $32.30 \pm 0.02$ | $1.83 \pm 0.39$ | $3.86 \pm 0.35$ | $23.58 \pm 0.33$ | $9.64 \pm 0.64$ | $10.20 \pm 0.49$ | - |
| | EqOdd | $92.44 \pm 0.17$ | $94.23 \pm 0.21$ | $91.04 \pm 0.33$ | $95.26 \pm 0.54$ | $98.61 \pm 0.12$ | $93.73 \pm 0.02$ | $89.72 \pm 0.49$ | $90.24 \pm 0.26$ | $93.32 \pm 0.23$ | $86.63 \pm 0.64$ | $92.56 \pm 0.41$ | - |
| | DP | $83.16 \pm 0.24$ | $87.06 \pm 0.44$ | $80.21 \pm 0.22$ | $99.26 \pm 0.12$ | $99.00 \pm 0.12$ | $94.81 \pm 0.01$ | $73.20 \pm 0.41$ | $74.62 \pm 0.19$ | $92.66 \pm 0.59$ | $76.33 \pm 0.33$ | $79.98 \pm 0.63$ | - |
| FairFace | ACC | $59.59 \pm 0.00$ | $61.57 \pm 0.00$ | $59.17 \pm 0.00$ | $63.98 \pm 0.01$ | $63.13 \pm 0.02$ | $63.00 \pm 0.01$ | $57.27 \pm 2.82$ | $57.15 \pm 0.05$ | $58.19 \pm 0.08$ | $51.70 \pm 0.15$ | $31.19 \pm 0.07$ | $40.66 \pm 0.18$ |
| | Worst | $58.54 \pm 0.00$ | $58.98 \pm 0.00$ | $58.01 \pm 0.00$ | $62.65 \pm 0.03$ | $61.41 \pm 0.01$ | $61.43 \pm 0.02$ | $55.35 \pm 3.04$ | $54.97 \pm 0.01$ | $56.09 \pm 0.06$ | $49.73 \pm 0.27$ | $28.99 \pm 0.30$ | $38.88 \pm 0.51$ |
| | Gap | $1.97 \pm 0.00$ | $5.49 \pm 0.00$ | $2.47 \pm 0.00$ | $2.83 \pm 0.04$ | $3.65 \pm 0.03$ | $3.33 \pm 0.06$ | $4.08 \pm 0.30$ | $4.64 \pm 0.11$ | $4.45 \pm 0.08$ | $4.18 \pm 0.31$ | $4.67 \pm 0.58$ | $3.78 \pm 0.72$ |
| | EqOdd | $94.55 \pm 0.00$ | $97.04 \pm 0.00$ | $96.82 \pm 0.00$ | $95.50 \pm 0.02$ | $97.11 \pm 0.01$ | $95.86 \pm 0.04$ | $93.07 \pm 1.93$ | $92.13 \pm 0.05$ | $92.12 \pm 0.12$ | $95.99 \pm 0.12$ | $97.11 \pm 0.2$ | $95.15 \pm 0.09$ |
| | DP | $95.64 \pm 0.00$ | $97.54 \pm 0.00$ | $97.06 \pm 0.00$ | $97.23 \pm 0.01$ | $97.75 \pm 0.01$ | $97.56 \pm 0.01$ | $95.80 \pm 1.12$ | $95.15 \pm 0.07$ | $95.38 \pm 0.04$ | $96.76 \pm 0.01$ | $98.69 \pm 0.09$ | $97.19 \pm 0.14$ |
| Facet | ACC | $66.80 \pm 0.25$ | $66.63 \pm 0.18$ | $55.20 \pm 0.48$ | $67.80 \pm 0.33$ | $67.19 \pm 0.13$ | $67.59 \pm 0.33$ | $54.47 \pm 0.17$ | $55.51 \pm 0.27$ | $55.88 \pm 0.47$ | $66.74 \pm 0.42$ | $63.69 \pm 0.08$ | $38.03 \pm 0.00$ |
| | Worst | $62.90 \pm 1.11$ | $63.26 \pm 0.55$ | $54.28 \pm 0.67$ | $63.16 \pm 1.33$ | $62.40 \pm 1.53$ | $62.41 \pm 1.34$ | $53.82 \pm 0.05$ | $54.91 \pm 0.08$ | $55.08 \pm 0.28$ | $61.23 \pm 0.49$ | $60.03 \pm 0.47$ | $36.65 \pm 0.20$ |
| | Gap | $5.09 \pm 1.29$ | $4.38 \pm 0.78$ | $3.95 \pm 0.90$ | $6.06 \pm 1.56$ | $6.24 \pm 1.84$ | $6.75 \pm 1.30$ | $2.80 \pm 1.81$ | $2.57 \pm 1.10$ | $1.41 \pm 0.50$ | $7.19 \pm 0.93$ | $4.75 \pm 0.67$ | $6.02 \pm 0.83$ |
| | EqOdd | $98.19 \pm 0.89$ | $98.16 \pm 0.46$ | $97.19 \pm 0.79$ | $97.67 \pm 0.75$ | $97.27 \pm 0.44$ | $96.70 \pm 0.05$ | $97.63 \pm 0.42$ | $96.97 \pm 1.30$ | $95.77 \pm 1.56$ | $93.68 \pm 1.85$ | $96.24 \pm 1.08$ | $98.79 \pm 0.25$ |
| | DP | $96.79 \pm 1.14$ | $96.79 \pm 1.42$ | $98.92 \pm 0.31$ | $97.22 \pm 0.57$ | $96.87 \pm 0.32$ | $95.67 \pm 0.17$ | $96.13 \pm 0.70$ | $95.60 \pm 1.33$ | $93.87 \pm 1.82$ | $95.72 \pm 1.70$ | $97.24 \pm 0.93$ | $99.48 \pm 0.44$ |

### C.3.3 OPEN-ENDED GENERATION WITH LVLMS

To complement the closed-set classification prompts used above, we also evaluate LVLMs on open-ended generation tasks. We use UTKFace, FairFace, and Waterbirds, replacing the classification prompts in Table 18 with open-ended instructions encouraging unconstrained image description:

- **UTKFace**: "Describe the person in this photo, including their apparent gender presentation and any other observable characteristics."
- **FairFace**: "Describe this person's ethnic and cultural background, including physical features, cultural markers, and any other relevant characteristics you can observe."
- **Waterbirds**: "Describe this bird in detail, including its species characteristics, habitat preferences, and behavioral traits you can infer from the image."

We use the same models and evaluation splits as in previous experiments. To make the results comparable to our classification-based evaluation, we post-process the generated text into a prediction in the original label space with a small lexicon of label-specific keywords and common synonyms.:

- **UTKFace**: words such as "man", "male", "boy", "gentleman" vs. "woman", "female", "girl", "lady".
- **FairFace**: terms such as "White/Caucasian/European", "Black/African/African American", "Latino/Latin/Hispanic", "East Asian/Chinese/Korean/Japanese)", "Southeast Asian/Thai/Vietnamese/Filipino)", "Indian/South Asian/Desi", and "Middle Eastern/Arab/Persian".
- **Waterbirds**: class names and taxonomic descriptors used in Sagawa* et al. (2020); Wah et al. (2011), including *albatross, auklet, cormorant, frigatebird, fulmar, gull, jaeger, kittiwake, pelican, puffin, tern, gadwall, grebe, mallard, merganser, guillemot*, and *Pacific loon*. We additionally use coarse-grained keywords such as "aquatic bird" / "waterfowl" versus "landbird" / "terrestrial bird" to supplement the class matching.

We use the same models and evaluation splits as in prior experiments. To map generated text back into the original label space, we apply case-insensitive keyword matching using small lexicons of class-specific terms and common synonyms; these cases constitute a small fraction of the data.

Table 19: LVLMs comparison under open-ended generation.

| Dataset | Metric | LLaVA-1.6 | | | Qwen2.5-VL | | | Gemma 3 | | | Llama | | |
|---|---|---|---|---|---|---|---|---|---|---|---|---|---|
| | | 7B | 13B | 34B | 7B | 32B | 72B | 4B | 12B | 27B | 3.2-11B | 3.2-90B | 4-Scout-109B |
| UTKFace | ACC | 92.24 | 92.93 | 95.38 | 97.26 | 97.31 | 96.14 | 96.71 | 96.83 | 96.71 | 97.38 | 98.02 | 96.09 |
| | Worst | 91.99 | 91.98 | 94.80 | 96.94 | 96.87 | 95.07 | 95.86 | 96.28 | 96.09 | 96.62 | 97.17 | 94.97 |
| | Gap | 0.59 | 2.15 | 1.32 | 0.72 | 1.01 | 2.44 | 1.95 | 1.24 | 1.42 | 1.73 | 1.82 | 2.60 |
| | EqOdd | 99.09 | 97.41 | 98.57 | 98.84 | 98.90 | 97.32 | 97.96 | 98.65 | 98.53 | 98.23 | 98.07 | 97.15 |
| | DP | 95.58 | 94.44 | 95.40 | 94.32 | 94.87 | 93.69 | 95.33 | 95.31 | 96.13 | 97.27 | 92.69 | 92.69 |
| FairFace | ACC | - | - | - | 51.93 | 41.96 | 60.88 | 51.29 | 51.58 | 52.91 | - | - | 52.92 |
| | Worst | - | - | - | 51.47 | 40.43 | 60.30 | 50.86 | 51.32 | 51.26 | - | - | 50.08 |
| | Gap | - | - | - | 0.88 | 2.94 | 1.24 | 0.92 | 0.55 | 3.48 | - | - | 6.07 |
| | EqOdd | - | - | - | 96.09 | 95.20 | 97.08 | 96.31 | 96.53 | 95.76 | - | - | 96.43 |
| | DP | - | - | - | 96.16 | 97.85 | 97.76 | 97.85 | 97.78 | 97.51 | - | - | 98.12 |
| Waterbirds | ACC | 82.10 | 85.63 | 83.84 | 95.63 | 95.16 | 95.89 | 92.59 | 93.65 | 94.56 | 92.90 | 95.23 | 93.35 |
| | Worst | 76.48 | 82.20 | 78.50 | 95.02 | 94.34 | 95.01 | 91.07 | 92.21 | 93.24 | 90.70 | 93.86 | 91.76 |
| | Gap | 11.59 | 7.02 | 10.82 | 1.25 | 1.65 | 1.76 | 3.08 | 2.99 | 2.70 | 4.44 | 2.75 | 3.20 |
| | EqOdd | 80.79 | 81.93 | 79.42 | 95.11 | 94.38 | 96.73 | 93.27 | 92.22 | 93.91 | 92.70 | 96.38 | 92.94 |
| | DP | 79.30 | 82.12 | 77.61 | 95.21 | 94.60 | 96.54 | 92.40 | 92.09 | 93.82 | 90.78 | 95.36 | 92.46 |

As shown in Table 19, the open-ended generation exhibits utility and fairness trends that closely mirror those observed in our LVLM classifications. Although accuracy generally increases or remains stable with model scale, fairness metrics such as *Gap*, *EqOdd*, and *DP* do not improve monotonically. For instance, on UTKFace, the Qwen2.5 sees an increase in the gap, despite high overall accuracy.

Furthermore, we observe that certain model families frequently refuse to generate descriptions for FairFace: LLaVA-1.6 exhibits refusal rates of 31% (7B), 26% (13B), and 47% (34B). The trend is even more pronounced in the Llama 3.2 series, where aggressive safety filters result in refusal rates of 81.1% for the 11B model and 95.0% for the 90B model. Consequently, the remaining valid outputs are insufficient for a robust fairness assessment, highlighting a critical trade-off: while safety alignment is essential, its over-application can render models untestable for demographic disparities in open-ended generations.

# D    ADDING NEW DATASETS AND ALGORITHMS TO THE FRAMEWORK

Our benchmarking framework provides a flexible and extensible design for integrating new image classification datasets. A new dataset can be inherited from `FairDataset` and specify image folder location, target labels, sensitive attributes, and an indices list to split the dataset. The core dataset class, `FairDataset`, is presented as below.

```python
class FairDataset(Dataset):
    def __init__(self, root, split='train', transform=None, seed=42):
        # Base folder and images directory
        self.root = root
        self.img_dir = None

        self.split = split
        self.indices = []  # Indices for the split dataset
        self.sensitive_attrs = np.array([])   # Sensitive attributes
        self.targets = np.array([])  # Target labels

        self.transform = transform  # Image transformation pipeline

        self.image_file_list = []

        # Option to store images in memory for faster access
        self.images = []
```

By inheriting from `ERM`, this new class has access to the default model, optimizer, and loss function. A new method needs to be implemented to rewrite the training process, ensuring that it aligns with the new algorithm's objectives. For example, the `MCDP` class extends `ERM` and introduces a fairness-aware loss function.

```python
class MCDP(erm):
    def __init__(self, args):
        super().__init__(args)
        self.fair_loss = MaxCDFdp(args.mcdp_temperature)

    def train(self, train_loader, epoch, args):
        self.model.train()
        for batch_index, (data, target, sensitive_attr) in
        ↪   enumerate(train_loader):
            output = self.model(data)
            self.optimizer.zero_grad()
            fairloss = args.mcdp_lambda * self.fair_loss(output,
            ↪   sensitive_attr)
            loss = self.criterion(output, target) + fairloss
            loss.backward()
            self.optimizer.step()
```

# E    SUPPLEMENTARY RESULTS

Here we present the standard deviation for Table 2 and Table 21 below:

Table 20: Full results on seven datasets with standard deviations for vision models.

| Dataset | Metric | ERM | RandAug | Mixup | Resampling | BM | FIS | Decoupled | LAFTR | FSCL | GapReg | MCDP | GroupDRO | DFR | Oxonfair |
|---|---|---|---|---|---|---|---|---|---|---|---|---|---|---|---|
| CelebA | ACC | 86.57 ± 0.18 | 86.72 ± 0.19 | 85.61 ± 0.27 | 86.35 ± 0.16 | 85.93 ± 0.14 | 83.05 ± 0.23 | 86.35 ± 0.17 | 86.55 ± 0.19 | 85.61 ± 0.14 | 85.62 ± 0.59 | 80.26 ± 0.75 | 86.12 ± 0.17 | 86.58 ± 0.15 | 86.49 ± 0.23 |
| | Worst | 83.76 ± 0.23 | 83.89 ± 0.33 | 82.74 ± 0.36 | 83.44 ± 0.22 | 82.86 ± 0.34 | 79.33 ± 0.50 | 83.46 ± 0.24 | 83.67 ± 0.33 | 82.56 ± 0.32 | 83.17 ± 0.40 | 77.13 ± 0.83 | 83.50 ± 0.23 | 83.78 ± 0.19 | 83.63 ± 0.30 |
| | Gap | 6.76 ± 0.34 | 6.80 ± 0.44 | 6.90 ± 0.29 | 6.98 ± 0.33 | 7.38 ± 0.50 | 8.94 ± 0.72 | 6.93 ± 0.42 | 6.93 ± 0.49 | 7.35 ± 0.59 | 5.90 ± 0.89 | 7.52 ± 1.23 | 6.31 ± 0.51 | 6.74 ± 0.41 | 6.87 ± 0.28 |
| | EqOdd | 81.91 ± 0.58 | 81.73 ± 1.12 | 87.08 ± 2.24 | 81.80 ± 0.71 | 78.84 ± 1.12 | 75.79 ± 1.84 | 80.59 ± 1.26 | 81.15 ± 1.66 | 85.45 ± 1.44 | 93.94 ± 5.76 | 89.63 ± 3.27 | 78.73 ± 1.30 | 81.83 ± 0.71 | 78.10 ± 2.53 |
| | DP | 67.20 ± 0.69 | 67.37 ± 0.42 | 70.83 ± 1.75 | 67.39 ± 0.31 | 66.91 ± 1.03 | 66.84 ± 1.84 | 66.68 ± 0.88 | 66.90 ± 1.22 | 69.72 ± 1.61 | 75.91 ± 4.47 | 93.11 ± 2.20 | 66.41 ± 1.16 | 67.30 ± 0.69 | 65.10 ± 1.80 |
| UTKFace | ACC | 92.75 ± 0.54 | 93.19 ± 0.31 | 92.62 ± 0.61 | 92.70 ± 0.53 | 93.33 ± 0.61 | 91.97 ± 0.46 | 91.68 ± 1.10 | 93.17 ± 0.27 | 93.52 ± 0.50 | 92.53 ± 0.88 | 92.49 ± 0.72 | 92.45 ± 0.39 | 92.73 ± 0.58 | 92.36 ± 0.86 |
| | Worst | 91.78 ± 0.61 | 92.19 ± 0.30 | 91.55 ± 0.74 | 91.60 ± 0.65 | 92.27 ± 0.81 | 90.91 ± 0.54 | 90.84 ± 0.85 | 92.05 ± 0.28 | 92.62 ± 0.40 | 91.70 ± 0.95 | 91.63 ± 1.14 | 91.41 ± 0.62 | 91.60 ± 0.93 | 91.11 ± 1.00 |
| | Gap | 2.26 ± 0.65 | 2.34 ± 0.59 | 2.50 ± 0.36 | 2.56 ± 0.43 | 2.47 ± 0.54 | 2.48 ± 0.38 | 1.97 ± 0.98 | 2.61 ± 0.31 | 2.10 ± 0.77 | 1.91 ± 0.59 | 2.00 ± 0.99 | 2.44 ± 0.99 | 2.63 ± 0.83 | 2.91 ± 1.04 |
| | EqOdd | 97.62 ± 0.53 | 97.62 ± 0.46 | 97.61 ± 0.41 | 97.49 ± 0.42 | 97.39 ± 0.50 | 97.51 ± 0.37 | 97.43 ± 0.44 | 97.44 ± 0.33 | 97.44 ± 1.29 | 98.10 ± 0.61 | 98.04 ± 0.97 | 97.06 ± 0.63 | 97.39 ± 0.87 | 96.41 ± 0.67 |
| | DP | 94.55 ± 1.20 | 94.83 ± 1.12 | 94.51 ± 1.46 | 95.34 ± 1.76 | 94.34 ± 1.52 | 94.69 ± 1.70 | 96.27 ± 2.42 | 95.44 ± 2.07 | 94.18 ± 2.11 | 95.30 ± 1.62 | 95.80 ± 0.64 | 94.78 ± 1.85 | 94.83 ± 1.42 | 94.68 ± 2.42 |
| FairFace | ACC | 66.76 ± 0.21 | 68.37 ± 0.30 | 65.40 ± 0.59 | 65.40 ± 0.19 | 65.66 ± 1.08 | 65.31 ± 0.51 | 67.03 ± 0.54 | 66.44 ± 0.78 | 65.42 ± 0.50 | 65.02 ± 1.02 | 66.06 ± 1.07 | 65.51 ± 0.52 | 63.20 ± 2.15 | – |
| | Worst | 66.34 ± 0.37 | 67.69 ± 0.35 | 64.50 ± 0.58 | 64.51 ± 0.41 | 65.20 ± 1.02 | 64.59 ± 0.37 | 66.61 ± 0.54 | 65.60 ± 0.86 | 64.64 ± 0.39 | 64.12 ± 1.28 | 65.62 ± 0.94 | 65.22 ± 0.46 | 62.45 ± 2.45 | – |
| | Gap | 0.87 ± 0.51 | 1.44 ± 0.62 | 1.93 ± 0.28 | 1.90 ± 0.59 | 0.97 ± 0.19 | 1.53 ± 0.57 | 0.87 ± 0.49 | 1.76 ± 0.40 | 1.66 ± 0.53 | 1.92 ± 0.94 | 0.90 ± 0.48 | 0.60 ± 0.58 | 1.59 ± 0.68 | – |
| | EqOdd | 96.22 ± 0.57 | 96.14 ± 0.08 | 96.55 ± 0.27 | 96.83 ± 0.33 | 95.70 ± 0.34 | 95.88 ± 0.32 | 95.14 ± 0.75 | 94.73 ± 1.04 | 97.05 ± 0.45 | 96.15 ± 0.12 | 97.85 ± 0.25 | 96.31 ± 0.73 | 95.81 ± 0.25 | – |
| | DP | 97.61 ± 0.41 | 97.55 ± 0.24 | 97.62 ± 0.24 | 97.80 ± 0.21 | 97.30 ± 0.21 | 97.40 ± 0.16 | 96.88 ± 0.43 | 96.86 ± 0.58 | 98.10 ± 0.11 | 97.51 ± 0.22 | 98.50 ± 0.23 | 97.47 ± 0.37 | 97.43 ± 0.10 | – |
| Facet | ACC | 67.55 ± 0.37 | 67.83 ± 0.69 | 67.86 ± 0.44 | 67.56 ± 0.24 | 65.87 ± 1.19 | 67.60 ± 0.43 | 67.33 ± 0.74 | 70.74 ± 7.49 | 67.79 ± 0.44 | 67.01 ± 0.24 | 67.91 ± 0.35 | 67.20 ± 0.24 | 66.87 ± 0.47 | 68.09 ± 0.41 |
| | Worst | 64.25 ± 0.97 | 64.94 ± 0.96 | 64.54 ± 1.01 | 64.13 ± 0.51 | 62.67 ± 0.91 | 63.53 ± 0.92 | 62.60 ± 1.08 | 68.60 ± 8.56 | 65.02 ± 0.94 | 62.22 ± 1.45 | 64.21 ± 1.22 | 64.07 ± 0.82 | 63.10 ± 1.51 | 64.11 ± 1.66 |
| | Gap | 4.31 ± 1.13 | 3.78 ± 0.97 | 4.33 ± 1.43 | 4.48 ± 0.42 | 4.18 ± 0.55 | 5.33 ± 1.34 | 6.17 ± 0.58 | 2.82 ± 1.39 | 3.61 ± 0.79 | 6.26 ± 1.66 | 4.84 ± 1.53 | 4.08 ± 0.87 | 4.92 ± 1.93 | 5.21 ± 1.71 |
| | EqOdd | 96.47 ± 1.23 | 96.68 ± 1.02 | 97.50 ± 1.22 | 94.37 ± 0.82 | 96.32 ± 1.77 | 98.10 ± 1.31 | 88.04 ± 1.87 | 96.38 ± 2.81 | 96.50 ± 1.65 | 98.92 ± 1.01 | 98.23 ± 0.97 | 93.76 ± 1.04 | 96.82 ± 0.96 | 83.40 ± 9.30 |
| | DP | 95.40 ± 1.12 | 95.91 ± 0.97 | 96.71 ± 1.16 | 93.96 ± 0.67 | 95.09 ± 1.85 | 97.84 ± 1.24 | 87.30 ± 1.73 | 95.00 ± 2.97 | 95.66 ± 1.38 | 98.73 ± 1.40 | 98.46 ± 1.33 | 93.16 ± 1.17 | 96.09 ± 1.12 | 83.95 ± 9.32 |
| HAM10000 | AUC | 88.35 ± 1.83 | 89.09 ± 1.26 | 86.51 ± 2.44 | 87.75 ± 2.21 | 89.54 ± 1.38 | 85.97 ± 1.09 | 87.87 ± 2.02 | 86.71 ± 1.79 | 89.40 ± 2.52 | 84.97 ± 3.18 | 82.96 ± 1.84 | 87.66 ± 2.72 | 87.06 ± 0.93 | 88.46 ± 2.01 |
| | Worst | 84.68 ± 2.02 | 84.67 ± 3.29 | 82.31 ± 2.33 | 84.77 ± 3.34 | 86.49 ± 1.49 | 82.98 ± 1.64 | 84.04 ± 3.62 | 81.68 ± 3.62 | 85.89 ± 3.15 | 82.57 ± 3.66 | 80.29 ± 1.70 | 83.98 ± 3.00 | 82.49 ± 2.48 | 83.83 ± 3.46 |
| | Gap | 4.11 ± 2.08 | 4.99 ± 3.70 | 4.14 ± 2.75 | 3.52 ± 2.68 | 3.04 ± 0.90 | 3.11 ± 1.85 | 5.17 ± 2.73 | 6.18 ± 2.99 | 3.71 ± 1.47 | 3.07 ± 2.34 | 3.10 ± 2.25 | 4.98 ± 2.51 | 5.30 ± 2.19 | 5.46 ± 2.89 |
| | EqOdd | 88.17 ± 3.10 | 88.43 ± 4.05 | 91.14 ± 5.43 | 91.84 ± 3.32 | 85.65 ± 4.07 | 94.05 ± 2.34 | 77.52 ± 14.21 | 93.28 ± 6.23 | 84.57 ± 2.32 | 98.15 ± 3.71 | 99.52 ± 0.95 | 90.94 ± 5.33 | 91.25 ± 4.95 | 99.27 ± 0.60 |
| | DP | 82.22 ± 4.78 | 84.73 ± 3.36 | 88.54 ± 8.88 | 85.05 ± 2.22 | 78.41 ± 4.89 | 88.58 ± 2.72 | 75.15 ± 11.56 | 91.00 ± 6.42 | 77.64 ± 4.51 | 96.74 ± 6.52 | 99.58 ± 0.85 | 83.86 ± 5.70 | 84.65 ± 4.61 | 99.34 ± 0.58 |
| Fitz17k | AUC | 89.74 ± 1.00 | 91.29 ± 0.52 | 90.62 ± 0.93 | 90.76 ± 0.72 | 91.02 ± 0.58 | 88.34 ± 0.76 | 89.63 ± 0.49 | 90.95 ± 0.50 | 90.71 ± 1.25 | 89.59 ± 1.10 | 91.65 ± 0.69 | 90.72 ± 0.91 | 89.99 ± 0.86 | 89.56 ± 0.99 |
| | Worst | 88.39 ± 1.05 | 90.15 ± 0.96 | 89.38 ± 1.86 | 88.99 ± 1.95 | 89.93 ± 0.84 | 87.02 ± 1.08 | 88.45 ± 1.31 | 89.67 ± 1.12 | 89.77 ± 1.10 | 88.52 ± 2.61 | 90.49 ± 0.81 | 90.06 ± 0.89 | 88.57 ± 0.50 | 88.40 ± 1.03 |
| | Gap | 2.92 ± 1.14 | 2.51 ± 0.99 | 2.43 ± 1.42 | 3.62 ± 2.19 | 2.34 ± 1.37 | 3.06 ± 1.98 | 2.55 ± 2.15 | 2.87 ± 2.02 | 2.22 ± 0.91 | 1.84 ± 2.01 | 2.87 ± 1.11 | 1.92 ± 1.07 | 2.93 ± 1.48 | 3.06 ± 1.08 |
| | EqOdd | 94.92 ± 2.48 | 95.61 ± 2.29 | 96.20 ± 3.05 | 95.27 ± 3.26 | 94.99 ± 1.87 | 95.17 ± 2.41 | 94.09 ± 3.28 | 93.68 ± 3.40 | 97.45 ± 0.79 | 95.47 ± 1.48 | 95.68 ± 2.68 | 94.36 ± 2.39 | 94.30 ± 1.11 | 99.26 ± 0.52 |
| | DP | 94.46 ± 1.40 | 94.53 ± 1.06 | 94.51 ± 1.37 | 93.31 ± 2.28 | 93.66 ± 2.13 | 95.60 ± 1.94 | 94.06 ± 1.91 | 94.46 ± 1.32 | 95.32 ± 1.40 | 96.42 ± 2.21 | 96.14 ± 2.08 | 95.24 ± 1.59 | 95.26 ± 1.19 | 99.88 ± 0.09 |
| Waterbirds | ACC | 85.63 ± 1.36 | 86.09 ± 1.40 | 87.67 ± 1.06 | 87.35 ± 0.79 | 88.20 ± 0.96 | 83.72 ± 0.85 | 74.64 ± 2.98 | 85.72 ± 1.48 | 86.83 ± 2.08 | 86.45 ± 3.89 | 85.98 ± 0.86 | 85.46 ± 1.48 | 89.83 ± 1.44 | 90.27 ± 0.17 |
| | Worst | 84.20 ± 0.94 | 84.52 ± 2.63 | 85.99 ± 1.53 | 84.85 ± 0.74 | 85.96 ± 1.06 | 82.67 ± 2.10 | 64.45 ± 5.72 | 83.29 ± 2.13 | 86.28 ± 2.34 | 85.72 ± 4.08 | 84.83 ± 1.36 | 84.45 ± 1.71 | 89.09 ± 1.71 | 89.52 ± 0.30 |
| | Gap | 2.87 ± 0.85 | 3.14 ± 2.52 | 3.36 ± 1.72 | 4.98 ± 0.34 | 4.48 ± 0.59 | 2.09 ± 2.92 | 20.38 ± 5.56 | 3.56 ± 1.18 | 1.10 ± 0.55 | 1.47 ± 0.87 | 2.31 ± 1.37 | 2.02 ± 1.39 | 1.47 ± 0.60 | 1.50 ± 0.92 |
| | EqOdd | 66.53 ± 3.31 | 68.99 ± 2.52 | 81.42 ± 4.38 | 90.87 ± 1.28 | 77.21 ± 3.25 | 65.89 ± 2.13 | 47.31 ± 4.15 | 68.22 ± 2.96 | 90.00 ± 1.82 | 87.48 ± 8.82 | 72.97 ± 3.35 | 67.31 ± 3.81 | 97.79 ± 0.75 | 94.99 ± 1.90 |
| | DP | 77.67 ± 3.52 | 77.25 ± 3.02 | 86.00 ± 4.59 | 90.93 ± 1.01 | 81.78 ± 2.33 | 75.30 ± 1.84 | 52.30 ± 5.81 | 77.47 ± 4.11 | 92.53 ± 1.37 | 91.39 ± 7.13 | 80.70 ± 1.77 | 76.91 ± 3.62 | 98.61 ± 0.90 | 97.38 ± 0.92 |

Table 21: Full results on seven datasets with standard deviations for multimodal models..

| Dataset | Metric | ERM | RandAug | BLIP2 | CLIP | Fairer-CLIP | CLIP-SFID | LLaVA-1.6 34B | Qwen2.5-VL 72B | Gemma3 27B | Llama4 Scout |
|---|---|---|---|---|---|---|---|---|---|---|---|
| CelebA | ACC | 86.57 ± 0.18 | 86.72 ± 0.19 | 47.38 ± 0.23 | 74.07 ± 0.41 | 73.78 ± 0.34 | 72.05 ± 0.27 | 44.83 ± 0.05 | 68.76 ± 0.09 | 74.04 ± 0.05 | 83.71 ± 0.18 |
| | Worst | 83.76 ± 0.23 | 83.89 ± 0.33 | 36.82 ± 0.59 | 67.43 ± 0.54 | 67.32 ± 0.55 | 66.36 ± 1.01 | 32.69 ± 0.27 | 66.02 ± 0.35 | 72.18 ± 0.03 | 80.54 ± 0.21 |
| | Gap | 6.76 ± 0.34 | 6.80 ± 0.44 | 18.09 ± 0.74 | 15.97 ± 0.40 | 15.56 ± 0.59 | 13.69 ± 1.86 | 20.75 ± 0.51 | 4.70 ± 0.71 | 4.47 ± 0.03 | 7.62 ± 0.44 |
| | EqOdd | 81.91 ± 0.58 | 81.73 ± 1.12 | 97.24 ± 0.57 | 83.72 ± 0.58 | 83.79 ± 0.78 | 95.70 ± 1.77 | 97.41 ± 0.16 | 92.69 ± 0.30 | 92.76 ± 0.75 | 84.81 ± 0.92 |
| | DP | 67.20 ± 0.69 | 67.37 ± 0.42 | 95.90 ± 0.65 | 81.32 ± 0.60 | 81.06 ± 0.58 | 93.23 ± 2.28 | 91.92 ± 0.15 | 91.71 ± 0.20 | 74.53 ± 0.80 | 71.52 ± 0.49 |
| UTKFace | ACC | 92.75 ± 0.54 | 93.19 ± 0.31 | 94.23 ± 0.40 | 96.72 ± 0.41 | 96.79 ± 0.37 | 96.70 ± 0.33 | 97.12 ± 0.19 | 96.78 ± 0.38 | 97.25 ± 0.46 | 97.02 ± 0.17 |
| | Worst | 91.78 ± 0.61 | 92.19 ± 0.30 | 94.00 ± 0.41 | 95.90 ± 0.25 | 96.05 ± 0.33 | 96.03 ± 0.28 | 96.34 ± 0.12 | 95.76 ± 0.42 | 96.89 ± 0.39 | 96.29 ± 0.24 |
| | Gap | 2.26 ± 0.65 | 2.34 ± 0.59 | 0.45 ± 0.28 | 1.90 ± 0.60 | 1.72 ± 0.22 | 1.55 ± 0.14 | 1.81 ± 0.20 | 2.36 ± 0.81 | 0.85 ± 0.21 | 1.70 ± 0.27 |
| | EqOdd | 97.62 ± 0.53 | 97.62 ± 0.46 | 99.24 ± 0.36 | 97.96 ± 0.52 | 98.17 ± 0.24 | 98.36 ± 0.16 | 98.16 ± 0.24 | 97.59 ± 0.82 | 99.20 ± 0.21 | 98.27 ± 0.29 |
| | DP | 94.55 ± 1.20 | 94.83 ± 1.12 | 95.61 ± 0.83 | 93.91 ± 1.11 | 94.27 ± 1.17 | 94.96 ± 1.12 | 95.22 ± 1.33 | 94.33 ± 1.63 | 95.28 ± 1.11 | 94.90 ± 1.31 |
| FairFace | ACC | 66.76 ± 0.21 | 68.37 ± 0.30 | 52.57 ± 0.00 | 57.36 ± 0.00 | 56.81 ± 0.00 | 52.73 ± 0.44 | 66.91 ± 0.00 | 65.57 ± 0.01 | 59.40 ± 0.05 | 56.52 ± 0.17 |
| | Worst | 66.34 ± 0.37 | 67.69 ± 0.35 | 50.74 ± 0.00 | 57.20 ± 0.00 | 56.41 ± 0.00 | 51.59 ± 0.62 | 66.04 ± 0.00 | 64.73 ± 0.01 | 56.71 ± 0.01 | 53.53 ± 0.05 |
| | Gap | 0.87 ± 0.51 | 1.44 ± 0.62 | 3.46 ± 0.00 | 0.34 ± 0.00 | 0.86 ± 0.00 | 2.40 ± 0.54 | 1.84 ± 0.00 | 1.79 ± 0.03 | 5.71 ± 0.10 | 6.35 ± 0.38 |
| | EqOdd | 96.22 ± 0.57 | 96.14 ± 0.08 | 93.86 ± 0.00 | 97.16 ± 0.00 | 96.78 ± 0.00 | 97.14 ± 0.12 | 96.22 ± 0.00 | 96.26 ± 0.01 | 95.13 ± 0.08 | 96.54 ± 0.11 |
| | DP | 97.61 ± 0.41 | 97.55 ± 0.24 | 96.89 ± 0.00 | 98.36 ± 0.00 | 98.32 ± 0.00 | 98.46 ± 0.09 | 98.07 ± 0.00 | 97.57 ± 0.01 | 97.16 ± 0.06 | 98.00 ± 0.05 |
| Facet | ACC | 67.55 ± 0.37 | 67.83 ± 0.69 | 41.16 ± 0.87 | 33.10 ± 0.25 | 33.17 ± 0.08 | 33.26 ± 0.23 | 58.14 ± 0.23 | 68.41 ± 0.50 | 57.75 ± 0.74 | 67.54 ± 0.42 |
| | Worst | 64.25 ± 0.97 | 64.94 ± 0.96 | 40.40 ± 0.91 | 31.43 ± 0.34 | 31.69 ± 0.16 | 31.54 ± 0.16 | 57.97 ± 0.35 | 63.79 ± 1.30 | 57.25 ± 0.52 | 64.61 ± 1.04 |
| | Gap | 4.31 ± 1.13 | 3.78 ± 0.97 | 3.22 ± 0.91 | 7.13 ± 0.94 | 6.32 ± 0.34 | 7.37 ± 1.22 | 0.72 ± 0.56 | 6.04 ± 1.18 | 1.42 ± 1.21 | 3.83 ± 0.99 |
| | EqOdd | 96.47 ± 1.23 | 96.68 ± 1.02 | 93.52 ± 1.44 | 98.72 ± 0.22 | 98.82 ± 0.33 | 99.81 ± 0.00 | 95.90 ± 1.41 | 97.04 ± 1.16 | 96.22 ± 1.00 | 96.35 ± 1.09 |
| | DP | 95.40 ± 1.12 | 95.91 ± 0.97 | 94.10 ± 1.20 | 98.92 ± 0.25 | 99.00 ± 0.36 | 99.80 ± 0.06 | 93.91 ± 1.47 | 96.15 ± 1.42 | 94.56 ± 0.60 | 96.28 ± 0.97 |
| Waterbirds | ACC | 85.63 ± 1.36 | 86.09 ± 1.40 | 52.30 ± 0.00 | 78.05 ± 0.00 | 77.77 ± 0.00 | 75.70 ± 0.34 | 86.95 ± 0.00 | 92.41 ± 0.00 | 90.77 ± 0.02 | 88.62 ± 0.09 |
| | Worst | 84.20 ± 0.94 | 84.52 ± 2.63 | 39.18 ± 0.00 | 74.53 ± 0.00 | 73.49 ± 0.00 | 72.81 ± 0.56 | 81.26 ± 0.00 | 91.27 ± 0.00 | 88.97 ± 0.19 | 85.74 ± 0.17 |
| | Gap | 2.87 ± 0.85 | 3.14 ± 2.52 | 26.23 ± 0.00 | 7.04 ± 0.00 | 8.56 ± 0.00 | 5.78 ± 0.47 | 11.39 ± 0.00 | 2.28 ± 0.00 | 3.59 ± 0.35 | 5.76 ± 0.16 |
| | EqOdd | 66.53 ± 3.31 | 68.99 ± 2.52 | 68.24 ± 0.00 | 73.96 ± 0.00 | 73.19 ± 0.00 | 95.57 ± 0.16 | 80.88 ± 0.00 | 94.03 ± 0.00 | 92.99 ± 0.48 | 89.38 ± 0.13 |
| | DP | 77.67 ± 3.52 | 77.25 ± 3.02 | 63.48 ± 0.00 | 78.12 ± 0.00 | 76.73 ± 0.00 | 94.09 ± 0.65 | 80.46 ± 0.00 | 94.62 ± 0.00 | 93.17 ± 0.14 | 89.44 ± 0.12 |
| HAM10000 | AUC | 88.35 ± 1.83 | 89.09 ± 1.26 | 38.87 ± 3.94 | 52.15 ± 2.59 | 51.88 ± 2.94 | 52.53 ± 3.94 | – | – | – | – |
| | Worst | 84.68 ± 2.02 | 84.67 ± 3.29 | 39.00 ± 4.98 | 51.56 ± 4.02 | 52.01 ± 3.88 | 53.41 ± 4.65 | – | – | – | – |
| | Gap | 4.11 ± 2.08 | 4.99 ± 3.70 | 6.89 ± 4.41 | 4.14 ± 3.92 | 3.12 ± 2.66 | 2.24 ± 2.13 | | | | |
| | EqOdd | 88.17 ± 3.10 | 88.43 ± 4.05 | 98.19 ± 1.11 | 96.24 ± 2.50 | 95.75 ± 2.39 | 99.16 ± 2.97 | – | – | – | – |
| | DP | 82.22 ± 4.78 | 84.73 ± 3.36 | 97.89 ± 1.15 | 99.09 ± 0.58 | 98.94 ± 0.66 | 98.32 ± 0.52 | – | – | – | – |
| Fitz17k | AUC | 89.74 ± 1.00 | 91.29 ± 0.52 | 67.08 ± 1.72 | 69.92 ± 0.83 | 69.81 ± 0.06 | 69.37 ± 0.84 | – | – | – | – |
| | Worst | 88.39 ± 1.05 | 90.15 ± 0.96 | 66.46 ± 1.95 | 69.78 ± 0.59 | 69.85 ± 0.11 | 69.35 ± 1.71 | – | – | – | – |
| | Gap | 2.92 ± 1.14 | 2.51 ± 0.99 | 3.74 ± 1.63 | 2.31 ± 1.54 | 2.52 ± 1.69 | 4.73 ± 4.58 | | | | |
| | EqOdd | 94.92 ± 2.48 | 95.61 ± 2.29 | 97.06 ± 1.56 | 89.87 ± 2.96 | 87.95 ± 3.74 | 85.88 ± 6.23 | – | – | – | – |
| | DP | 94.46 ± 1.40 | 94.53 ± 1.06 | 98.40 ± 1.66 | 95.03 ± 1.96 | 93.02 ± 1.19 | 92.19 ± 0.61 | – | – | – | – |

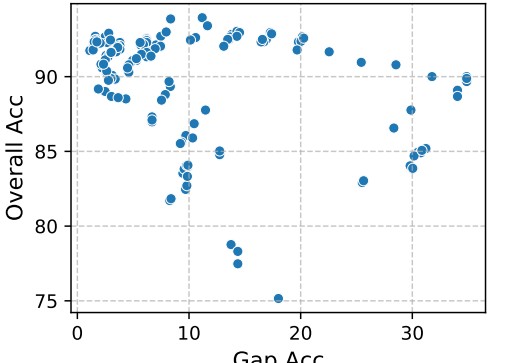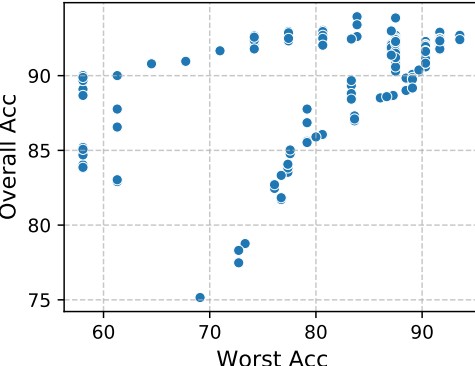

Figure 12: Intersectional analysis on UTKFace using race × age groups under different hyperparameter settings. Each point corresponds to a different configuration. Appropriate hyperparameter choices can reduce the gap while maintaining comparable overall accuracy.

## F    LIMITATIONS AND FUTURE DIRECTIONS

Despite its breadth, NH-Fair is only an intermediate step toward a truly comprehensive fairness benchmark, yet several gaps remain:

- **Dataset**: Although we span three domains, most datasets are still for human faces, due to the scarcity of open, well-annotated fairness datasets. Some datasets include multiple protected attributes. Due to computational resource limitations, we did not fully use them but selected the attribute with a clear performance gap. Future work may consider intersectional groups (e.g., gender × race) and more visual domains. We conducted a small-scale intersectional analysis on UTKFace, where we formed groups by combining race and age (below 60 and above 60). The scatter plots in Figure 12 show that intersectional biases might also be severe, but rigorous hyperparameter tuning can significantly mitigate intersectional disparities at comparable overall accuracy. Extending NH-Fair to systematically cover intersectional groups (e.g., gender × race) across more datasets and to include additional non-face visual domains is a direction for future work. In addition, our zero-shot evaluations cannot guarantee that images were not used during the pretraining phase of LVLMs, which may lead to potential data leakage.

- **Task scope**: To bridge single-modality bias mitigation algorithms with multi-modal models, we limited experiments to single-image classification. Segmentation, detection, and captioning could be further explored with task-specific fairness notions.

- **Fairness Notions**: As we stated in Section 2, there are also other fairness notions to quantify the model fairness, like individual fairness and counterfactual fairness, which could be used in future experiments.

- **Experiments**: We evaluated LVLMs in zero-shot only. Fine-tuning them could further reveal their fairness performance and bring more valuable insights. In the comparison, we divide the method into data-centric and algorithmic. However, combining all algorithmic approaches with data-centric approaches is a promising solution to achieve better fairness. To provide preliminary evidence, we evaluate the combination of RandAug (data-centric) with three algorithmic debiasing methods, and we report results in Table 22. On UTKFace, integrating RandAug with MCDP and GroupDRO improves both utility and fairness relative to ERM, achieving fairness-without-harm. However, this improvement does not consistently generalize; for example, on HAM10000 the same combinations do not yield similar benefits. These mixed outcomes suggest that pairing data-centric and algorithmic methods is a promising direction, but its effectiveness is dataset-dependent and requires broader validation. We consider a more systematic investigation of such combinations to be an important avenue for future work.

- **Analysis of LVLM Fairness**: The paper considers fairness differences across LVLMs with different architectural or alignment choices but does not account for confounding factors like differences in training data or finetuning protocols due to the lack of transparency and standardization in

recently released LVLMs. For example, the Gemma 3 Technical Report discloses only high-level information, such as the number of training tokens and the fact that both SFT and RLHF were used for alignment. Further details of implementation are unavailable, which prevents controlled comparisons. This lack of visibility is common in current LLM and LVLM releases, making some in-depth analysis difficult. As a result, our study intentionally focuses on aspects that are quantifiable and reproducible.

Table 22: Preliminary results of combining RandAug and algorithmic methods. (RA = RandAug)

| Dataset | Metric | ERM | GapReg | | MCDP | | GroupDRO | |
|---|---|---|---|---|---|---|---|---|
| | | | Base | +RA | Base | +RA | Base | +RA |
| UTKFace | ACC | $92.75 \pm 0.54$ | $92.53 \pm 0.88$ | $91.43 \pm 1.07$ | $92.49 \pm 0.72$ | $\mathbf{92.92} \pm 0.83$ | $92.45 \pm 0.39$ | $\mathbf{92.86} \pm 0.40$ |
| | Worst | $91.78 \pm 0.61$ | $91.70 \pm 0.95$ | $90.73 \pm 1.14$ | $91.63 \pm 1.14$ | $\mathbf{92.20} \pm 0.91$ | $91.41 \pm 0.62$ | $\mathbf{92.17} \pm 0.41$ |
| | Gap | $2.26 \pm 0.65$ | $\mathbf{1.91} \pm 0.59$ | $\mathbf{1.63} \pm 0.49$ | $\mathbf{2.00} \pm 0.99$ | $\mathbf{1.66} \pm 0.69$ | $2.44 \pm 0.99$ | $\mathbf{1.60} \pm 0.86$ |
| | EqOdd | $97.62 \pm 0.53$ | $\mathbf{98.10} \pm 0.61$ | $97.88 \pm 0.52$ | $98.04 \pm 0.97$ | $98.05 \pm 0.69$ | $97.06 \pm 0.63$ | $\mathbf{98.31} \pm 0.55$ |
| | DP | $94.55 \pm 1.20$ | $\mathbf{95.30} \pm 1.62$ | $\mathbf{95.82} \pm 1.55$ | $95.80 \pm 0.64$ | $96.42 \pm 1.68$ | $94.78 \pm 1.85$ | $\mathbf{95.15} \pm 0.98$ |
| HAM10000 | AUC | $88.35 \pm 1.83$ | $84.97 \pm 3.18$ | $85.55 \pm 2.27$ | $82.96 \pm 1.84$ | $87.33 \pm 2.12$ | $87.66 \pm 2.72$ | $87.08 \pm 1.85$ |
| | Worst | $84.68 \pm 2.02$ | $82.57 \pm 3.66$ | $81.60 \pm 2.42$ | $80.29 \pm 1.70$ | $84.35 \pm 1.67$ | $83.98 \pm 3.00$ | $83.96 \pm 2.42$ |
| | Gap | $4.11 \pm 2.08$ | $\mathbf{3.07} \pm 2.34$ | $4.85 \pm 2.48$ | $3.10 \pm 2.25$ | $3.35 \pm 2.39$ | $4.98 \pm 2.51$ | $3.37 \pm 2.55$ |
| | EqOdd | $88.17 \pm 3.10$ | $\mathbf{98.15} \pm 3.71$ | $\mathbf{96.98} \pm 1.24$ | $99.52 \pm 0.95$ | $92.84 \pm 5.89$ | $90.94 \pm 5.33$ | $87.70 \pm 5.66$ |
| | DP | $82.22 \pm 4.78$ | $\mathbf{96.74} \pm 6.52$ | $\mathbf{97.40} \pm 2.72$ | $99.58 \pm 0.85$ | $86.56 \pm 6.11$ | $\mathbf{83.86} \pm 5.70$ | $80.31 \pm 6.96$ |

## G  BROADER IMPACTS

NH-Fair provides a transparent and public benchmark that lets researchers and developers evaluate both single-modal and vision-language models under the same fairness metrics, motivating the community to build more fair machine learning models. However, the visibility also may lead to "benchmark gaming": a model can be fine-tuned to excel on the reported sub-groups while quietly marginalizing untested or intersectional minorities, posing potential downstream AI-safety concerns.

## H  LLM USAGE

In accordance with the ICLR policy, we disclose the use of large language models (LLMs) in preparing this paper. LLMs are also a part of the subjects of our research, and all experimental results are based on our own implementations and evaluations of these models. Separately, we used LLMs as general-purpose assistive tools for language editing, improving the clarity of writing. They were not used for research ideation, system design, or the generation or analysis of experimental results.

