# OpenReview forum: "Benchmarking Bias Mitigation Toward Fairness Without Harm from Vision to LVLMs"
_ICLR.cc/2026/Conference — ICLR 2026 Poster_

### Official Review · Reviewer_dLUc · 2025-10-29

**Soundness:** 4
**Presentation:** 3
**Contribution:** 2
**Rating:** 8
**Confidence:** 3

**Summary:**

In this paper, the authors created a big standardized benchmark called NH-fair. The main idea is to be able to compare debiasing methods against an ERM baseline. So they first tunned the base model (ERM) exhaustively using a DTO selection strategy, for over 10,000 A100 GPU hours, which proved that optimizer choice and pretrained wiehgts are crucial, fixing these choices leads to unfair comparisons. The results show that most specialized debiasing algorithms do not realiby outperform this ERM baseline. Instead, simple data augmentation (like RandAug) often achieves "Fairness without harm"(FWH), improving fairness without sacrificing utility. From their analysis, they revealed that huge LVLMs still exhibit significant subgroup disparities and simply scaling model size doesnt guarantee fairness.

**Strengths:**

I think the paper has a good originality as the NH-Fair unifies bias mitigation evaluation across tradicional supervised vision models and also LVLMs. It has a big significance as they introduce the FWH methodology. They did a big execution, solving the idea of insufficiently trained baseline problem as they spent over 10,000 A100 GPU hours with exhaustive hyperparameter optimization.

**Weaknesses:**

The paper skips theoretical fairness concepts, as they only focus on group parity metrics, omitting individual fairness and counterfactual fairness.
Future work should try to incorporate these for a wider view of bias.
Most part of data was about human faces, for future work needs to include more visual domains.
They did not fully use all sensitive attributes for evaluate intersectional groups (like gender vs race).
LVLMs were evaluated only with zero shot prediction mode (classification) even the paper suggesting finetuning.

**Questions:**

Data Augmentation is claimed to emerge as a promising practical strategy that "most often achieves fairness without utility loss" . Given that the FWH methodology allows for model selection in either the Optimal Zone or the Sub-Optimal Zone, could you quantify, across all seven datasets, the success rate of a method like RandAugment when classifying its best-found model into the Optimal Zone versus the Sub-Optimal Zone?

---

> ### Author Response · Authors · 2025-11-25
> **Reply to Reviewer dLUc**
>
> We thank the reviewer for the positive assessment of our paper. We address the raised weakness and question as follows.
>
> > Weaknesses: The paper skips theoretical fairness concepts, as they only focus on group parity metrics, omitting individual fairness and counterfactual fairness. Future work should try to incorporate these for a wider view of bias. Most part of data was about human faces, for future work needs to include more visual domains. They did not fully use all sensitive attributes for evaluate intersectional groups (like gender vs race). LVLMs were evaluated only with zero shot prediction mode (classification) even the paper suggesting finetuning.
>
> We agree that individual fairness and counterfactual fairness are important components of the broader fairness landscape, as acknowledged in our Limitations section. At the same time, these notions remain particularly challenging to evaluate in vision settings. Individual fairness requires a well-defined similarity metric between individuals, which is difficult to formulate for high-dimensional image data. Counterfactual fairness typically assumes access to a structural causal model, which demands additional domain knowledge that are rarely available for large-scale image datasets. Incorporating these fairness notions is a promising direction for future work, and we view our benchmark as a foundation that can be extended in these directions.
>
> Regarding intersectional fairness, we conducted preliminary experiments on UTKFace using intersectional groups during the rebuttal period. These results are presented in Appendix F Figure 12 (page 43). We found that accuracy disparities across intersectional groups (e.g., gender × race) can be larger than those seen under single-attribute groupings, but we also found that these gaps can be substantially reduced through appropriate hyperparameter tuning. This further reinforces one of our main messages: fairness evaluations are sensitive to model configuration choices, and fixed hyperparameters may yield misleading conclusions. Our benchmark is easy to extend to intersectional group studies and more visual domains.
>
> Finally, while our LVLM evaluations focus on a zero-shot setting, this choice isolates the effect of the pretrained model without introducing confounding factors from diverse fine-tuning pipelines (e.g., SFT, LoRA). As discussed in our Limitations section, incorporating fine-tuning into the benchmark is an important next step, and we expect the empirical insights from the zero-shot setting to provide useful guidance for selecting models and configurations for downstream fine-tuning.
>
> > Data Augmentation is claimed to emerge as a promising practical strategy that "most often achieves fairness without utility loss" . Given that the FWH methodology allows for model selection in either the Optimal Zone or the Sub-Optimal Zone, could you quantify, across all seven datasets, the success rate of a method like RandAugment when classifying its best-found model into the Optimal Zone versus the Sub-Optimal Zone?
>
> In our framework, a method is placed in the Optimal Zone if both subgroup accuracies are at least as high as those of the DTO-selected ERM baseline. If no such configuration exists, we place it in the Sub-Optimal Zone, where the disadvantaged group improves while the advantaged group drops.
>
> Across all seven datasets, RandAugment’s FWH-selected models fall in the Optimal Zone during the selection phase on the validation set. Moreover, the corresponding test-set results also satisfy the Optimal Zone criteria, meaning both subgroup accuracies exceed those of the ERM baseline on test data. We have updated Table 2 to report (i) the zone in which each method’s selected model lies, and (ii) whether the selection results on validation match the final results on test.
>
> Finally, we note that mismatches between validation and test zones can occur for some methods. This is an inherent nature of most model selection methods that rely on validation results for selection, since data distribution may change between validation and test.

---

> > ### Comment · Reviewer_dLUc · 2025-11-25
> >
> > Thanks for the authors for clarifying those points. I have increased both my confidence score and my "contribution score" to reflect the rebuttal.

---

### Official Review · Reviewer_ZcM2 · 2025-10-30

**Soundness:** 2
**Presentation:** 3
**Contribution:** 2
**Rating:** 6
**Confidence:** 3

**Summary:**

This paper introduces NH-Fair, a benchmark for "Fairness Without Harm" (FWH). Its core contribution is demonstrating that a rigorously tuned ERM baseline (using a novel DTO/FWH selection) often outperforms complex bias mitigation methods. The benchmark is extended to LVLMs, finding they still suffer from bias and that scaling does not automatically fix it.

**Strengths:**

1.Practical Problem: The "Fairness Without Harm" (FWH) principle is highly relevant for real-world deployment (e.g., healthcare).

2.Strong Baseline: The paper's key finding—that a well-tuned ERM can beat specialized algorithms—is a crucial, critical finding for the field, backed by extensive HPO.

3.Rigorous Protocol: The DTO/FWH model selection strategy provides a novel and fair method for comparing models.

4.Timely LVLM Analysis: The paper correctly identifies that LVLMs are not a panacea for fairness, a critical and timely observation.

**Weaknesses:**

1.Limited LVLM Scope: The LVLM evaluation is zero-shot only. This is a major gap, as models are typically fine-tuned, a process which could significantly alter fairness outcomes.

2.No Intersectional Analysis: The benchmark only considers single sensitive attributes, ignoring intersectional biases (e.g., race and gender), which can be more severe.

3.Lacks Deep Insight: The paper is excellent at observing (e.g., "RandAug works well" , "Optimizers matter" ), but provides little analysis as to why these phenomena occur.

**Questions:**

1.LVLM Fine-tuning: How do you expect your zero-shot LVLM findings to generalize to the more common fine-tuning setting?

2.HPO Cost: Your finding relies on 10k+ GPU-hours. What is the "budget-friendly" HPO recommendation for a practitioner who cannot afford this?

3.Waterbirds: You note Waterbirds is a spurious correlation dataset, not a social bias one. Why include it in a social fairness benchmark if it's "easier to resolve" and may dilute the main message?

---

> ### Author Response · Authors · 2025-11-25
> **Reply to Reviewer ZcM2 (1/2)**
>
> We thank the reviewer for the positive and constructive feedback of our paper. We address all concerns below.
>
> > W1 & Q1: Fine-tuning setting concern and generalization
>
> We agree that fine‑tuned LVLMs are important in practice. As noted in Appendix F (Limitations and Future Directions), "*Fine-tuning them could further reveal their fairness performance and bring more valuable insights.*" In this work, we start from the zero‑shot setting because it (i) isolates the effect of the pretrained model without introducing confounding factors from many possible fine‑tuning pipelines (e.g., SFT, LoRA), and (ii) reflects a common usage pattern where LVLMs are deployed off‑the‑shelf or with very light adaptation. We view these zero‑shot findings as a first step toward understanding fairness in more common fine‑tuning setting.
>
> Our study covers four widely used LVLM families across multiple model sizes. Models showing better utility–fairness trade‑offs in the zero‑shot setting are natural candidates for fine‑tuning, and we expect them to require smaller updates to achieve both good utility and fairness. Additionally, our results indicate that larger models are not always necessary when fairness is a primary objective under limited compute budgets; practitioners can often start from smaller models to reduce cost.
>
> > W2 No Intersectional Analysis: The benchmark only considers single sensitive attributes, ignoring intersectional biases (e.g., race and gender), which can be more severe.
>
> We agree that intersectional bias is an important and challenging aspect of fairness. As we also noted in Appendix F (Limitations and Future Directions), extending our evaluation to intersectional groups is a valuable direction for future work. In our current benchmark, only a few datasets provide multiple demographic annotations to support intersectional analysis. For those cases, forming intersectional groups may lead to very small per-group sample sizes, resulting in unstable or noisy disparity estimates.
>
> Nevertheless, during the rebuttal period, we conducted additional experiments on UTKFace using intersectional groups (e.g., combining race and age). We observed that accuracy gaps across intersectional groups can indeed be larger than those observed under single-attribute groupings. Importantly, we also found that these disparities can still be significantly mitigated through careful hyperparameter tuning. We have added Figure 12 to Appendix F (page 43), highlighted in magenta, showing that an appropriate choice of hyperparameters can reduce the accuracy gap from over 30\% to below 5\%. These results also support one of our main messages: using fixed hyperparameters across baselines or datasets can lead to misleading or biased conclusions.
>
> > W3 Lacks Deep Insight: The paper is excellent at observing (e.g., "RandAug works well" , "Optimizers matter" ), but provides little analysis as to why these phenomena occur.
>
> The goal of this work is to establish a reliable empirical foundation for fairness evaluation across models, datasets, and modalities, and to surface under-explored factors (e.g., training strategies, optimizer choice, augmentation, and model size) that substantially influence fairness outcomes but are often overlooked in prior work. Our design of these experiments is grounded in practical observations from the fairness literature. For example, many prior studies fix a single optimizer or learning-rate schedule across all baselines and datasets, without justification or tuning. Our results provide concrete empirical evidence that such choices can meaningfully alter fairness metrics and may unintentionally bias comparisons. Similarly, by examining commonly used spurious-correlation datasets such as Waterbirds, we show that these benchmarks can make fairness mitigation appear easier than it is on more realistic datasets, highlighting the importance of dataset selection when drawing fairness conclusions. These findings are valuable and may help encourage more equitable, transparent, and methodologically sound evaluation practices. We agree that developing deeper principled or theoretical explanations is important, and we believe our benchmark and empirical observations provide a foundation for such future investigations.

---

> ### Author Response · Authors · 2025-11-25
> **Reply to Reviewer ZcM2 (2/2)**
>
> > Q2 Your finding relies on 10k+ GPU-hours. What is the ‘budget-friendly’ HPO recommendation for a practitioner who cannot afford this?
>
> A key contribution of Sec. 4.1 and Appendix C.1 is precisely a budget-friendly HPO recipe. Our results show that practitioners may obtain strong fairness–utility trade-offs by focusing on just a few impactful strategies: (1) start from strong pretrained weights; (2) tune the optimizer and its learning rate, which has the largest effect on both utility and fairness; and (3) decide whether to apply data augmentation such as RandAug. This summary is included in the Introduction section for easier reference.
>
> > Q3.Waterbirds: You note Waterbirds is a spurious correlation dataset, not a social bias one. Why include it in a social fairness benchmark if it's "easier to resolve" and may dilute the main message?
>
> We include Waterbirds for two reasons. First, despite not being a socially meaningful fairness dataset, it is widely used in the fairness and bias-mitigation literature. Including it allows consistency and comparability with prior work. Second, we intentionally use Waterbirds to illustrate that spurious-correlation datasets can be substantially easier than real fairness datasets, which may lead to overly optimistic conclusions about fairness methods. This aligns with our discussion in Sec. 4.2 ("Rethinking spurious correlation datasets for fairness evaluations"), where we emphasize that such datasets should be used with caution when drawing broader fairness insights.

---

### Official Review · Reviewer_ByuY · 2025-10-31

**Soundness:** 3
**Presentation:** 3
**Contribution:** 2
**Rating:** 4
**Confidence:** 4

**Summary:**

This paper presents a new benchmark for bias mitigation methods across 7 image classification tasks. It benchmarks traditional mitigation methods with a ResNet18 as well as vision-language models like CLIP and LLMs. It also includes analyses on the impact of hyper-parameters on model fairness.

**Strengths:**

S1: The authors conduct a systematic hyper-parameter optimisation and model selection protocol, a step which is often overlooked in bias mitigation benchmarks.

S2: The authors present some unique analyses which are not usually discussed in bias mitigation papers, for instance on the impact of hyper-parameter choice on fairness metrics (e.g., choice of optimiser appears more impactful than choice of weight decay), whether pre-training is done (I liked the plots in the Appendix Fig 5), and the size of LLMs.

S3: They conduct extensive experiments and include a range of model types.

S4: I appreciated this insightful point: "Over-reliance on datasets like Waterbirds may therefore underestimate the difficulty of fairness challenges and overstate algorithmic effectiveness."

**Weaknesses:**

**Major weaknesses**

W1: My primary criticism to the paper is that I am not sure what the field needs is another benchmark suggesting that overall ERM performs better/more reliably than existing mitigation methods (I would say there is already a broad consensus on this). I would argue that actually, instead of having an aggregate benchmark where many methods/models are compared across different datasets and metrics, it would make more sense to do a tailored analysis looking at which mitigation methods work when. This is I believe in line with current research in the field [1,2,3,4] which suggest that it is important to understand what the cause of bias is, what type of bias it is, what fairness metric you want to optimise for, in order to determine if you should conduct mitigation and with what method. Similarly to this point, I think looking at so many fairness metrics at once is also counter-productive, as you will only really care about optimising one metric for a given setting (for instance there are very few settings where it would make sense to optimise for demographic parity) [5].

W2: I think the statement that "the scaling law" does not apply to AI fairness is over-stated given that the authors have only conducted a couple experiments on classification.

W3: The authors say they "suggest not confusing fairness datasets with domain generalization dtatasets (e.g., Waterbirds, Colored MNIST). While DG datasets probe robustness to distribution shifts, they do not contain socially meaningful sensitive attributes". While I agree (as mentioned in S4) that it is important to not just evaluate mitigation methods on over-simplistic spurious correlation benchmarks like Waterbirds, I disagree that there is an inherent difference between domain generalisation and fairness datasets. I think what matters for bias mitigation methods is just to have some coherent grouping of data (and that can be based on demographic attributes or on other groupings, like where the image was taken, what device was used etc.). You can have super "simple" socially meaningful groups (e.g., skin tone in skin lesion) the same way you could have very complex “domains" (e.g., geographic location of a sample) so I don't believe there is a hard difference for the mitigation algorithms. Furthermore, I don't think the authors can argue about using socially meaningful groups when they are using very un-meaningful classification targets for some of their experiments (e.g., whether someone has wavy hair in CelebA)!.

W4: I generally think there are a lot of tables that are hard to interpret. Many also do not have boldings or standard deviations and inconsistent numbers of significant decimal places. This gives the manuscript an unfinished aspect. I would recommend that the authors try to make some more of the tables into plots (for instance Table 15 on LLM size). I would also recommend that they vary the interpretations, for example when you say that optimizer choice matters, instead of just listing all the different metric values for different datasets and optimiser it would be helpful to include some summary stats, e.g., overall accuracy varies by a max of this much, or accuracy gap can increase by 20%.

W5: It would be interesting for the authors to check other LLM tasks like fairness of generated outputs.

W6: In fig3 it would be helpful to show overall accuracy as well because we don’t know whether the models are harming overall performance or not. The authors could also include some kind of statistical testing to see if there are any significant differences relative to ERM.

W7: One of your datasets fairface has no gap (0.86)! What’s the point of including it in the fairness analysis?

W8: Small concern that gender and race classification tasks may be problematic.

W9: There are lots of missing details on the way the mitigation methods work, even in the appendix. This is particularly true for bias mimicking, FIS, DFR (should specify that you are retraining the last layer on a balanced distribution! - right?), CLIP/BLIP models and for related post-training debiasing methods!

W10: It would be helpful to show variance of results for each method across hyper-parameters.

**Minor weaknesses**

W11: Min max fairness definition shouldn't have the min as you are presenting the metrics not the way they're optimised (to keep it consistent with the other fairness definitions).

W12: All the left quotes are wrong. Please use `` for left quotes and '' for right quotes in LaTeX!

W13: Table 1 columns are a bit confusing. I would just put the proportion of the targets.

W14: Colouring the cells in table 3 to give an indication of better worse metrics would be helpful for the reader.

W15: The wording in the conclusion “the fairness issue remains unresolved." is overly simplified and referring to "the fairness issue" feels imprecise, as fairness is so multi-faceted.

W16: What does this wording mean in section B1 “Without additional illustration”?

W17: It would be helpful to add a note on which methods use demographic information.

W18: Undefined reference line 1114.

W19: Table 15: you can also get an SD for the average metrics.

[1] Rethinking Fair Representation Learning for Performance-Sensitive Tasks, ICLR 2024.
[2] Change is Hard: A Closer Look at Subpopulation Shift, ICML 2023.
[3] Mind the Graph When Balancing Data for Fairness or Robustness, NeurIPS 2024.
[4] Automatic dataset shift identification to support safe deployment of medical imaging AI, MICCAI 2025.
[5] Critical Appraisal of Fairness Metrics in Clinical Predictive AI, arXiv 2025.

**Questions:**

Q1: Why did the authors select the datasets they did? Why did they include two medical datasets of the same modality (skin images) instead of considering other medical modalities?

Q2: Are you comparing two model selection methods, or just applying DTO to ERM and then doing fairness without harm for the rest? Later you say “best model selected under DTO or FWH criteria”.

Q3: Why did the authors not include post-processing methods? these are often suggested to be the most useful in practice, as suggested here [1].

Q4: Why do you use resnet18 as a backbone?

Q5: Why do you only look at group x class grouping for resampling?

Q6: Is there any literature on optimisers and fairness? Can you explain your results? Similarly do you have a hypothesis for the effect of batch size?

Q7: Why do you think there is so much prompt sensitivity - this seems like a significant weakness? Also for that table 17, it would be helpful to summarise the variation in results as SD or mean difference in outputs.

Q8: How do you obtain SDs? Average over random seeds?

[1] Oxonfair, NeurIPS 2024.

---

> ### Author Response · Authors · 2025-11-25
> **Reply to Reviewer ByuY (1/5)**
>
> We appreciate the constructive and detailed feedback. All revisions made in response to your comments have been marked in blue.
>
> > W1: My primary criticism to the paper is that I am not sure what the field needs is another benchmark suggesting that overall ERM performs better/more reliably than existing mitigation methods (I would say there is already a broad consensus on this). I would argue that actually, instead of having an aggregate benchmark where many methods/models are compared across different datasets and metrics, it would make more sense to do a tailored analysis looking at which mitigation methods work when. This is I believe in line with current research in the field [1,2,3,4] which suggest that it is important to understand what the cause of bias is, what type of bias it is, what fairness metric you want to optimise for, in order to determine if you should conduct mitigation and with what method. Similarly to this point, I think looking at so many fairness metrics at once is also counter-productive, as you will only really care about optimising one metric for a given setting [5].
>
> We agree that there is a consensus that ERM performs better/comparable with many mitigation methods. This paper is not to claim a contribution here, and therefore, we devote only a small portion of Sec. 4.2 to this point. Rather, NH-Fair introduces several contributions that address concrete limitations of existing benchmarks such as MEDFAIR, FFB, and ABCFair:
>
> - **Two-stage selection for fairness without harm**: The no-harm constraint is critical in many real-world systems but is largely absent from prior benchmarks. We consider a two-stage model selection strategy, including DTO and FWH, to identify those methods that can improve both fairness and utility. And we identified that data augmentation is a simple yet effective path to fairness without harm.
> - **What makes ERM strong?**: Instead of simply claiming ERM performs better than others, we ask which training details have greater impacts on utility and fairness, yielding actionable insights for model developers and helping them reduce the time and cost of hyperparameter optimization (e.g., Ray Tune, Optuna) by narrowing the search space, especially in industry settings. In addition, we found that some of the prior fairness papers and benchmarks may use a fixed optimizer and learning rate for all datasets, which leads to unfair comparisons. Our findings underscore the need for more equitable and transparent evaluation practices in future research.
> - **Unified benchmarking across single and multi-modal models**: NH-Fair enables standardized fairness evaluation across both vision models and LVLMs. This allows for cross-modality comparison, providing practitioners with insights into trade-offs between model types. We found that larger models do not guarantee improved fairness, and that smaller models can achieve similar fairness–utility performance, offering practical implications for reducing model serving costs.
>
> We fully agree with the reviewer that tailored analyses of “which mitigation method works when...” are extremely valuable, and we have included discussion of [1-4] in the revised related work (Appendix A.4). We also see connections between this line of work and our motivations. For example, [2] raises concerns about current evaluation practices and questions the applicability of fair representation learning methods; similarly, our work highlights that using identical hyperparameters across datasets in prior evaluations can lead to misleading conclusions.  Our work is complementary to these works: we aim to provide a comprehensive benchmark for **fairness without harm** under standardized protocols, addressing a different but equally important aspect of real-world evaluation and deployment.
>
> Regarding the reviewer’s point about tailored diagnosis before mitigation: in many real application scenarios, especially under rapid production cycles, practitioners often do not have the capacity to fully diagnose the causal source or type of bias (and causal graphs are often unavailable). A common workflow is: (i) begin with a strong ERM model, (ii) select a small set of promising mitigation methods, and (iii) tune hyperparameters under limited budgets. NH-Fair is explicitly designed to support this workflow by quantifying how recent fairness algorithms behave, highlighting simple but effective strategies (e.g., RandAug) that often improve utility and fairness, and helping reduce the search space.
>
> Regarding the choice and number of fairness metrics: we agree that in a specific deployment, one typically optimizes a single metric relevant to that context. However, a benchmark must be comprehensive to serve the broader research community. While our FWH framework prioritizes accuracy (to prevent harm), reporting a diverse set of metrics allows researchers to see a complete picture of the trade-off surface, not to suggest they should all be optimized simultaneously.

---

> ### Author Response · Authors · 2025-11-25
> **Reply to Reviewer ByuY (2/5)**
>
> > W2: I think the statement that "the scaling law" does not apply to AI fairness is over-stated given that the authors have only conducted a couple experiments on classification.
>
> We thank the reviewer for pointing out that our statement about scaling laws and fairness was phrased too strongly. To better align the paper with this evidence, we have softened our claims about the scaling law in Sec. 1 and Sec. 4.3. We have also added new experiments on open-ended generation tasks in the revised Appendix C.3.3 (page 40). We evaluated LVLMs (LLaVA, Qwen, Gemma, Llama) on UTKFace, FairFace, and Waterbirds using open-ended prompts (e.g., “Describe this person's ethnic and cultural background...”) rather than multiple-choice constraints. We then mapped the generated text back to label spaces to calculate utility and fairness metrics. Consistent with our closed-set results, increasing model size does not guarantee improved fairness.
>
> > W3 The authors say they "suggest not confusing fairness datasets with domain generalization dtatasets (e.g., Waterbirds, Colored MNIST). While DG datasets probe robustness to distribution shifts, they do not contain socially meaningful sensitive attributes". While I agree (as mentioned in S4) that it is important to not just evaluate mitigation methods on over-simplistic spurious correlation benchmarks like Waterbirds, I disagree that there is an inherent difference between domain generalisation and fairness datasets. I think what matters for bias mitigation methods is just to have some coherent grouping of data (and that can be based on demographic attributes or on other groupings, like where the image was taken, what device was used etc.). You can have super "simple" socially meaningful groups (e.g., skin tone in skin lesion) the same way you could have very complex “domains” (e.g., geographic location of a sample) so I don't believe there is a hard difference for the mitigation algorithms. Furthermore, I don't think the authors can argue about using socially meaningful groups when they are using very un-meaningful classification targets for some of their experiments (e.g., whether someone has wavy hair in CelebA)!.
>
> We agree that, from a technical standpoint, there is no inherent difference between domain generalization (DG) and fairness datasets: most mitigation methods only require group labels, whether these arise from simple demographic attributes (e.g., gender, skin type) or more complex factors such as device or geographic location.
>
> Our discussion in Sec. 3.1 was intended primarily as a conceptual distinction for readers from social-science domains, to avoid conflating DG datasets (which probe robustness to distribution shifts) with fairness evaluation, and to caution against over-simplistic datasets like Waterbirds. From a fairness evaluation perspective, the key requirement is coherent and well-justified groupings, regardless of whether the groups are socially meaningful or derived from other factors. We have updated Sec. 3.1 (page 4) and Sec 4.2 (page 9) to clarify this.
>
> Regarding CelebA, while some targets (e.g., wavy hair) may appear less socially meaningful, our focus is on group attributes (e.g., gender) that have social relevance, and we maintain consistency with prior studies to ensure comparability.
>
> > W4 I generally think there are a lot of tables that are hard to interpret. Many also do not have boldings or standard deviations and inconsistent numbers of significant decimal places. This gives the manuscript an unfinished aspect. I would recommend that the authors try to make some more of the tables into plots (for instance Table 15 on LLM size). I would also recommend that they vary the interpretations, for example when you say that optimizer choice matters, instead of just listing all the different metric values for different datasets and optimiser it would be helpful to include some summary stats, e.g., overall accuracy varies by a max of this much, or accuracy gap can increase by 20\%.
>
> We thank the reviewer for the suggestions. We have updated the manuscript to ensure consistent numeric precision, bolded the best values per dataset/method across tables, and added summary statistics to highlight key trends. We also added visualization plots (Fig.9 on page 31 and Fig.10 on page 34) to improve clarity and interpretation. Table 15 on LLM size was presented as a radar plot in Fig.8 in the original submission.
>
> We have also added standard deviations for mean values of hyperparameter choices in Appendix C.1.6 (page 33) to illustrate variability and highlight the impact of different choices. Additionally, to maintain readability, we did not include standard deviations for all tables in the main text, but they are available in Appendix E (page 42).

---

> ### Author Response · Authors · 2025-11-25
> **Reply to Reviewer ByuY (3/5)**
>
> > W5: It would be interesting for the authors to check other LLM tasks like fairness of generated outputs.
>
> We have added LVLM generation tasks using open-ended prompts for reference in the Appendix C.3.3 (page 40). Since the main focus of this paper is on vision domains, we did not incorporate pure text tasks, but the added results provide initial insights into LLM-generated outputs and fairness.
>
> > W6: In fig3 it would be helpful to show overall accuracy as well because we don’t know whether the models are harming overall performance or not. The authors could also include some kind of statistical testing to see if there are any significant differences relative to ERM.
>
> We have updated Figure 3 to include overall accuracy. Additionally, we have added statistical testing in Sec. 4.2 (page 8). The resulting critical difference plots indicate that there is no significant difference relative to ERM.
>
> > W7: One of your datasets fairface has no gap (0.86)! What’s the point of including it in the fairness analysis?
>
> A small gap for a particular model does not imply that the dataset is inherently “near-fair.” On FairFace, other models (such as BLIP2, Llama4-Scout) can exhibit larger disparities. Moreover, while the disparities appear small in some cases, this occurs at the price of reducing accuracy for all groups. Including FairFace therefore provides a useful reference point for understanding how different methods trade off utility and fairness.
>
> > W8: Small concern that gender and race classification tasks may be problematic.
>
> We understand and appreciate the reviewer’s ethical concerns. Some datasets provide only limited demographic labels, and our intention is not to promote or endorse demographic classification for real-world deployment. These tasks are included solely to enable controlled academic analysis of fairness interventions. We have revised the Ethics Statement (page 11) to explicitly clarify this point and to emphasize that these models are not intended for operational use.
>
> > W9: There are lots of missing details on the way the mitigation methods work, even in the appendix. This is particularly true for bias mimicking, FIS, DFR (should specify that you are retraining the last layer on a balanced distribution! - right?), CLIP/BLIP models and for related post-training debiasing methods!
>
> We introduced each method in two places: Appendix B.3 (method introduction) and Appendix B.4 (implementation details). Appendix B.3 introduces the background for each method (e.g., “DFR is a post-processing method that retrains only the final layer...”), while Appendix B.4 focuses on adaptations we made when a method’s pipeline differs from its original implementation (e.g., how we adapt FIS to our setting). For methods where we rely on their official implementations, we had previously omitted some details.
>
> If we understand the reviewer correctly, you would like more details on how each method mitigates bias, not only how we implement it. Therefore, in the revised version, we have expanded Appendix B.3 (page 21-23) to include more detailed descriptions of bias mitigation methods.
>
> > W10: It would be helpful to show variance of results for each method across hyper-parameters.
>
> We appreciate the suggestion. However, visualizing the full variance across hyperparameters for every method–dataset–metric combination would require well over 100 additional figures, which would make the manuscript extremely long and difficult to navigate. Instead, we believe a clearer and more practical approach is to release these detailed per–hyper-parameter results and variance visualizations in our public code repository upon acceptance, where they can be thoroughly organized and easily explored. This allows readers to inspect the full distributions without compromising the readability of the paper.
>
> > W11 Min max fairness definition; W12 left quotes; W18: Undefined reference; W19: Table 15
>
> Thanks for pointing them out. We have fixed them in the revision.
>
> > W13: Table 1 columns are a bit confusing. I would just put the proportion of the targets.
>
> In Table 1, we chose to report the proportion of sensitive attributes because group imbalance itself can be an important source of bias and is therefore directly relevant for fairness evaluation.
>
> > W14: Colouring the cells in table 3 to give an indication of better worse metrics would be helpful for the reader.
>
> We have updated Table 3 by bolding the values to visually highlight the best-performing metrics.
>
> > W15: The wording in the conclusion “the fairness issue remains unresolved.” is overly simplified and referring to “the fairness issue” feels imprecise, as fairness is so multi-faceted.
>
> We have revised the phrase with: “AI fairness issues remain challenging in computer vision domains, even as new methods are proposed and model capacity continues to increase.”

---

> ### Author Response · Authors · 2025-11-25
> **Reply to Reviewer ByuY (4/5)**
>
> > W16: What does this wording mean in section B1 “Without additional illustration”?
>
> The phrase was intended to mean “unless otherwise specified,” since certain datasets (e.g., Waterbirds) provide predefined splits. We have replaced it with clearer wording: “Unless otherwise noted, we randomly split each dataset...” to avoid confusion.
>
> > W17: It would be helpful to add a note on which methods use demographic information.
>
> We have added explicit notes in Appendix B.3 (page 23) indicating whether demographic attributes are required.
>
> > Q1: Why did the authors select the datasets they did? Why did they include two medical datasets of the same modality (skin images) instead of considering other medical modality?
>
> We selected datasets that (1) are widely used in prior studies; (2) contain explicit demographic annotations so that group fairness metrics are well-defined; and (3) have sufficient data to avoid unstable evaluation from random splits. Early on, we considered other medical datasets like the fundus imaging dataset PAPILA, which contains only 420 images, making results extremely sensitive to random splits. We also explored chest X-ray images from MIMIC-IV; however, these images are challenging for zero-shot VLMs and would require domain-specific models such as MedCLIP. Considering these factors, we finalized our benchmark with two skin image datasets to ensure stable, meaningful, and comparable evaluations.
>
> > Q2 Are you comparing two model selection methods, or just applying DTO to ERM and then doing fairness without harm for the rest? Later you say “best model selected under DTO or FWH criteria”.
>
> We are not comparing two model selection methods against each other. DTO is applied only once to select a strong ERM baseline among hyperparameter configurations. Given this selected ERM, FWH is then used to select the best models for other mitigation methods. We have rephrased the original sentence “with the best model selected under DTO or FWH criteria” to: “with the best ERM model selected under DTO criteria and the best models for mitigation methods selected under FWH criteria.”
>
> > Q3: Why did the authors not include post-processing methods? these are often suggested to be the most useful in practice, as suggested here [1].
>
> We included post‑processing in two places: DFR for supervised models and FairerCLIP/SFID as post‑hoc debiasers for CLIP. Following your suggestion, we have added OxonFair as a new baseline in the revision.  OxonFair provides multiple optimization targets and constraints; we also run hyperparameter searching on those metrics relevant to our tasks.
>
> > Q4: Why do you use resnet18 as a backbone?
>
> We found that deeper networks do not always yield better utility or fairness. Using ResNet-18 allows us to allocate our GPU budget to more methods and hyperparameter sweeps.
>
> > Q5: Why do you only look at group x class grouping for resampling?
>
> Our resampling actually searched over both “group” and “group × class” strategies, as indicated in the hyperparameter search space (Table 7).
>
> > Q6: Is there any literature on optimisers and fairness? Can you explain your results? Similarly do you have a hypothesis for the effect of batch size?
>
> There is limited prior work directly analyzing how optimizer choice affects group fairness. However, a recent study, “Some Optimizers are More Equal: Understanding the Role of Optimizers in Group Fairness” (NeurIPS 2025), provides evidence that the choice of optimizer can meaningfully influence fairness outcomes. Their findings, that adaptive optimizers tend to yield better group-fairness properties than SGD, are consistent with our observations across multiple settings.
>
> Our experiments similarly show that adaptive optimizers can lead to models with both higher utility and lower disparity simultaneously in multiple datasets. One potential explanation is that adaptive methods more effectively smooth the loss landscape across subgroups, reducing sensitivity to group-specific sharp minima that SGD may fall into.
>
> Regarding the batch-size effect, we hypothesize that batch size influences fairness through its impact on gradient noise and subgroup representation. Smaller batches introduce higher stochasticity, which can cause greater variance in fairness outcomes; larger batches produce more stable gradients but may over-represent majority-group patterns in each update, thereby amplifying existing biases.

---

> ### Author Response · Authors · 2025-11-25
> **Reply to Reviewer ByuY (5/5)**
>
> > Q7: Why do you think there is so much prompt sensitivity - this seems like a significant weakness? Also for that table 17, it would be helpful to summarise the variation in results as SD or mean difference in outputs.
>
> Prompt sensitivity is a weakness of existing LVLMs, and the performance of LVLMs can vary substantially depending on prompt design. Indeed, several prior studies (e.g., [1]) have specifically analyzed this sensitivity. In our experiments, since the prompts were primarily hand-crafted, we tested multiple variants and selected the one achieving the best overall performance for the final evaluations to ensure fair comparisons across methods.
>
> [1] PARC: A Quantitative Framework Uncovering the Symmetries within Vision Language Models. CVPR25
>
> > Q8: How do you obtain SDs? Average over random seeds?
>
> Yes, they come from different random seeds.

---

> > ### Comment · Reviewer_ByuY · 2025-11-26
> >
> > Thank you for your extensive efforts on the rebuttal, I have read through your responses and changes. I appreciate the manuscript revisions, particularly on the wording, the presentation of their results, and additional clarifications in the appendix.
> >
> > I still believe that this paper is not in the direction we should be going into as a field (e.g., tailored analysis of which methods are appropriate for a given type of bias/data/model/desired outcome), however I concede this may be a personal opinion (and I appreciate the extra paragraph in A4), so I have increased my score to a weak accept.

---

### Official Review · Reviewer_RqaZ · 2025-11-03

**Soundness:** 3
**Presentation:** 2
**Contribution:** 2
**Rating:** 4
**Confidence:** 4

**Summary:**

The paper presents a comprehensive study of fairness-utility trade-off in modern vision and multimodal foundation models. The authors create a new benchmark consisting of various domains and tasks, a systematic model selection protocol, and a set of group fairness and utility metrics. This benchmark is used to evaluate and compare SoTA models, training strategies and debiasing methods to explore the optimal recipe for accurate and fair models. Experiments show that a well-optimized baseline with large-scale pretraining and data augmentation matches or beats sophisticated mitigation methods in fairness while maintaining performance on utility tasks.

**Strengths:**

- This work addresses several shortcomings in fairness evaluation in prior work, such as inconsistent hyperparameter selection, overlooked utility performance and not including pre-trained foundation models.
- The evaluation framework is systematic, consisting of multiple datasets and domains, principled training and model selection, and numerous utility and bias metrics. This is crucial to ensure fair comparison between methods as well as reproducibility.
- Extensive ablation experiments on training practices such as model size, batch size and weight decay, as well as evaluation of VLMs putting them in the same playing field with dedicated classification models.

**Weaknesses:**

- The proposed benchmark is limited to classification task. Despite being reformulated for image-text matching (CLIP) and generative (LVLM) models, it does not truly address utility and fairness beyond closed-set predictions. I would have liked to see open-set tasks like free-form image-text retrieval and open-ended VQA, captioning or reasoning, as these are the tasks where vision(-language) foundation models truly overtake task-specific vision models. This would also enable the holistic evaluation of VLM debiasing methods, which also tend to overlook utility performance while optimizing for fairness.
- Even in classification, I am slightly concerned that the performance is already saturating for most models and datasets, with group accuracy gap under 5% and DP, EqOdd over 95%. I am not sure how perceivable these disparities are in practice. It might also be helpful to construct a "fairness-hard" subset with more challenging datasets/subtasks (e.g., more attributes in CelebA correlated with either gender; only "wavy hair" is used in the current version).
- In general, I feel that the conclusions from the experiments confirm anecdotal observations bias mitigation (e.g., that hyperparameter tuning and model selection make a big difference), but not groundbreaking in context of existing fairness benchmarks (e.g., FFB https://arxiv.org/abs/2306.09468).
- Certain conclusions in hyperparameter choice are qualitative (e.g., optimizer choice and learning rate are more important than other hyperparameters) and could have benefitted from a quantified sensitivity metric.

**Questions:**

See weakness section. In addition, I wonder if combining data curation and algorithmic mitigation can produce the best trade-off between utility and accuracy? Should future research explore the intersection between both, or prioritize data-centric methods as suggested in the paper?

---

> ### Author Response · Authors · 2025-11-25
> **Reply to Reviewer RqaZ (1/3)**
>
> We appreciate the reviewer for the insightful comments. We have highlighted the revisions made in response to your comments in orange. We address your concerns below.
>
> > The proposed benchmark is limited to classification task. Despite being reformulated for image-text matching (CLIP) and generative (LVLM) models, it does not truly address utility and fairness beyond closed-set predictions. I would have liked to see open-set tasks like free-form image-text retrieval and open-ended VQA, captioning or reasoning, as these are the tasks where vision(-language) foundation models truly overtake task-specific vision models. This would also enable the holistic evaluation of VLM debiasing methods, which also tend to overlook utility performance while optimizing for fairness.
>
> We thank the reviewer for this valuable suggestion. To address this concern, we have expanded our benchmark to include open-ended generation tasks in the revised Appendix C.3.3 (page 40). We evaluated LVLMs (LLaVA, Qwen, Gemma, Llama) on UTKFace, FairFace, and Waterbirds using open-ended prompts (e.g., “Describe this person's ethnic and cultural background...") rather than multiple-choice constraints. We then mapped the generated text back to label spaces to calculate  utility and fairness metrics. Consistent with our closed-set results, increasing model size does not guarantee improved fairness. In addition, open-ended generation reveals an additional phenomenon not observable in classification settings: aggressive safety alignment can suppress model willingness to describe sensitive attributes. For example, LLaVA and Llama models frequently refuse to output descriptions on FairFace, resulting in a large fraction of unusable generations and reduced evaluability. This highlights that open-set tasks can introduce new fairness challenges, where alignment strategies, not model capability, limit both utility and the feasibility of fairness assessment.
>
> > Even in classification, I am slightly concerned that the performance is already saturating for most models and datasets, with group accuracy gap under 5% and DP, EqOdd over 95%. I am not sure how perceivable these disparities are in practice. It might also be helpful to construct a “fairness-hard" subset with more challenging datasets/subtasks (e.g., more attributes in CelebA correlated with either gender; only “wavy hair" is used in the current version).
>
> In our benchmark, the seemingly “saturated” performance is primarily a consequence of effective hyperparameter optimization (HPO) and the novel model selection strategy.  Several of our classification tasks have been identified as fairness-hard in prior work, yet extensive HPO substantially reduces the observed disparities. For example: 1) on HAM10000 (Age), MedFair benchmark reports ~14\% AUC gaps for ERM, while under our protocol, the gap decreases to ~4\% while maintaining similar utility; 2) on CelebA (Wavy Hair/Gender), prior benchmarks such as FFB find that ERM exhibits large EqOdd discrepancies, while our tuned ERM achieves both higher ACC and significantly lower EqOdd.
>
> Regarding the reviewer’s suggestion of constructing more challenging “fairness-hard” subsets: We agree this is valuable. When selecting dataset configurations, we evaluated multiple candidate targets and sensitive attributes and prioritized those with larger disparities in preliminary experiments and prior benchmarks, rather than selecting attributes arbitrarily. As noted in Sec. 3.1,  “we focused on attributes that exhibited the clear disparity in model predictions".
>
> During the rebuttal period, we further evaluated intersectional groups (Race × Age on UTKFace) and observed that disparities can indeed be larger than in single-attribute settings. However, we also found that these gaps can be substantially mitigated through appropriate hyperparameter tuning, consistent with our broader message that HPO plays a critical role in fairness evaluation. These new results are included in Appendix F Figure 12 (page 43), highlighted in magenta.

---

> ### Author Response · Authors · 2025-11-25
> **Reply to Reviewer RqaZ (2/3)**
>
> > In general, I feel that the conclusions from the experiments confirm anecdotal observations bias mitigation (e.g., that hyperparameter tuning and model selection make a big difference), but not groundbreaking in context of existing fairness benchmarks.
>
> We respectfully argue that our findings go beyond confirming anecdotal observations and address concrete gaps in existing benchmarks such as FFB.
>
> While FFB is a valuable contribution, it evaluates older algorithms (pre-2021) using fixed training pipelines (e.g., fixed Adam optimizer with 0.001 learning rate). Other benchmarks (MEDFAIR, ABCFair) and some methodological studies similarly fix the optimizer or search over only a narrow hyperparameter range. This practice can lead to unfair or misleading comparisons and hinder understanding of the true performance of both baselines and debiasing methods. In contrast, NH-Fair conducts more systematic hyperparameter sweeps and quantified analyses, challenging this practice and calling for more rigorous experimentation in future research. Our results further yield actionable insights for model developers and help reduce the need for costly hyperparameter searches.
>
> FFB focuses on in-processing group fairness methods, whereas NH-Fair covers pre-processing, in-processing, and post-processing methods (referred to as data-centric and algorithmic methods in the paper). Building on this, we introduce a two-stage, fairness-without-harm model selection procedure and show that RandAugment, a simple and generic method, consistently delivers fairness gains without utility loss. This provides an immediately actionable, lower-cost strategy for practitioners
>
> We also discussed key differences from existing fairness benchmarks in the introduction (lines 70 - 81, page 1) and the related work section (Appendix A.4, page 19). Existing benchmarks either focus on classical vision models (FFB, MEDFAIR, ABCFair) or on vision–language models (CARES, VLBiasBench) in isolation, often in narrow domains or with smaller models. NH-Fair instead unifies evaluation across supervised vision models, VLMs, and LVLMs up to 109B parameters within the same experimental and metric framework, enabling cross-modality comparisons. We find that large-scale pretrained LVLMs are not inherently more fair and that simply increasing LVLM size does not guarantee improved fairness. This leads to practical guidance for model selection in real-world applications: switching model families can yield larger fairness gains than merely testing different sizes of a single model.
>
> > Certain conclusions in hyperparameter choice are qualitative (e.g., optimizer choice and learning rate are more important than other hyperparameters) and could have benefitted from a quantified sensitivity metric.
>
> Following your suggestions, we have added quantified metrics to Appendix C *Additional Experiments* tables. Specifically, we have reported the standard deviation of performance when varying one class of hyperparameters at a time (optimizer, batch size, or model size) for all datasets and metrics. These results are highlighted in orange. We have also added a new Appendix section C.1.6 (Sensitivity Analysis, page 33) and a new Table 13 (page 35) summarizing this analysis.  We observe that optimizer choice accounts for the largest variability in most cases, which confirms our observation. Moreover, while batch size can affect certain datasets, its search space is typically much larger than that of the optimizer, making optimizer tuning a more practical first priority. We also provide explanations of suggestions in this new section.

---

> ### Author Response · Authors · 2025-11-25
> **Reply to Reviewer RqaZ (3/3)**
>
> > I wonder if combining data curation and algorithmic mitigation can produce the best trade-off between utility and accuracy? Should future research explore the intersection between both, or prioritize data-centric methods as suggested in the paper?
>
> We agree that exploring the intersection of data-centric and algorithmic methods is a critical direction and we mentioned intersection is a promising solution to achieve better fairness in the limitations and future directions section.  To further address your question, we conducted new experiments combining RandAugment with three algorithmic methods (GapReg, MCDP, and GroupDRO) on the UTKFace and HAM10000 (Due to time limits, we can not sweep all methods and datasets). The results, reported in Appendix F Table 22 and highlighted in orange, show that hybrid data-centric + algorithmic approaches can sometimes deliver better trade-offs: for example, on UTKFace, RandAug+MCDP and RandAug+GroupDRO simultaneously improve all utility and fairness metrics. However, the benefits are dataset and method-dependent and not guaranteed.
>
> For model developers, a good starting point is to establish strong data-centric baselines such as RandAug (together with careful ERM tuning and model selection), and then explore algorithmic debiasing on top, given the extra complexity and tuning effort typically associated with these methods. For future research, we view both data-centric and algorithmic approaches as important, since they correspond to different deployment scenarios. Hybrid evaluations could be included in experiments to better understand the extent to which models can reduce parity gaps while maintaining utility.
>
> We also present the results below; values that outperform ERM are bolded.
> | Dataset  | Metric | ERM | GapReg Base | GapReg +RA | MCDP Base | MCDP +RA | GroupDRO Base | GroupDRO +RA |
> |----------|--------|------|--------------|--------------|------------|------------|----------------|----------------|
> | UTKFace | ACC   | 92.75 ± 0.54 | 92.53 ± 0.88 | 91.43 ± 1.07 | 92.49 ± 0.72 | **92.92 ± 0.83** | 92.45 ± 0.39 | **92.86 ± 0.40** |
> | UTKFace | Worst | 91.78 ± 0.61 | 91.70 ± 0.95 | 90.73 ± 1.14 | 91.63 ± 1.14 | **92.20 ± 0.91** | 91.41 ± 0.62 | **92.17 ± 0.41** |
> | UTKFace | Gap   | 2.26 ± 0.65  | **1.91 ± 0.59** | **1.63 ± 0.49** | **2.00 ± 0.99** | **1.66 ± 0.69** | 2.44 ± 0.99 | **1.60 ± 0.86** |
> | UTKFace | EqOdd | 97.62 ± 0.53 | **98.10 ± 0.61** | **97.88 ± 0.52** | **98.04 ± 0.97** | **98.05 ± 0.69** | 97.06 ± 0.63 | **98.31 ± 0.55** |
> | UTKFace | DP    | 94.55 ± 1.20 | **95.30 ± 1.62** | **95.82 ± 1.55** | **95.80 ± 0.64** | **96.42 ± 1.68** | **94.78 ± 1.85** | **95.15 ± 0.98** |
> | HAM10000 | AUC   | 88.35 ± 1.83 | 84.97 ± 3.18 | 85.55 ± 2.27 | 82.96 ± 1.84 | 87.33 ± 2.12 | 87.66 ± 2.72 | 87.08 ± 1.85 |
> | HAM10000 | Worst | 84.68 ± 2.02 | 82.57 ± 3.66 | 81.60 ± 2.42 | 80.29 ± 1.70 | 84.35 ± 1.67 | 83.98 ± 3.00 | 83.96 ± 2.42 |
> | HAM10000 | Gap   | 4.11 ± 2.08 | **3.07 ± 2.34** | 4.85 ± 2.48 | 3.10 ± 2.25 | 3.35 ± 2.39 | 4.98 ± 2.51 | 3.37 ± 2.55 |
> | HAM10000 | EqOdd | 88.17 ± 3.10 | **98.15 ± 3.71** | **96.98 ± 1.24** | **99.52 ± 0.95** | **92.84 ± 5.89** | **90.94 ± 5.33** | 87.70 ± 5.66 |
> | HAM10000 | DP    | 82.22 ± 4.78 | **96.74 ± 6.52** | **97.40 ± 2.72** | **99.58 ± 0.85** | **86.56 ± 6.11** | **83.86 ± 5.70** | 80.31 ± 6.96 |

---

### Author Response · Authors · 2025-12-01
**Rebuttal Summary**

We sincerely thank the reviewers for their thorough feedback and the ACs for their time and efforts in managing the review process. We have updated the manuscript with the required changes highlighted in color. We are encouraged that our responses have addressed the concerns of Reviewers ByuY and dLUc. Below, we summarize the major improvements made during the rebuttal.

1. **Reviewer RqaZ  (initial rating 4, no further discussion)**

   - **Benchmark is limited to closed-set prediction for LVLMs**: We have added new open-ended generation experiments (**Appendix C.3.3**). Results are consistent with our previous experiments that simply increasing model size does not guarantee improved fairness.
   - **Performance seems saturating**: The seemingly “saturated” performance is primarily a result of our rigorous hyperparameter optimization (HPO) and model selection strategy. The disparities on the same “fairness-hard” tasks (same target and sensitive attributes) in prior benchmarks are substantially reduced by our extensive HPO. In NH-Fair, we specifically targeted “fairness-hard” subsets, selecting tasks that showed relatively larger disparities to incorporate into the benchmark. We have also added a new intersectional group experiment, which is also “fairness-hard” (**Appendix F Figure 12**). While intersectional gaps are larger, they are still mitigated by appropriate HPO.
   - **Not groundbreaking to existing benchmarks**: We address the weaknesses of existing benchmarks (e.g., evaluation of older algorithms, fixed training pipelines, limited bias mitigation method types, and limited model coverage). We conduct systematic hyperparameter sweeps and quantified analyses for fairer comparison, include a broader range of bias mitigation methods, and span classical vision models to large vision–language models up to 109B parameters under a single protocol. We also introduce a new model selection method to achieve simultaneous utility and fairness gains, targeting both academic and industrial audiences with actionable insights.
   - **Quantified Sensitivity**: Addressing the request for quantification, we have added a sensitivity analysis (**Appendix C.1.6**) reporting the deviation of performance when varying model training choices. This further confirms our findings in Appendix C.1.
   - **Hybrid Methods**: We have evaluated combinations of RandAugment with GapReg, MCDP, and GroupDRO (**Appendix F, Table 22**). Results show that while hybrid methods can offer gains, they are dataset-dependent, whereas RandAugment remains a robust, low-cost baseline.
2. **Reviewer ByuY (initial rating 4, increased to 6 on Nov 26 2025)**

   - **Wording & Presentation**: We have significantly improved the manuscript based on suggestions. We have converted tables into plots (W4), added statistical testing in Sec. 4.2 (W6), updated ethical statements (W8), introduced mitigation method mechanisms (W9), and polished the manuscript for clarity (W11-12, W14-19).
   - **Additional Experiments**: We have added open-ended generation tasks (**Appendix C.3.3**)to extend our study of fairness across varying LVLM sizes (W2, W5) beyond conditioned generations. We also included a new post-processing method, OxonFair (**Table 2**), to complement the existing baselines (Q3).
   - **Clarifications**: We have discussed the relationship between our benchmark and tailored diagnosis (W1), and refined the discussion on fairness vs. domain generalization dataset (W3). We have also clarified our dataset choices (W7, Q1), model selection methods (Q2), and other experimental details (Q4-8)
3. **Reviewer ZcM2 (initial rating 6, no further discussion)**

   - **Intersectional Analysis**: We have added an intersectional group experiment (**Appendix F Figure 12**), showing that while intersectional gaps are larger than single-attribute gaps, they can still be mitigated through appropriate hyperparameter choices, reinforcing our core message about the importance of rigorous baselines with HPO.
   - **Budget-friendly HPO**: We explicitly summarized a "budget-friendly" recipe in the Introduction (line 90 - 114 *Takeaways*) and illustrated it in detail in Section 4.1.
4. **Reviewer dLUc (initial rating 8, increased confidence and contribution on Nov 25 2025)**

   - **Quantifying "Fairness Without Harm"**: We have updated **Table 2** to explicitly classify models into "Optimal" vs. "Sub-Optimal" zones. We confirm that RandAugment consistently falls into the Optimal Zone (improving fairness without sacrificing utility) on both validation and test sets.
   - **Limitations and Future Works**: We discussed these limitations in **Appendix F** alongside future directions for other fairness notions and LVLM finetuning.

Again, we appreciate the Area Chairs and all reviewers. Your valuable feedback and constructive suggestions have made this paper stronger and more accessible to a broader audience.

---

### Meta-Review · Area_Chair_F6gj · 2026-01-05

**Summary:**

After the rebuttal, the final scores of the paper are 4, 6, 6, and 8. Three reviewers provided positive evaluations of the work. The only low-scoring reviewer did not participate in the discussion; however, the authors addressed that reviewer’s concerns by adding substantial experimental results in the rebuttal. I believe the authors’ responses adequately resolve the remaining concerns, and therefore I recommend acceptance.

**Reviewer Concerns:**

I think Reviewer RqaZ's concerns are addressed by the rebuttal.

**Reviewer Scores:**

I think Reviewer RqaZ would increase the score to weak accept if participate fully in the discussion.

---

### Decision · Program_Chairs · 2026-01-26

Accept (Poster)